**Investigation**

# Predicting the functional impact of single nucleotide variants in *Drosophila melanogaster* with FlyCADD

Julia Beets (ID) ,[1,*] Julia Höglund (ID) ,[2,3] Bernard Y. Kim (ID) ,[4] Jacintha Ellers (ID) ,[1] Katja M. Hoedjes (ID) ,[1,*] Mirte Bosse (ID) [1,3,*]

[1]Amsterdam Institute for Life and Environment, Vrije Universiteit Amsterdam, Amsterdam 1081 BT, The Netherlands
[2]Centre for Palaeogenetics, Dept of Zoology, Stockholm University, Stockholm 106 91, Sweden
[3]Animal Breeding and Genomics Group, Wageningen University & Research, Wageningen 6700 AH, The Netherlands
[4]Department of Ecology and Evolutionary Biology, Princeton University, Princeton, NJ 08544, United States

*Corresponding authors: Julia Beets, Amsterdam Institute for Life and Environment, Vrije Universiteit Amsterdam, Van der Boechorststraat 3, Amsterdam 1081 BT, The Netherlands. Email: j.beets@vu.nl; Katja M. Hoedjes, Amsterdam Institute for Life and Environment, Vrije Universiteit Amsterdam, Van der Boechorststraat 3, Amsterdam 1081 BT, The Netherlands. Email: k.m.hoedjes@vu.nl; Mirte Bosse, Amsterdam Institute for Life and Environment, Vrije Universiteit Amsterdam, Van der Boechorststraat 3, Amsterdam 1081 BT, The Netherlands. Email: m.bosse@vu.nl

Understanding how genetic variants drive phenotypic differences is a major challenge in molecular biology. Single nucleotide polymorphisms form the vast majority of genetic variation and play critical roles in complex, polygenic phenotypes, yet their functional impact is poorly understood from traditional gene-level analyses. In-depth knowledge about the impact of single nucleotide polymorphisms has broad applications in health and disease, population genomic, and evolution studies. The wealth of genomic data and available functional genetic tools make *Drosophila melanogaster* an ideal model species for studies at single nucleotide resolution. However, to leverage these resources for genotype–phenotype research and potentially combine it with the power of functional genetics, it is essential to develop techniques to predict functional impact and causality of single nucleotide variants. Here, we present FlyCADD, a functional impact prediction tool for single nucleotide variants in *D. melanogaster*. FlyCADD, based on the Combined Annotation-Dependent Depletion (CADD) framework, integrates over 650 genomic features—including conservation scores, GC content, and DNA secondary structure—into a single metric reflecting a variant's predicted impact on evolutionary fitness. FlyCADD provides impact prediction scores for any single nucleotide variant on the *D. melanogaster* genome. We demonstrate the power of FlyCADD for typical applications, such as the ranking of phenotype-associated variants to prioritize variants for follow-up studies, evaluation of naturally occurring polymorphisms, and refining of CRISPR-Cas9 experimental design. FlyCADD provides a powerful framework for interpreting the functional impact of any single nucleotide variant in *D. melanogaster*, thereby improving our understanding of genotype–phenotype connections.

Keywords: single nucleotide polymorphism; *Drosophila melanogaster*; impact prediction; annotations; genotype-to-phenotype; GWAS; genome editing; evolutionary constraint; purifying selection

## Introduction

Unraveling the causative relationships between phenotypes, genes and their variants represents a significant challenge in molecular biology, especially at the resolution of single nucleotides (Gallagher and Chen-Plotkin 2018; Cano-Gamez and Trynka 2020). Species typically harbor millions of single nucleotide polymorphisms (SNPs) in their genomes; for instance, over five million SNPs have been identified in the fruit fly, *Drosophila melanogaster* (Auton et al. 2015; Wang et al. 2015; Nunez et al. 2025). Genomic studies, such as genome-wide association studies (GWAS) and Evolve and Resequence (E&R) studies, have identified vast numbers of SNPs associated with traits of interest. This led to the discovery of many candidate loci associated with a broad range of phenotypes, for example Type 2 diabetes and loci linked to fitness strategies such as behavioral choices in insect pupation (Visscher et al. 2017; Wangler et al. 2017; Xue et al. 2018; Zhang et al. 2020; Uffelmann et al. 2021). Elucidating the functional effects of individual SNPs is critical for understanding how genetic variation shapes phenotypes and advances our understanding of evolutionary adaptation, population dynamics, disease susceptibility, and other fundamental biological processes (Ungerer et al. 2008; de Visser and Krug 2014). Yet, despite the unprecedented resolution and availability of genotype data, the functional impact of individual SNPs is only rarely elucidated after identification of phenotype-associated variants (Katzenberger et al. 2015; Visscher et al. 2017; Schaid et al. 2018; Hoedjes et al. 2022, 2023; Perlmutter et al. 2024).

There are several technical and biological challenges that complicate interpretation of genetic variation at single nucleotide level. Many of the SNPs identified in association studies are genetically linked variants that are not under selection themselves but merely show allele frequency (AF) changes by hitchhiking with loci that are under selection (Smith and Haigh 1974; Gallagher and Chen-Plotkin 2018; Cano-Gamez and Trynka 2020). Epistatic interactions between loci, environmental factors and additive effects of SNPs can further complicate the interpretation of the

effect(s) of individual SNPs (Huang et al. 2012; Yashin et al. 2012; Gallagher and Chen-Plotkin 2018). Statistical identification of SNPs associated with traits can lead to false positives if non-causal SNPs appear significant due to linkage. False negatives can occur if for example true causal variants are masked by complex genetic architecture. As a result, statistical associations do not always reflect the context in which variants occur and, therefore, cannot reliably be taken as an indication for a functional relationship between a variant and a phenotype. For understanding any genotype–phenotype link it is imperative to distinguish causal SNPs from (linked) neutral loci, yet this remains a significant challenge (Smith and Haigh 1974; Franssen et al. 2015; Wang et al. 2022). Developing techniques and tools to assess the functional impact of SNPs is therefore essential for advancing genotype–phenotype research at SNP-level.

Functional genetic tests such as gene knockdown or knockout through RNAi or CRISPR-Cas9 are typically applied to study the function of genetic loci of interest (Akhund-Zade et al. 2017; Parker et al. 2020; Mokashi et al. 2021; Hoedjes et al. 2022, 2023; Wolf et al. 2023). These techniques are often applied at the gene-level, meaning that they focus on (nearby) genes associated with the SNP of interest, but do not directly test the candidate SNP(s) (Mokashi et al. 2021; Hoedjes et al. 2022, 2023; Perlmutter et al. 2024). Such gene-level approaches might not identify effects of the targeted SNPs but rather all phenotypic effects that a gene might have (Hoedjes et al. 2022, 2023). As different SNPs within the same gene could have different functional impacts or pleio-tropic effects, it is important to understand the functional impact of SNPs instead of genes to understand phenotypic variation (Zhang and Yang 2015; Mokashi et al. 2021; Hoedjes et al. 2023). Precise genome editing, for example using CRISPR-Cas9, enables targeted investigations at single nucleotide resolution, but these methods are costly and time-consuming so only few SNPs can be tested. This is one of the reasons why experimental validation of candidate SNPs remains scarce even though functional validation provides the most direct evidence for SNP function (de Visser and Krug 2014; Mokashi et al. 2021; Hoedjes et al. 2023; Perlmutter et al. 2024). To be able to focus functional validation on the most promising SNPs, without spending time and resources on SNPs without functional impact, it is critical to narrow down the set of candidates based on expected functional impact.

Computational approaches can be instrumental in prioritization of candidate SNPs prior to functional studies. Existing computational approaches for predicting the effects of SNPs include SIFT (Ng and Henikoff 2001), PolyPhen-2 (Adzhubei et al. 2010), VEP (McLaren et al. 2016), SnpEff (Cingolani et al. 2012), ANNOVAR (Wang et al. 2010), AlphaMissense (Cheng et al. 2023), ProteoCast (Abakarova et al. 2025), and GPN-MSA (Benegas et al. 2023). These tools are each based on a different narrow set of genomic annotations—primarily conservation scores, genomic structure, and locus proximity. Moreover, they differ in their scale of application, focusing on the whole genome, (nearby) genes, or coding sequence (Wang et al. 2022). Studies on SNP impact typically integrate insights from multiple tools to take advantage of various annotations (Gibert et al. 2017; Hall et al. 2019). Yet, many additional annotations, such as regulatory elements, amino acid changes, chromatin states, and DNA secondary structures, remain underutilized, and could provide essential insights for predicting the functional significance of genetic variants at single-nucleotide resolution (de Visser and Krug 2014; Öztürk-Çolak et al. 2024).

A promising computational approach that combines a broad range of annotations to predict functional impact of genetic variants is the Combined Annotation-Dependent Depletion (CADD)

framework (Kircher et al. 2014). The CADD framework is a first of its kind tool integrating diverse annotations into a single metric reflecting predicted functional impact of any SNP across the genome. CADD differs from other VEPs like SIFT (Ng and Henikoff 2001) or REVEL (Wang et al. 2022), which have been developed based on pathogenic variants in (mostly human) coding sequence. Although these VEPs outperform CADD on amino acid sequence variants, CADD is able to distinguish neutral from impactful variants across the entirety of the genome (Livesey and Marsh 2023). A unique feature of CADD is that it is not restricted to SNPs in coding regions, gene-related information, or specific SNP categories by using a large set of diverse annotations. CADD was initially applied to the human genome, and subsequently CADD models were specifically designed for chicken, mouse, and pig genomes (Kircher et al. 2014; Groß et al. 2018; Groß, Bortoluzzi, et al. 2020; Groß, Derks, et al. 2020; Schubach et al. 2024). These different species-specific models have shown that the CADD framework can be applied to different species provided that high-quality genome annotations are available (Derks et al. 2021; Wang et al. 2022; Boshove et al. 2024; Schubach et al. 2024; Speak et al. 2024).

While the CADD framework has been successfully applied to predict functional impact of variants in several species, it has not yet been extended to insects. The fruit fly *D. melanogaster* is a powerful genetic model, for which a wealth of genotyping data, high-quality genomes, and single-nucleotide resolution annotations are available, including nucleotide conservation scores and regulatory domain annotations required for the development of a species-specific CADD (Wangler et al. 2017; Kim et al. 2024; Öztürk-Çolak et al. 2024; Nunez et al. 2025). Moreover, the rich repertoire of gene editing approaches available for the *Drosophila* model system provides powerful opportunities to functionally validate genotype–phenotype links at SNP-level, for example by applying CRISPR-Cas9 (Hoedjes et al. 2023; Perlmutter et al. 2024). Functional impact predictions can guide the design of genetic tests, making a species-specific CADD for *D. melanogaster* a valuable tool to further our understanding of how genetic variation translates into phenotypic diversity by integrating diverse annotation with location-based information, which can provide crucial insights into the function(s) of SNPs.

Here, we introduce FlyCADD, the first adaptation of the CADD framework to an insect genome, applied to *D. melanogaster*. FlyCADD combines the available high-quality annotations into a robust and versatile tool for studying the functional impact of SNPs by providing impact prediction scores for all possible genomic variants in the genome of *D. melanogaster*. Combined annotations include conservation scores, gene structures, regulatory elements, GC content, and DNA secondary structure, among others. Combining in total 691 features into a single impact prediction metric, FlyCADD enables researchers to prioritize SNPs based on their predicted functional impact across different contexts and in any region of the *D. melanogaster* genome.

FlyCADD offers a complementary approach when used alongside population data or experimental validation by refining the identification of potentially causal variants, thereby improving the understanding of genotype–phenotype relationships. Although FlyCADD, as a functional prediction tool for individual SNPs, cannot resolve the full complexity of genotype–phenotype relationships resulting from for example non-additive effect, epistasis or environmental effects, it can help researchers fine-scale their genotype–phenotype interpretations by distinguishing neutral from causal variants. Functional genomic studies to understand the impact of SNPs rather than genes in *D. melanogaster* are still rare, but the few precise genome editing studies done so far show unique and strong phenotypic effects of individual SNPs (Mokashi et al. 2021; Hoedjes et al. 2023;

Perlmutter et al. 2024). SNPs identified in natural populations of *D. melanogaster*, experimental strains or disease models can now be computationally studied based on predicted functional impact, with the potential to complement such experimental studies or prioritize SNPs for further (functional) studies. The FlyCADD model represents a significant step forward in integrating computational predictions into the study of genotype–phenotype link in *D. melanogaster*.

We describe the FlyCADD impact prediction model and demonstrate that impact predictions for SNPs can improve interpretation of SNPs and experimental design. FlyCADD scores have been validated using experimentally tested point mutations with lethal outcome. To illustrate the usability of FlyCADD, we applied the FlyCADD impact prediction scores (i) to study naturally segregating SNPs, (ii) to enhance interpretation of GWAS-identified SNPs, (iii) to rank SNPs prior to functional studies, and (iv) to improve experimental design for genome editing approaches. To facilitate variant impact analysis, we have made FlyCADD impact prediction scores available for all possible single nucleotide variants in the *D. melanogaster* Release 6 reference genome. Additionally, we provide scripts and pre-processed annotations for annotating novel variants. Both the precomputed scores and the locally executable pipeline can be accessed on Zenodo (https://doi.org/10.5281/zenodo.14887337).

## Methods

FlyCADD is a logistic regressor trained on 691 genomic features to assign an impact prediction score to any single nucleotide variant in the *D. melanogaster* genome. Impact refers to a mutation that on an evolutionary time scale is likely not tolerated due to an associated fitness cost. FlyCADD is aimed at scoring the impact of single nucleotide variants; no other types of genetic variants were included in model training, testing, or application. An outline of FlyCADD development and application is illustrated in Fig. 1. We build upon the existing CADD models for human, pig, and mouse by adapting the pipeline to fit *D. melanogaster* through incorporation of *D. melanogaster*-specific annotations, sequences, alignments, and the reference genome (Kircher et al. 2014; Groß et al. 2018; Groß, Derks, et al. 2020).

### Overview of FlyCADD model training

Briefly, the model consists of a logistic regressor trained to distinguish between two sets of training data: nearly-fixed derived variants—assumed to be depleted of impactful variants—and simulated variants—containing the full spectrum of impactful variants due to the absence of purifying selection. The FlyCADD model was created across chromosomes 2L, 2R, 3L, 3R, 4, and X of the Release 6 reference genome (GCA_000001215.4). Ancestral sequence reconstruction was obtained from a Cactus whole-genome alignment of 166 *Drosophila* species in multiple alignment format (Armstrong et al. 2020) (see Supplementary File 1). This alignment was made available for FlyCADD development and represents an earlier version of the 298-way alignment described in (Kim et al. 2024). While both alignments were generated using the same methodology, the more recent version includes additional species and should be used for future analyses.

The set of derived variants was constructed by comparing the most recent common ancestor of *D. melanogaster* (melanogaster subgroup) and *Drosophila tani* (montium subgroup) to the Release 6 *D. melanogaster* reference genome (GCA_000001215.4). Details of ancestral sequence reconstruction and extraction are described below (Methods "Obtaining training datasets"). Derived variants that are fixed or nearly-fixed (AF ≥ 0.9) in *D. melanogaster*

populations across Europe and America with a different allele relative to the reconstructed ancestral genome are assumed to be depleted of deleterious variants due to purifying selection and belong to the derived variant set (proxy-benign variants). The second set of single nucleotide variants was obtained by simulating de novo variants using mutation rates based on comparison between the ancestral and reference sequence. This set of simulated variants has not experienced natural selection and is, therefore, enriched for impactful alleles compared to the set of derived variants. In the following section ("Obtaining training datasets"), construction of these training datasets and variant simulation are described in more detail.

The second step in the CADD pipeline was to annotate the derived and simulated variants with a range of *Drosophila*-specific annotations and train the logistic regression model. The annotations included a total of 38 individual annotations such as sequence information, conservation scores, chromatin states, and combinations thereof, resulting in a set of 691 features (see Supplementary File 2). The FlyCADD model was trained as a logistic regressor to classify annotated single nucleotide variants as belonging to either the derived or simulated class. The logistic regressor returns the probability of a variant belonging to the set of simulated variants and a probability of a variant belonging to the set of derived variants, which sum up to 1. For each single nucleotide variant on the reference genome, the impact prediction score representing the probability of the variant belonging to the simulated variants class is readily available. The impact score is a measure for predicted functional impact to fitness between 0 (low functional impact) and 1 (high functional impact). All annotations and impact prediction scores are 1-based. The pipeline for FlyCADD development can be found on GitHub (https://github.com/JuliaBeets/FlyCADD).

### Obtaining training datasets

A Cactus 166-way alignment of *Drosophila* species was provided for ancestral sequence reconstruction, prior to publication of the 298-way alignment (Kim et al. 2024). The Cactus alignment software reconstructs ancestral genomes for each node, calling individual bases as the maximum likelihood estimate from a Jukes–Cantor substitution model (Armstrong et al. 2020). The 166-way alignment including ancestral sequence reconstructions is available on Zenodo (https://doi.org/10.5281/zenodo.14887337). The ancestral sequence was obtained by extracting the reconstructed sequence on the sense strand relative to coordinates in Release 6 of the reference genome for the most recent common ancestor between *D. melanogaster* and *D. tani* within the 166-way alignment (see Supplementary File 1). This ancestral node was chosen as the reconstructed sequence was largest, with 16.4% of the reference sequence nucleotides having a corresponding nucleotide in the ancestral sequence, whereas the rest of the ancestral sequence consisted of gaps. Additionally, the chosen ancestral sequence showed no excessive bias toward specific chromosomes or coding sequences. Accounting for these biases ensures that derived and simulated variants are as evenly distributed across the genome as possible. A more detailed description of ancestral sequence reconstruction can be found in Supplementary File 1.

To construct the set of derived variants, reference genome alleles differing from those in the ancestral genome were identified. Subsequently, variants with an AF below 0.9 were removed to ensure that the set included (nearly) fixed variants. The *D. melanogaster* populations used to define allele frequencies included 46 genomes from 12 *D. melanogaster* populations in Europe and America (Rech et al. 2022) and >200 Drosophila Genetic Reference Panel 2 (DGRP)

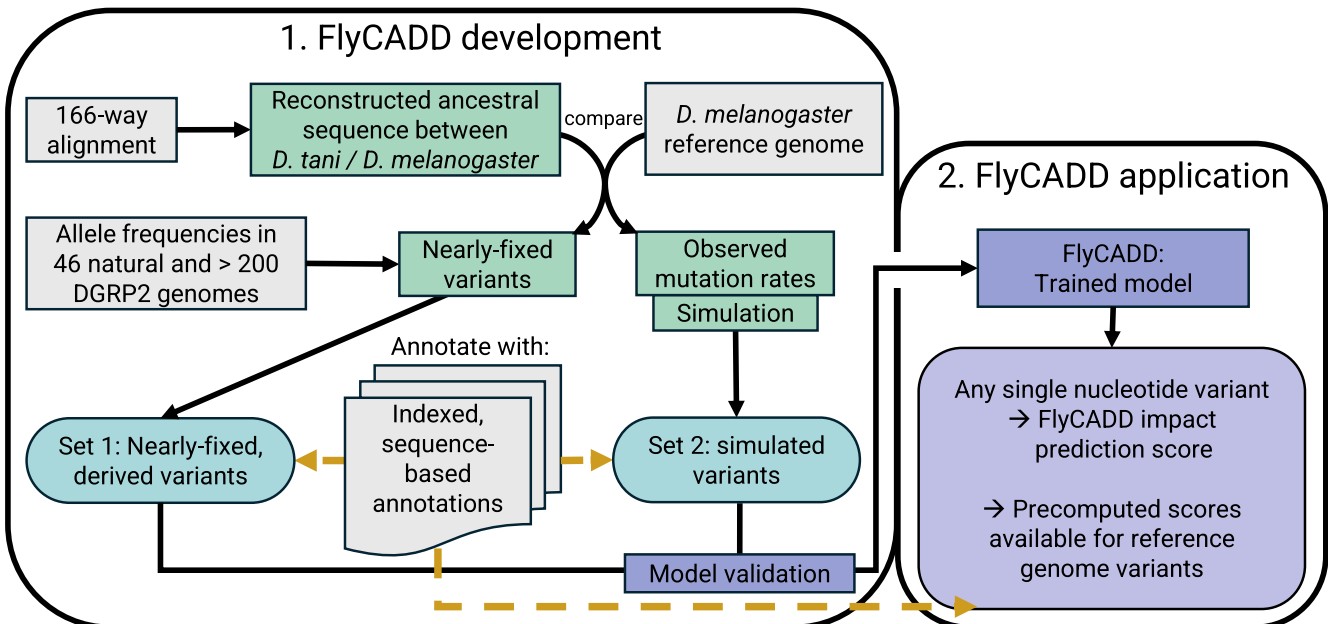

**Fig. 1.** Overview of the FlyCADD pipeline consisting of two phases: FlyCADD i) development and ii) application. Resources (gray) were used to extract information (green) that was applied to obtain the training datasets 1 and 2 (blue, rounded) for development of the FlyCADD model. These sets were annotated using 691 annotations (dashed arrows). The model is trained on these training datasets and can now be applied to any single nucleotide variant in the *D. melanogaster* genome, or precomputed scores can be used. To obtain impact prediction scores, variants of interest should be annotated (dashed arrow) using the same annotations applied to the training datasets.

lines (Mackay et al. 2012). This resulted in a set of 2,733,695 derived variants, assumed be mostly benign or neutral.

An equal number of variants was obtained through simulation using the variant simulator of the original CADD model to create a set of variants assumed to be enriched for impactful variants compared to the set of derived variants (Kircher et al. 2014). Mutation rates and CpG sites were calculated per chromosome based on comparison between the reconstructed ancestral sequence and the reference genome. These were applied to simulate variants that are absent in the reference genome but could have naturally occurred based on mutation rates in the absence of purifying selection and free from the influence of biological selection. The final mutation rates applied in the simulation can be found on GitHub (https://github.com/JuliaBeets/FlyCADD). The set of simulated de novo mutations was randomly trimmed to match the number of derived variants.

The final numbers of derived and simulated variants per chromosome are summarized in Table 1. The Y chromosome was excluded from model training due to the lack of a sufficiently reconstructed ancestral sequence required to construct the datasets of derived and simulated variants. In total, the training dataset contained 5,467,390 single nucleotide variants, equally divided over the set of derived variants and the set of simulated variants.

## Annotations

The training dataset was annotated with 691 features (38 distinct annotations and relevant combinations thereof). The Ensembl Variant Effect Predictor (VEP, v105.0) provided 13 sequence-based annotations in offline mode (McLaren et al. 2016). The proportion of variants per VEP consequence in the derived and simulated variant datasets can be found in Supplementary File 3. These were supplemented with Grantham-scores, secondary structure predictions, repeat annotations, regulatory elements, PhyloP, PhastCons, GERP, chromatin state, miRNA, coding regions, transcription factor

**Table 1.** Final numbers of derived and simulated SNPs used in training and testing the logistic regression model.

|  | Derived | Simulated |
| --- | --- | --- |
| Chromosome 2L | 594,922 | 662,282 |
| Chromosome 2R | 471,645 | 435,022 |
| Chromosome 3L | 563,599 | 577,720 |
| Chromosome 3R | 658,177 | 659,379 |
| Chromosome 4 | 25,216 | 15,959 |
| Chromosome X | 420,148 | 383,345 |
| Total | 2,733,695 | 2,733,695 |

binding sites, and *cis*-regulatory motifs. Details for all features can be found in Supplementary File 2, and the sources of these annotations are described below.

Grantham scores were incorporated based on the Grantham-matrix that annotates the chemical difference induced by the amino acid change upon variation (Grantham 1974). Secondary structure predictions for the reference genome were obtained using the *getshape*() function from DNAshapeR (Chiu et al. 2016). The included features were minor groove width, roll, helix twist, propeller twist, and electrostatic potential.

Coordinates for eight repeat types (DNA repeat elements, low complexity repeats, long interspersed nuclear elements, long terminal repeat elements, rolling circle, satellite repeats, simple repeats, and RNA repeats) were retrieved from the RepeatMasker Track of the UCSC Genome Browser (v460, last updated 2014-08-28) (Perez et al. 2025).

ReMap records of regulatory elements, PhyloP base-wise and PhastCons element conservation scores, were obtained from the UCSC Genome Browser (v460, last updated 2014-08-28) (Perez et al. 2025). ReMap records were incorporated in FlyCADD as the density of records per position. PhyloP and PhastCons scores were based on different multi-species alignment sizes: Multiz alignments of 27 insects (PhyloP/PhastCons) and 124 insects (PhyloP124/

PhastCons124). Additionally, PhyloP scores (PhyloP_all) and GERP-like scores (GerpRS_all and GERPN_all) were computed from the 166-way Cactus alignment with PHAST (Hubisz et al. 2011), using the implementation of the tools in the halPhyloPMP.py script as part of the HAL toolkit (Hickey et al. 2013), excluding the *D. melanogaster* sequence to avoid bias.

Chromatin states for BG3 and S2 cells, miRNA encoding sequence coordinates, and coordinates for coding sequences were retrieved from FlyBase (Release FB2023_03) (Öztürk-Çolak et al. 2024). Coordinates for *cis*-regulatory motifs, predicted *cis*-regulatory motifs, and transcription factor binding sites were retrieved from REDfly (v9.6.2) (Keränen et al. 2022).

Several annotations were combined with VEP annotations to create the final set of 691 features (see Supplementary Table 2 in File 2). These include all combinations of amino acid changes, variant consequences, conservation scores, ReMap scores, and distances to coding sequences. All features were scaled by their standard deviation based on variation within the training dataset. In the final annotated dataset, derived variants were annotated with "1" and simulated variants with "0".

## Logistic regression model

The annotated training dataset was split into two parts; 90% for hyperparameter tuning and model training, and 10% exclusively for model testing. To select the optimal L2 regularization parameter for the logistic regression model, we used repeated random sub-sampling validation within the 90% training dataset. Specifically, this subset was internally split five times into 90% training and 10% validation subsets for each candidate value of L2 (0.01, 0.1, 1.0, 10.0, 100.0). For each split, a model was trained and evaluated. The mean accuracy across splits was used to select the best L2 penalty.

The final Turi Create Logistic Classifier (v6.4.1) was then trained on the entire 90% training dataset, containing variants in two categories—"derived" and "simulated"—with L2 penalization set to 1.0, using the Newton solver, and a maximum of 100 iterations with *tc.logistic_classifier.create*(). A trained model was obtained for the full genome, excluding the Y chromosome due to its absence in the training dataset. The model assigns weight to each feature, representing its predictive power. The directionality of the contribution cannot be interpreted with biological meaning regarding functional impact and rather stems from the encoding of the features.

Model accuracy (the proportion of correct predictions over total predictions based on a held-out testing dataset) was evaluated using a built-in testing functionality of the logistic regression function applied to the held-out 10% of the training dataset, ensuring that none of these SNPs were used for model training. Model accuracy reflects the model's ability to distinguish between the two variant categories on unseen data drawn from the same distribution.

## Computing FlyCADD scores

The FlyCADD model can be applied to any variant in the *D. melanogaster* genome annotated with the same set of features and scaled by the standard deviation per feature within the training data. With *loaded_model.predict*(), FlyCADD generates a raw probability score for each variant, indicating its likelihood of belonging to the simulated class. FlyCADD scores range from 0, indicating no predicted functional impact, to 1, indicating high predicted functional impact. The FlyCADD model was applied to all possible single nucleotide variants on the *D. melanogaster* reference genome autosomes and X chromosome, resulting in

posterior probabilities for each variant belonging to the simulated variants set and thus being impactful. Since these precomputed impact prediction scores for all possible genomic variants include those genomic variants not observed in natural populations, predictions are unbiased by AF, function or location of known variants and solely based on variant annotations. The precomputed impact prediction scores, trained model and scripts to score novel variants are readily available on Zenodo (https://doi.org/10.5281/zenodo.14887337).

## Variant impact at codon positions

Nucleotide positions within codons differ in biological importance, providing an opportunity to evaluate the impact prediction scores (Crick 1966). Genome-wide codon positions were retrieved from Ensembl (v111) and filtered for unique transcripts on the sense strand, resulting in transcripts of 6,894 genes. Genes with multiple transcripts were excluded to avoid interaction effects. The distributions of minimum FlyCADD scores within a codon were compared per position among all transcripts and within each gene using the Mann–Whitney U-test. Statistical analyses were performed using scipy.stats (v1.12.0) in Python3, with Bonferroni corrected *P*-values (Virtanen et al. 2020).

## Impact predictions of lethal, chemically induced point mutations

Performance assessments of variant effect prediction tools often center on how well they recognize experimentally validated or phenotype-associated mutations. To assess the performance of FlyCADD on empirically tested point mutations, we retrieved data on point mutations that were chemically induced using ethyl methanesulfonate (EMS) with lethal phenotype using QueryBuilder (FlyBase FB2025_02). This resulted in 2,202 records of which 2,168 variants had a known position and base change. After excluding duplicates, FlyCADD scores were extracted and analyzed for 2,118 unique point mutations. The point mutations and their FlyCADD scores can be found on Zenodo (https://doi.org/10.5281/zenodo.14887337).

## Allele frequency of natural variants

To investigate the distribution of FlyCADD scores for naturally occurring SNPs with differing AFs, positions and AFs for over 4.8 million SNPs were obtained from the VCF file (version dest.all.PoolSNP.001.50.24Aug2024) of the *Drosophila* Evolution over Space and Time (DEST) 2.0 resource, the most comprehensive genomic resource for *D. melanogaster* genomes worldwide (Nunez et al. 2025). Only samples with quality filtering label "PASS" ($n = 529$) were included to avoid inclusion of low-quality variants.

## Results

FlyCADD predicts the functional impact of SNPs in *D. melanogaster*. Impact scores were precomputed for all possible single nucleotide variants on the Release 6 reference genome (excluding the Y chromosome). FlyCADD scores range from 0 to 1, where lower scores suggest a benign or neutral effect, while higher scores (eg >0.60) indicate predicted functional impact of the SNP. These scores allow researchers to rank SNPs based on their potential impact and assess the functional relevance of candidate SNPs. Here, we evaluate these scores based on the accuracy metrics of the model, contributions of different features, biologically relevant impact predictions, point mutations with known phenotypic effect and we show the tool's applicability across diverse use cases.

## High accuracy achieved by training the logistic regression model

The performance of the FlyCADD logistic regression model was evaluated using a held-out 10% of the training dataset (see Methods). The overall performance for FlyCADD is ~0.83 (area under the receiver operator characteristic curve [ROC]) with an accuracy of ~0.76, reflecting the proportion of correct predictions. It is important to note that FlyCADD's accuracy evaluation relies on a binary classification approach, where variants in the testing dataset were labeled strictly as 0 (neutral) or 1 (impactful), like the variants in the training dataset. However, functional impact is a continuous spectrum, and some variants labeled as 0 may still have (minor) effects. A prediction could be interpreted as misclassification during testing despite reflecting a biologically meaningful ranking, therefore, the accuracy of the model is likely to be higher when considering an impact gradient or using validation datasets for model evaluation.

## Key predictors are combined features

The strength of the CADD framework lies in its ability to integrate both individual and combined annotations, enabling more precise discrimination among variants. FlyCADD incorporated 38 individual annotations and combinations thereof, totaling 691 features, of which 581 were combinatorial features (see Supplementary File 2). In the logistic regression model, each feature is assigned a weight representing its predictive power in the impact prediction. The weights for all features can be found in Supplementary File 3. Ultimately, the impact prediction score is a combined score, considering all 691 features of a variant.

Figure 2 shows the twenty features with the highest predictive power, ie those that most strongly influence the model's ability to predict whether a variant is impactful. These features enhance predictive power but do not independently determine the predicted functional effect of a SNP. When considering the twenty most influential features in the model (Fig. 2), the composed feature names show that most key features are combinations of annotations. These combinations involve annotations on conservation scores ("_PhyloP"), proximity to coding sequence ("_relCDSpos"), and consequences (synonymous and non-synonymous, "SN_"/"NS_"). The individual annotations among the most influential features of FlyCADD are position within the protein ("protPos"), consequences stop loss, non-synonymous, synonymous and stop gain ("Consequence_"), stop codon as alternative allele (nAA_*), and conservation score (PhyloP).

Moreover, the combined features in the model capture complex interactions that enhance predictive performance beyond what individual annotations can provide. For instance, features related to stop codon loss and gain, when considered together with their position within the coding sequence, rank among the top predictors of variant impact. This reflects the biological importance of these polymorphisms, as changes affecting protein length and their location within the protein sequence strongly influence functional consequences (Lee and Reinhardt 2012). Overall, these twenty features demonstrate the highest predictive power in distinguishing impactful variants, underscoring the effect of integrating multiple annotations into a functional impact score.

## Genome-wide FlyCADD scores reflect biological relevance of polymorphisms

Next, we analyzed the precomputed FlyCADD scores for all possible single nucleotide variants on the *D. melanogaster* reference

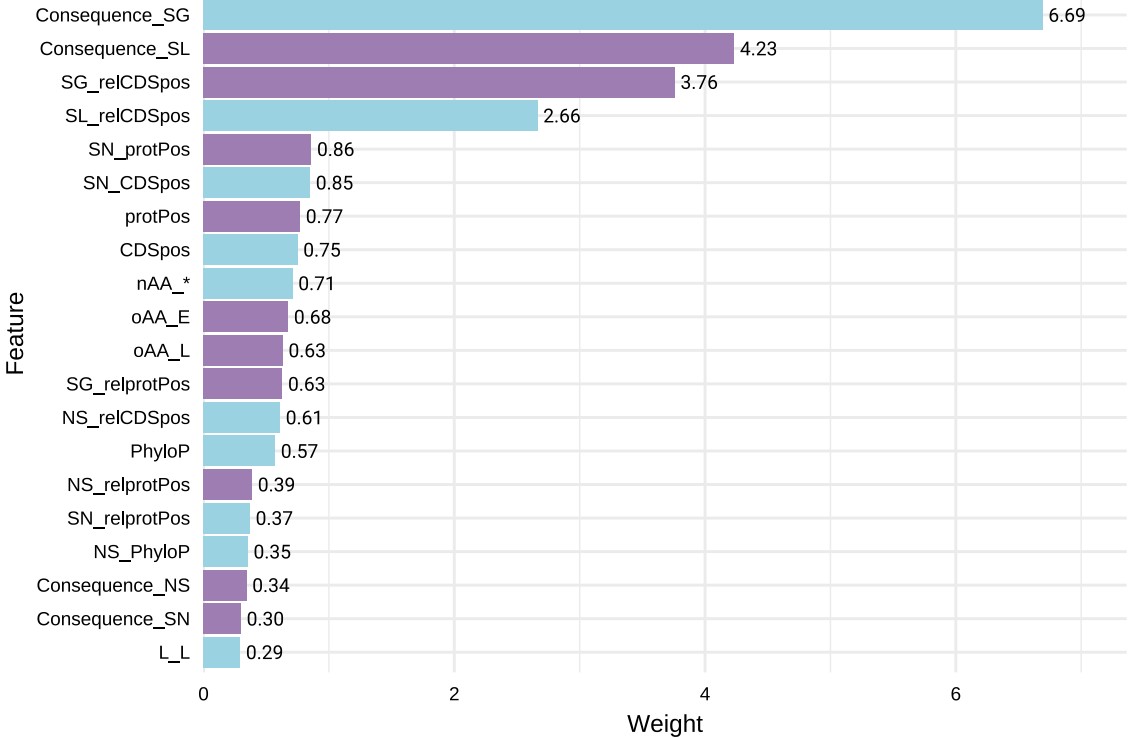

**Fig. 2.** Feature weights for the twenty features with most predictive power. The color indicates a negative (blue) or positive (purple) weight as computed by the logistic regression model. Features names "Consequence_" reflect the predicted VEP consequence derived from the Ensembl Variant Effect Predictor. Abbreviations: (rel)CDSpos: (relative) position within the coding sequence; (rel)Protpos: (relative) position within the protein; E: glutamine; L: leucine; nAA: alternative amino acid; NS: non-synonymous substitution; oAA: original amino acid; SG: stop-gain; SL: stop-loss; SN: synonymous; *: stop codon. Detailed explanations of all features can be found in Supplementary File 2.

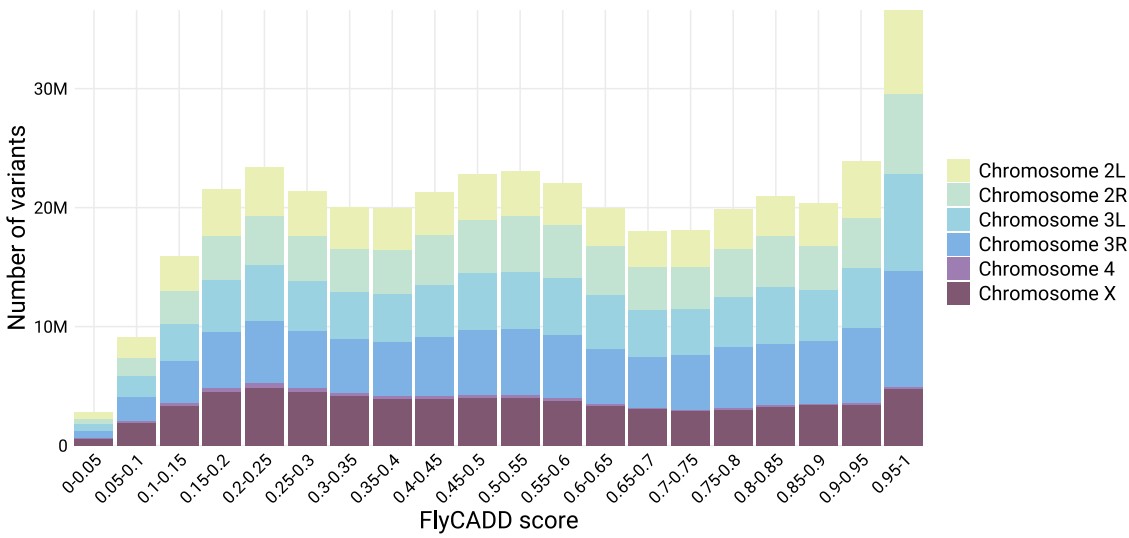

**Fig. 3.** Distribution of FlyCADD scores in the set of precomputed scores for all possible single nucleotide variants on chromosomes 2L, 2R, 3L, 3R, 4, and X of the *D. melanogaster* reference genome.

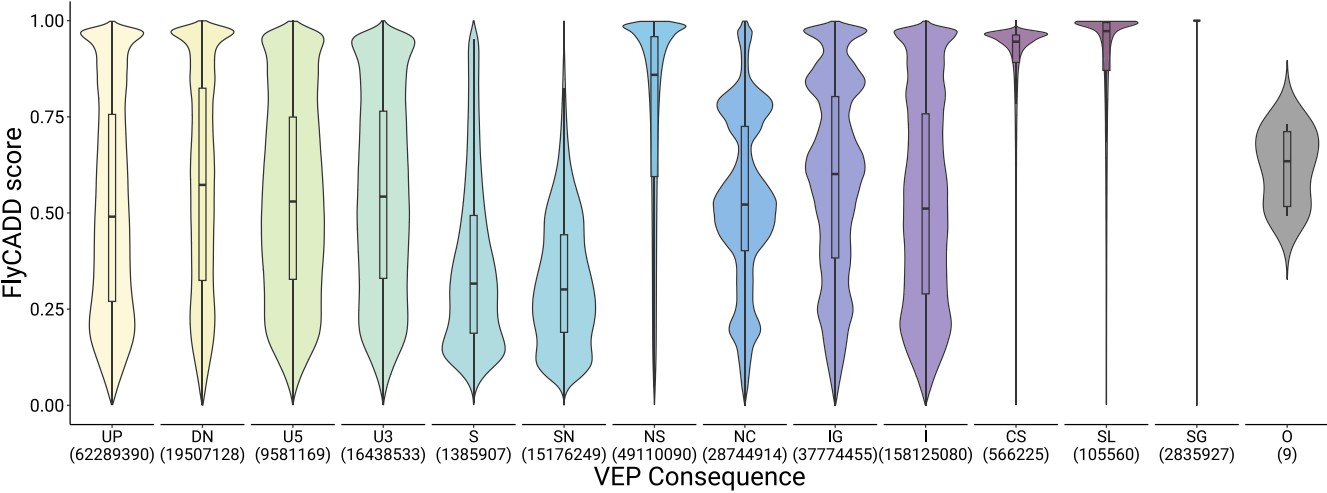

**Fig. 4.** Distribution of FlyCADD scores for all possible single nucleotide variants on the *D. melanogaster* reference genome categorized by predicted consequences from VEP. Violin plots show score density per consequence; the median and interquartile range are indicated. UP: upstream; DN: downstream; U5: 5 prime UTR; U3: 3 prime UTR; S: splice site; SN: synonymous; NS: non-synonymous; NC: non-coding exon change; IG: intergenic; I: intronic; CS: canonical splice; SL: stop loss; SG: stop gain; O: other.

genome. Importantly, this set does not represent naturally occurring variants but rather includes all theoretically possible variants relative to the reference genome. The genome-wide distribution of FlyCADD scores shows that predicted high-impact variants (>0.95) are the most prevalent, while predicted low-impact variants (<0.1) are the least frequent, a pattern that is consistent across chromosomes (Fig. 3). The limited number of low-impact scores (<0.1) likely reflects that many of the theoretically possible SNPs in this dataset do not actually segregate in natural populations. For example, the DEST genomic resource has identified ~5 million SNPs segregating in natural populations, whereas this set of all theoretically possible variants contains > 400 million variants, most of which are not occurring naturally. These non-natural SNPs likely have higher functional impact and therefore receive higher FlyCADD scores, resulting in a low proportion of low-impact variants in this dataset. Variants with such low predicted impact reflect a pattern of annotations like the

nearly-fixed, derived variants that are the proxy-benign variants in training. The set of all theoretically possible SNPs does not reflect such patterns as most are unlikely to become nearly-fixed or occur at all, resulting in high predicted impact for most variants in this dataset.

Subsequently, we assessed whether FlyCADD scores vary among variants categorized by predicted consequence (based on VEP), such as gain of a stop codon, intronic or synonymous variants, to provide insights into FlyCADD's ability to predict functional impact across classes of variants (Fig. 4). For example, it is known that non-synonymous mutations and stop codon polymorphisms are impactful as they alter the protein. In line with our hypothesis, synonymous (SN) variants have lower FlyCADD scores, whereas non-synonymous (NS) variants have higher scores, potentially due to their amino acid changing properties (Fig. 4). However, there are SN variants in the dataset with higher predicted impact and, alternatively, NS variants with lower

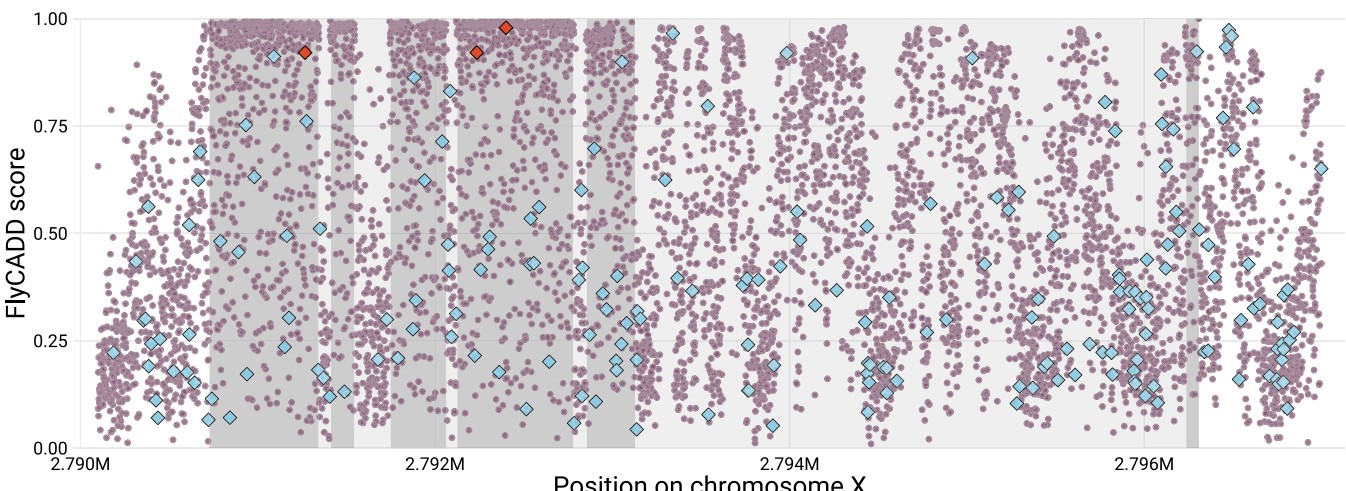

**Fig. 5.** Mean FlyCADD scores (circles) per position for all possible single nucleotide variants in the gene *white* on chromosome X, including 500 bp upstream and downstream of the gene. Exons of the gene are marked dark gray, introns marked in light gray. Naturally occurring variants retrieved from DEST2.0 are indicated with blue diamonds. Three chemically induced and phenotypically validated SNPs are indicated with red diamonds.

predicted impact are present (Fig. 4). The range of predicted impact for these types of variants is large and overlapping. Therefore, applying FlyCADD to NS or SN point mutations can help refine interpretation of the functional impact of these variants instead of solely relying on the individual VEP consequence prediction. Variants with other consequences showed a broader FlyCADD score distribution compared to SN or NS variants, reflecting the additional information captured by integrating 691 features. For example, intergenic and non-coding variants exhibited widely spread scores, highlighting how combining diverse annotations into a single metric enhances interpretation, especially in less characterized genomic regions where single annotations are often less informative. FlyCADD thus enables meaningful ranking of SNPs across the genome, including those in regions with limited prior functional insight.

## FlyCADD scores refine impact prediction within gene structure

Genes are composed of distinct functional regions, such as introns, exons, and transcription factor binding sites. We found that FlyCADD scores reflect the underlying genomic arrangement by recognizing, among others, exonic variants with generally higher FlyCADD scores compared to intronic variants when considering all possible single nucleotide variants within genes. This is illustrated in Fig. 5 for the gene *white* where, as expected, variants in exons generally receive higher FlyCADD scores compared to intronic variants. However, both regions contain SNPs that cover the full range of FlyCADD scores. Additionally, naturally occurring variants in this gene exhibit a wide spread of predicted impact (Fig. 5). Three chemically induced functional variants in *white* were described to be involved with eye color (Mackenzie et al. 1999). These variants have a predicted high impact with FlyCADD scores above 0.97, which agrees with the experimentally validated functional impact (Fig. 5). By evaluating SNP impact using combined annotations rather than relying solely on gene structure, FlyCADD enables functional impact prediction at single-nucleotide resolution rather than considering all variants within a functional domain as equivalent. Traditional approaches often infer SNP impact based on proximity to genes or annotated domains, overlooking functional differences between

variants within the same region. FlyCADD allows researchers to distinguish between neutral and functionally relevant variants within functional regions.

## Nucleotides at each codon position show differing predicted impact

To investigate the performance of FlyCADD impact predictions at single nucleotide resolution, we examined variation in FlyCADD scores across codon positions, which differ in functional importance due to their unique roles in DNA secondary structure, transcription, translation, and mutation accumulation (Crick 1966). The third codon position is generally least impactful, as variation at this site rarely results in amino acid changes (Crick 1966).

We compared the distribution of minimum FlyCADD scores of all possible variants on the three codon positions across 6,894 genes (Fig. 6a), using the lowest-scoring variant at each position to provide a conservative estimate of functional impact and avoid bias from outliers. Variants at the first and second codon position both have significantly higher FlyCADD scores compared to the third nucleotide. This is in line with expectations since the third codon position is often referred to as the "wobble" nucleotide, where base pairing is less strict and nucleotide changes often do not alter amino acids (Crick 1966). The third nucleotide has the least functional impact, followed by the first nucleotide, whereas the second nucleotide has the most functional impact (Fig. 6a). Additionally, we compared the distribution of FlyCADD scores of the three codon positions per gene (Fig. 6b).

Analysis shows that the first nucleotide in each codon has significantly higher FlyCADD scores compared to the third nucleotide in 50.4% of the genes, whereas the second nucleotide scores higher compared to the third nucleotide in 62.2% of the genes (Fig. 6b). Therefore, the third nucleotide has the lowest FlyCADD scores in most genes. The first codon position was scored significantly higher by FlyCADD compared to the second nucleotide in only 12% of the genes, indicating that the second codon position is more important than the first in many genes (48.4%). Taken together, the impact predictions indicate that third-position nucleotides are least functionally important, followed by first-position nucleotides, and second-position nucleotides have the greatest impact. This is consistent with observations across all genes (Fig. 6a) and indicates

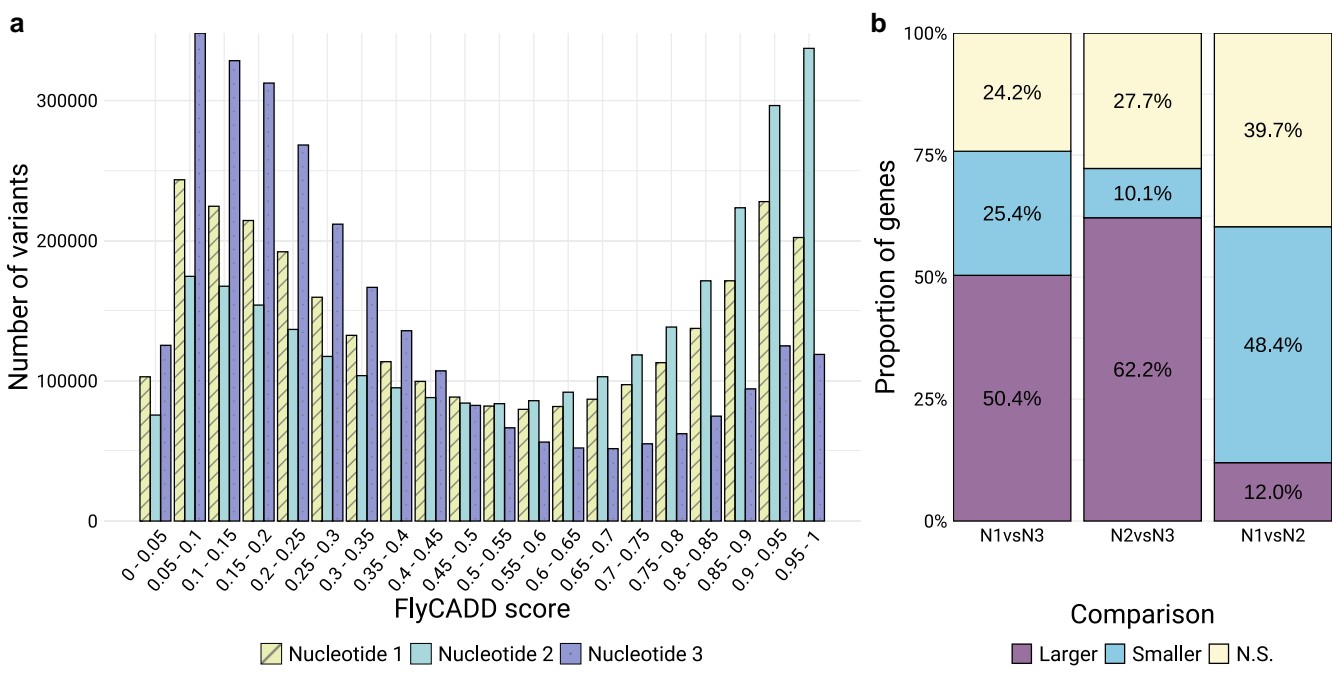

**Fig. 6.** Distribution of minimum FlyCADD scores of variants at each of the positions within the codon; a) distribution across 6,894 transcripts and b) FlyCADD scores at each nucleotide in a codon compared within each gene. The proportion of genes where the distribution of FlyCADD scores significantly differed between two nucleotide positions after Bonferroni corrected Mann–Whitney U tests is shown in panel b.

that FlyCADD scores reflect biologically expected patterns of functional impact at the single-nucleotide level.

## Lethal point mutations predicted as high impact variants

A well-curated set of both functionally impactful and neutral variants, such as those used for validation of hCADD scores (eg ClinVar [Landrum et al. 2018]), is currently unavailable for *D. melanogaster*. Moreover, the utility of natural variation for validating FlyCADD's functional impact predictions is limited because of the lack of phenotypic annotations for these variants. Therefore, we used EMS-induced point mutations with confirmed lethal outcome as a proxy for functionally impactful variants. These mutations have been experimentally tested and thus provide a relevant benchmark for evaluating predictive performance, representing model recall. A subset of the EMS point mutations has previously been part of a custom benchmarking method of the proteome variant effect predictor ProteoCast (Abakarova et al. 2025). In the absence of a validated set of neutral or benign variants, specificity of FlyCADD predictions cannot be directly calculated using gold-standard labels, but other applications of the CADD framework showed that its variant separation ability is high (Gudkov et al. 2025).

FlyCADD was developed without the use of experimental phenotype annotations, nor was it trained on known impactful mutations. To evaluate the performance of FlyCADD to predict the impact of mutations with significant phenotypic impact, we applied FlyCADD to EMS-induced point mutations with known lethal outcome retrieved from FlyBase. A total of 2,118 unique point mutations with lethal outcome were identified and FlyCADD scored 93.3% of these mutations as impactful with impact prediction scores above 0.6 (Fig. 7). 88.6% even had a predicted impact of 0.8 and higher, strongly indicating functional impact consistent with their lethality. These variants with high phenotypic severity are recognized as impactful mutations by FlyCADD, indicating

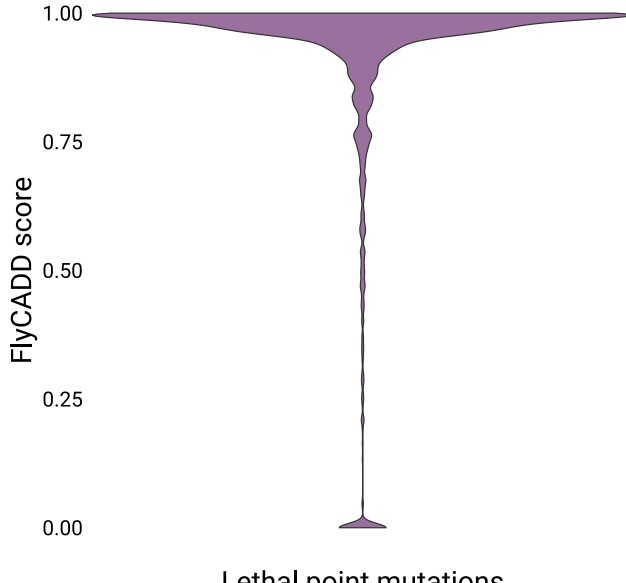

**Fig. 7.** Distribution of FlyCADD scores for 2,118 chemically induced point mutations with lethal outcome retrieved from FlyBase.

that the evolutionarily informed framework has a high recall to identify impactful variants. Several variants with a low predicted impact are also present in this dataset and might indicate variants lethal under specific environmental or genetic conditions. For example, two point mutations in the gene *kon* receive a FlyCADD score of 0 indicating no functional impact. However, they are associated with the lethal phenotypic outcome in a specific genetic background; a T to A point mutation at 2L:18492868 is lethal when occurring with a four base pair deletion and the point

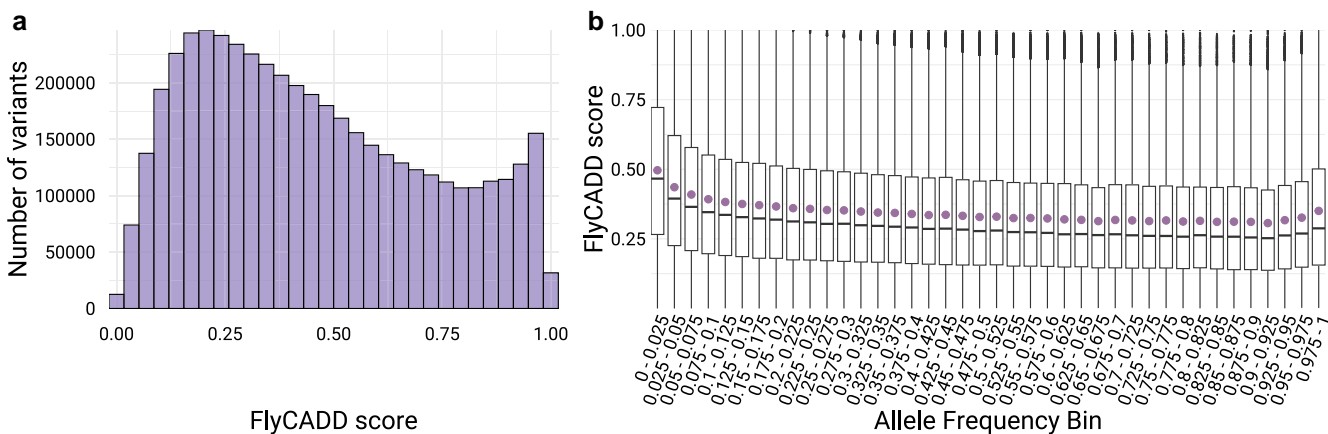

**Fig. 8.** FlyCADD scores of naturally occurring SNPs, identified in the DEST2.0 genomic resource. a) Distribution of FlyCADD scores and b) boxplots showing the distribution of FlyCADD scores across AF bins (bin size 0.025). AF represents the frequency of the non-reference allele (as mapped to the reference genome). The boxplots display the median, interquartile range and spread of FlyCADD scores. The mean FlyCADD score for variants per AF bin is indicated.

mutation at 2L:18494189 is lethal when co-occurring with other point mutations in this gene (Estrada et al. 2007; Schnorrer et al. 2007). FlyCADD scores indicate which of these point mutations is likely to be the causal factor of the lethal phenotype.

## Use cases for FlyCADD scores

Approaches such as GWAS and E&R often yield large numbers of candidate SNPs associated with a phenotype, but these often include false positives and false negatives due to linkage or technical limitations, such as statistical power (Wangler et al. 2017). In addition, experimental validation of candidate SNPs is costly, time-consuming and not yet feasible for large panels of SNPs. The FlyCADD model generates impact prediction scores reflecting a fitness-reducing effect of SNPs, making it a promising tool for genetic research by facilitating the prioritization and interpretation of identified SNPs across a range of studies and topics. Here, we outline four key applications of FlyCADD to study *D. melanogaster* genomic variation at single nucleotide resolution, each illustrated with a specific example demonstrating the utility of impact prediction scores in various research contexts.

### Impact of SNPs in natural populations of D. melanogaster

Natural populations of *D. melanogaster* exhibit extensive genetic diversity, with large numbers of segregating alleles continuously being identified (Nunez et al. 2025). While this natural variation is a driver of adaptation, genetic variation without functionally relevant impact is likely to circulate in populations too. Methods for identification of functional loci from population genomic resources are often not focused on individual variants, but rather on sets of SNPs or nearby genomic regions (Harr et al. 2002; Jha et al. 2015). The FlyCADD model can filter out hitchhiking loci or variants with negligible functional effects, and identify high-impact SNPs in population genomics studies. Furthermore, FlyCADD scores offer a basis for identifying ranges to classify variants by their potential impact, providing insights into the tolerability of variation within natural populations.

The genomic resource "DEST" 2.0 contains high-quality whole-genome sequence information of 529 *D. melanogaster* populations worldwide (Nunez et al. 2025). When applying FlyCADD to the ~4.8 million SNPs from this dataset, we observed a mean FlyCADD score of 0.45, with an increased mean impact prediction score (0.50) observed for rare alleles (AF < 0.05) (Fig. 8). When

excluding rare alleles, the mean FlyCADD score is 0.36. This data is indicative of a higher predicted functional impact for rare alleles. Naturally occurring variants show a distribution of FlyCADD scores across the full range (0 to 1), irrespective of AF. Outliers with FlyCADD scores exceeding 0.80 (high impact) are more commonly observed among rare alleles, while becoming less frequent as AF increases. The presence of both low and high impact variants highlights the potential value of FlyCADD in differentiating functionally relevant variants from neutral loci within population genomic resources like DEST2.0.

Based on the distribution of FlyCADD scores of naturally occurring SNPs (Fig. 8), we propose a range for interpreting functional impact of naturally occurring SNPs. Variants with scores between 0 and approximately 0.40 appear to be tolerated in natural populations as they are frequently present, whereas those with scores above 0.60 emerge as candidates for further study due to their high potential for functional impact. However, the degree to which functional impact influences populations may be shaped by ecological and selective pressures.

### FlyCADD-based ranking compared to P-value ranking of SNPs from GWAS

Currently, *P*-values are the most common metric for prioritization of phenotype-associated candidate SNPs from GWAS. Relying solely on *P*-values for ranking GWAS SNPs is prone to confounding factors such as linked variants, statistical limitations, and environmental factors, which are well-known to influence these *P*-values (Korte and Farlow 2013; Fadista et al. 2016; Chen et al. 2021). Therefore, *P*-values give a first insight into candidate loci, but identification of causal SNPs remains challenging and subject to varying cut-offs (Korte and Farlow 2013; Cano-Gamez and Trynka 2020). As we demonstrate below, FlyCADD scores can complement *P*-value ranking derived from association studies by providing an impact prediction score per SNP and potentially offer additional insight into evolutionary fitness consequences of candidate SNPs.

A GWAS study by Katzenberger et al. (2015) provides an opportunity to apply FlyCADD scores alongside *P*-values to assess the functional impact of SNPs. The study investigated the genetic basis of mortality after induced traumatic brain injury in DGRP lines, identifying 216 significantly associated candidate SNPs (*P* < 1e-5). Although the use of CRISPR technology was suggested to

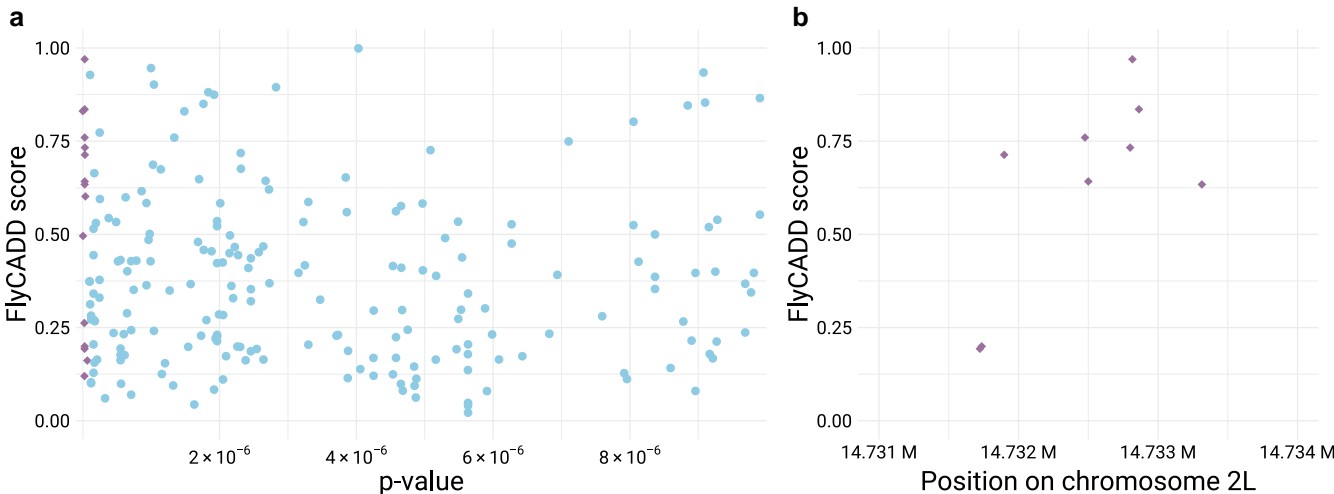

**Fig. 9.** FlyCADD score and *P*-value compared for a) 216 SNPs associated with death after brain trauma in GWAS by Katzenberger et al. (2015). The fifteen SNPs with the lowest *P*-values are highlighted (purple diamonds). b) Nine of these SNPs with the lowest *P*-values are in a 2,000 bp intergenic region of chromosome 2L and scores indicate predicted functional impact.

understand how each of these SNPs would affect the phenotype, no functional validation was performed (Katzenberger et al. 2015).

We applied FlyCADD to complement *P*-values to rank the candidate SNPs based on predicted functional impact. Comparison of the GWAS *P*-values to the FlyCADD scores did not reveal a significant correlation between the two metrics (Fig. 9a). Applying FlyCADD scores showed that variants with significant *P*-values can even have a very low (<0.2) predicted impact, and alternatively, variants with lower yet significant *P*-values can have high predicted impact (Fig. 9a). For example, in the fifteen most significant candidate SNPs, 9 variants have a predicted high impact above 0.6, and six variants have a predicted low (<0.3) impact. Nine of these SNPs are in a 2,000 bp intergenic region on chromosome 2L with two of these SNPs having low predicted impact and seven having high predicted impact (Fig. 9b). This suggests the presence of both causal (potentially those with high FlyCADD scores) and hitchhiking (potentially those with low FlyCADD scores) SNPs among these candidate SNPs with significant GWAS-derived *P*-values. Therefore, FlyCADD impact prediction scores apply an additional layer of information with regards to fitness-associated traits applicable to SNPs detected through GWAS.

### Ranking to identify causal SNP(s) or eliminate hitchhiking SNPs before experimental studies

Prioritization of SNPs using impact prediction scores helps eliminate hitchhiking loci and focus follow-up studies on functionally relevant variants, which will ultimately clarify the genetic architecture of complex traits. To show how prioritization using FlyCADD scores can be applied to GWAS-identified SNPs and how this compares to experimental studies, we apply FlyCADD to the results of a GWAS on a fitness-affecting phenotype.

Natural variation in pigmentation has been observed among *D. melanogaster* populations, which can affect fitness by influencing temperature tolerance, mate choice, energy allocation, and visibility among others (Dembeck et al. 2015; Freoa et al. 2023). Moreover, experimental evolution demonstrated that darker *D. melanogaster* populations showed higher fecundity and significantly reduced lifespan compared to the less pigmented populations (Rajpurohit et al. 2016). Bastide et al. performed a GWAS identifying naturally occurring SNPs associated with female abdominal pigmentation, which resulted in seventeen significant candidate SNPs and

hundreds of non-significant SNPs associated with a darker phenotype (Bastide et al. 2013). The three most significantly associated SNPs were in the cis-regulatory region of *tan*, a pleiotropic gene involved in pigmentation, its evolution, and other traits such as vision or mating behavior, suggesting these SNPs influence pigmentation via *tan* regulation (Gibert et al. 2017; Massey et al. 2019). Transgenic lines with each one out of eight combinations of these three SNPs were screened to determine the functional impact of the SNPs in vivo (Gibert et al. 2017). This demonstrated that all three SNPs (X:9227096, X:9227061, X:9226889) showed an additive, complex epistatic effect on pigmentation with most of the variance explained by X:9227096. In addition, the allele at X:9227061, associated with the dark phenotype in GWAS, resulted in less abdominal pigmentation in transgenic lines. This effect is in the opposite direction of what the GWAS initially suggested, which Gibert et al. attributed to linkage of the SNP with the other candidate SNPs as epistasis showed a less strong effect (Gibert et al. 2017).

We applied FlyCADD to complement the GWAS results and in vivo study by ranking these SNPs based on predicted functional impact. The FlyCADD impact prediction scores for the SNPs are as follows: 0.30 for X:9227096, 0.04 for X:9227061, 0.41 for X:9226889 (Fig. 10). With FlyCADD scores between 0.30 and 0.60, two of these SNPs are predicted to have a moderate phenotypic impact. The low FlyCADD score of X:9227061 (0.04) suggests that its association to pigmentation in the GWAS was due to linkage with functionally relevant SNPs rather than a direct phenotypic effect. This aligns with the findings of Gibert et al. (2017) where functional assays on this SNP showed opposite phenotypic effects compared to the GWAS prediction. Based on FlyCADD scores, researchers would have been able to identify and potentially eliminate this SNP with little functional relevance prior to the functional validation study. The additional GWAS-identified SNPs, which have not yet been functionally validated (triangles in Fig. 10) can now be ranked to focus future functional studies on those with highest FlyCADD scores, and, therefore, highest potential impact. The prioritization of candidate SNPs using FlyCADD scores facilitates targeted functional assays, focusing on top-ranked SNPs by filtering out (possibly linked) SNPs with lower functional impact scores such as X:9227061.

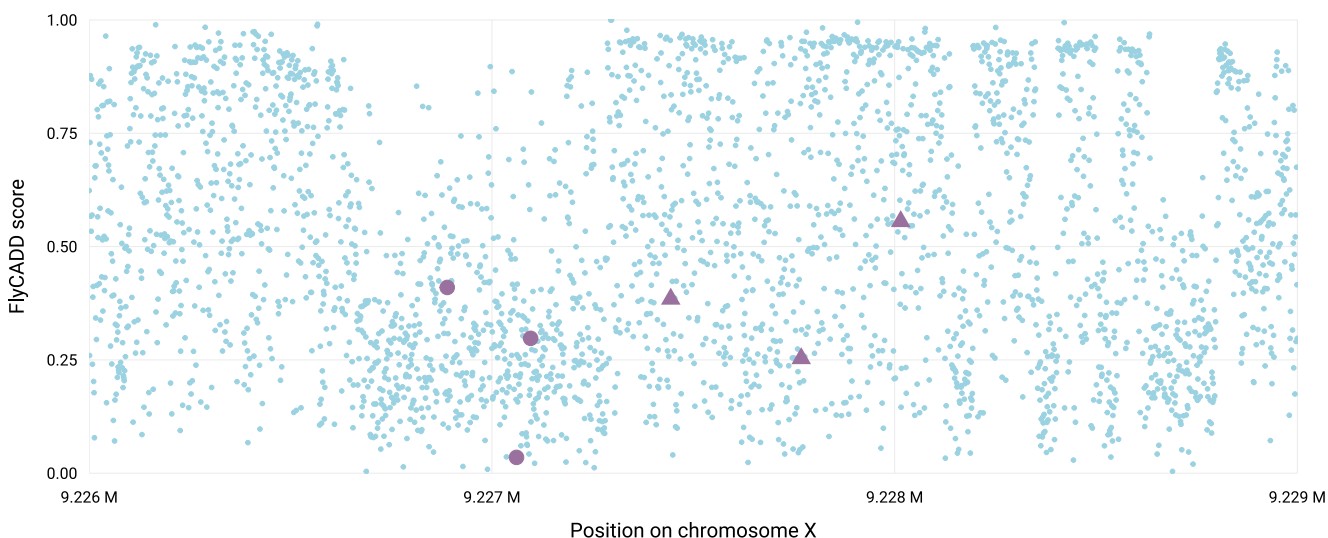

**Fig. 10.** Mean FlyCADD scores per position (blue circles) for all possible single nucleotide variants in the regulatory domain associated with *tan*, with the FlyCADD score of the GWAS identified SNPs in purple of which the SNPs indicated with circles were functionally validated, and triangles were not functionally validated.

## Evaluation of off-target effects in functional studies

Genome editing using CRISPR-Cas9 or other gene-editing technologies is widely used for functional genomics, both in vivo and in cell lines (Adli 2018). However, these techniques can introduce off-target effects, including unintended point mutations that may confound experimental results (Zhang et al. 2015). In some cases, intentional modifications in the target regions are required to increase genome editing efficiency, for example by disrupting the protospacer adjacent motif (PAM) sites essential to Cas9 (Zhang et al. 2015; Perlmutter et al. 2024). These are not off-target effects of the CRISPR-Cas9 mechanism but deliberate alterations to improve genome editing outcomes. FlyCADD can be applied in genome editing experiments to prioritize variants of interest, to evaluate the potential functional impact of intentional modifications, as well as to evaluate the potential functional impact of unintended off-target effects resulting from genome editing. This can increase the confidence that observed phenotypic effects result from the on-target modification of interest, rather than from confounding (intentional or unintended) off-target point mutations.

In their study, Perlmutter et al. (2024) applied CRISPR-Cas9 to assess functional differences between two alleles of a non-synonymous polymorphism in the gene *Metchnikowin* (*Mtk*) in *D. melanogaster*. *Mtk* is involved in microbe defense, response to wounding, neuropathology, and fitness among others (Levashina et al. 1998; Swanson et al. 2020; Perlmutter et al. 2024). Perlmutter et al. (2024) focused on infection and life-history assessments of CRISPR-Cas9 edited and control lines. In addition to the desired single nucleotide modification, multiple silent point mutations were introduced in the target region to increase efficiency. The authors deemed the effect of these intended point mutations neglectable as these were synonymous mutations with limited expression differences observed (Perlmutter et al. 2024).

We applied FlyCADD to determine whether the introduced silent mutations could have altered the results of the phenotypic screening. The SNP of interest (2R:15409024) has a predicted moderate functional impact with a FlyCADD score of 0.47, and the introduced silent mutations have FlyCADD scores of 0.52 (2R:15409016), 0.34 (2R:15409019), and 0.10 (2R:15409033). These impact prediction scores reveal that, although the intentional point mutations are synonymous, they are predicted to have an impact on fitness, with one of these silent mutations (2R:15409016) having a predicted impact higher than the predicted impact of the SNP of interest. These intended "silent" point mutations could potentially have an impact on the measured fitness-related phenotypes and therefore alter the phenotypic outcomes of the study. FlyCADD scores can now be applied during experimental design to more precisely place intended mutations and thereby avoid phenotypic effects caused by these mutations.

Additionally, FlyCADD can be utilized after genome editing by applying the impact prediction scores to sequencing data of the obtained lines to estimate whether observed phenotypic differences are influenced by unintended off-target modifications. Perlmutter et al. identified 69 SNPs differing between the edited and unedited lines, most of which were deemed cases of mapping uncertainty (Perlmutter et al. 2024). Of these, however, 14 SNPs have FlyCADD scores indicating a probability of being impactful (>0.60), therefore it is valuable to further assess mapping quality for these SNPs to make sure they do not confound the phenotypic results. The authors identified one missense SNP (2R:16854417) as a confirmed off-target mutation (Perlmutter et al. 2024). This SNP has a very high FlyCADD score (0.99), indicating that the SNP is very likely to impact fitness and potentially influenced the experimental results. This information on predicted variant impact allows for a more cautious interpretation of phenotypic measurements.

## Discussion

FlyCADD is a versatile new tool for genetics research in *D. melanogaster*, offering valuable insights into the functional impact of SNPs across the entire genome. By integrating 691 genomic features into a single impact prediction score, it enables researchers to prioritize SNPs, assess the potential phenotypic impact of natural variation, evaluate off-target effects of genome editing, and clarify patterns of causal and neutral variants based on predicted functional impact. The CADD framework is based on the assumption that (nearly) fixed derived variants are depleted of impactful mutations, while purifying selection has not affected simulated variants, and therefore, the set of simulated variants contains

variants on the full spectrum of impact (Kircher et al. 2014). It is important to note that while high impact scores indicate functional impact, they do not necessarily imply a harmful or deleterious fitness effect. Variants with a high predicted impact score have a higher probability of having an impact on fitness, either harmful or beneficial. FlyCADD can advance our understanding of the functional importance of SNPs and genotype–phenotype connections in *D. melanogaster* by providing impact scores for every SNP throughout the genome.

## Ensuring model reliability through training and validation

The CADD framework works best when rich genomic information is available, like for model species. The available sequencing datasets, high-quality reference genome, species alignments, and nucleotide-based annotations made it feasible to develop FlyCADD tailored to *D. melanogaster* based on the existing CADD framework (Mackay et al. 2012; Kircher et al. 2014; Hoskins et al. 2015; Groß, Derks, et al. 2020; Rech et al. 2022; Kim et al. 2024; Nunez et al. 2025). Ancestral sequence reconstruction was performed on a node that resulted in an ancestral sequence with minimized bias toward specific chromosomes or coding sequence, though challenges remained in reconstruction of heterochromatic regions such as the Y chromosome, repeat regions, and centromeric regions as they are often lacking in genome assemblies (Chang and Larracuente 2019). Therefore, FlyCADD is trained on the autosomes and X chromosome, excluding the Y chromosome. The reconstructed ancestral genome was sufficiently large to develop a robust predictive model trained on more than five million annotated single nucleotide variants. Our data shows that the CADD approach can be effectively extended to insect genomes, especially when high-quality genomic resources are present.

Our analyses indicate that FlyCADD has a high accuracy with ROC-AUC of 0.83 and a prediction accuracy of 0.76 on the held-out test dataset. These numbers indicate that FlyCADD performance is higher compared to the existing mouse (ROC-AUC of 0.67) and pig (ROC-AUC of 0.68) CADD models (Groß et al. 2018; Groß, Derks, et al. 2020). mCADD (mouse) and humanCADD (hCADD, [Rentzsch et al. 2019; Schubach et al. 2024]), tested on available validation datasets containing variants of known pathogenicity rather than held-out training data, reported AUC scores of >0.95. However, there is no extensive, genome-wide validation dataset available for functional impact of SNPs in *D. melanogaster* beyond the binary phenotypic class of lethal point mutations from FlyBase included in the current study. Therefore, direct comparison of model accuracy is difficult as the CADD models are species-specific. However, application of FlyCADD scores to known lethal mutations demonstrated that FlyCADD correctly predicts functional impact for experimentally tested point mutations and identifies the causal mutation when multiple point mutations co-occur. The distribution of FlyCADD scores for lethal point mutations clearly differs from the distribution for naturally occurring variants presented in this study. Naturally occurring variants harbor less impactful mutations, while lethal mutations are impactful, demonstrating the separation of lethal and non-lethal mutations through impact prediction scores. This is consistent with findings from other variant effect predictors, such as ProteoCast, which also identify the impact of lethal mutations and highlight the challenge of disentangling co-occurring genetic variants (Abakarova et al. 2025).

## Species-specific functional predictions

Species-specific models are important as, for example, mammal and insect genomes differ widely, and therefore, predictability of

functional impact of SNPs and importance of different genomic features differ between species (Huber et al. 2017). Previous CADD models demonstrated that annotations derived from the sequence and species-specific annotations contribute most to the predictions (Groß et al. 2018). Out of a total of 691 features in FlyCADD, the most important features were sequence-derived combinations of positional, functional or conservation scores, emphasizing the importance of composite features over single annotations for *D. melanogaster*. Species-specific application of the CADD framework is important to preserve differences between species and accurately predict functional impact for the species of interest. As a species-specific implementation of the CADD framework, FlyCADD enhances genotype–phenotype research by providing predictions optimized for the genomic architecture of *D. melanogaster*.

It is currently unknown whether species-specific CADD models can reliably be applied to (closely) related species, for example applying FlyCADD to *D. suzukii*, *D. simulans*, or *D. yakuba*. ChickenCADD (chCADD, [Groß, Bortoluzzi, et al. 2020]) has been applied to the pink pigeon genome, but no validation was performed to assess its accuracy on this pink pigeon genome (Speak et al. 2024). The availability of genome editing techniques and high-quality genomic resources in *D. melanogaster* and related *Drosophila* species provides opportunities to functionally validate impact prediction scores from FlyCADD within and among related species. Future research could evaluate FlyCADD's transferability to other *Drosophila* species through score lift over or by directly scoring novel SNPs, as the scripts and impact prediction scores of FlyCADD are publicly available.

## Variant effect predictors

The strength of the CADD framework lies in combining features and providing a score for functional impact of each SNP across the entire genome, which raises the question how this framework performs compared to other effect predictors (Livesey and Marsh 2023; Riccio et al. 2024; Schubach et al. 2024; Tabet et al. 2024; Benegas et al. 2025; Gudkov et al. 2025). Recent benchmarking efforts across diverse eukaryotic and model organism systems have revealed strengths and limitations among variant effect predictors, which vary widely in scope, training data, and intended application (Riccio et al. 2024). CADD was developed to predict functional impact, rather than solely pathogenicity. To date, however, most benchmarking datasets have focused on human variants with (clinically) validated functional impact (Riccio et al. 2024). When benchmarked based on these specific tasks, CADD ranks in the middle range, whereas variant effect predictors specifically designed to identify pathogenic variants or assess protein impact perform best (Livesey and Marsh 2023; Riccio et al. 2024; Schubach et al. 2024; Tabet et al. 2024; Benegas et al. 2025; Gudkov et al. 2025).

The CADD framework uses a supervised machine learning approach, which performs well in distinguishing deleterious alleles in human datasets (Gudkov et al. 2025). CADD impact prediction in coding regions is enabled through the incorporation of coding sequence annotations, whereas additional annotations provide information about non-coding regions. To improve variant effect predictions across the genome, the latest hCADD release incorporates innovations, such as protein language model embeddings (Schubach et al. 2024; Gudkov et al. 2025). Recent benchmarks have highlighted the rising performance of unsupervised methods, indicating benefits of effect predictors leveraging for example protein language models like ESM-1v (Livesey and Marsh 2023). CADD, however, has the potential of species-specific prediction and continuous development (Schubach et al. 2024).

FlyCADD, specialized for *D. melanogaster*, adapts variant effect prediction to a key model organism, expanding beyond the human-focused variant effect prediction landscape. Its comparative performance against other predictors has not yet been characterized. However, the underlying CADD framework is widely used and benchmarked (Schubach et al. 2024). Currently, no validation datasets are available for *D. melanogaster* that minimize bias toward specific tools and circularity while including both neutral and impactful variants. A comprehensive benchmarking study comparing FlyCADD to alternative methods, including those that rely on fewer features or unsupervised learning, would provide valuable insights into its predictive power and clarify its utility for specific variant assessments in *Drosophila*.

## Genome-wide functional insights from FlyCADD

CADD applied to the human genome is widely used for analysis of disease-related polymorphisms, while pCADD and chCADD have been applied to agriculture and conservation genomics (Groß, Derks, et al. 2020; Derks et al. 2021; Speak et al. 2024). By applying FlyCADD to different use cases, we demonstrated several examples of its application in prioritizing SNPs from GWAS prior to follow-up studies, design and evaluation of genome editing experiments and assessment of natural variation in the fruit fly. FlyCADD enables researchers to examine functional impact of candidate loci associated with complex traits at the SNP level across the entire *D. melanogaster* genome and in different research contexts, offering insights into the genetic basis of phenotypic variation. Given that linkage between causal and hitchhiking loci is common, both in natural and experimental populations, FlyCADD can help distinguish functionally impactful SNPs from neutral loci, especially in non-coding regions where traditional methods fall short (Smith and Haigh 1974; Smit-McBride et al. 1988). We have shown that FlyCADD scores can easily be applied as an additional layer of information on candidate SNPs and as scores are available across the entire genome, no variant or region is overlooked. Currently, FlyCADD can only be used to score functional impact of point mutations and not structural variants or other types of genomic variants. Future expansion of FlyCADD toward predicting functional impact of other types of genomic variants is possible, however. For example, the CADD framework has been applied to score functional impact of structural variants in the human genome (Kleinert and Kircher 2022).

In conclusion, we present FlyCADD, an impact prediction tool for SNPs in the genome of *D. melanogaster*. Besides the selected use cases, FlyCADD impact prediction scores can be used on a genome-wide scale in many more research contexts to study the genetic basis of phenotypes in *D. melanogaster*. Prioritization of promising SNPs or elimination of predicted low-impact SNPs based on FlyCADD impact prediction scores will help researchers reduce time, cost, and effort of follow-up studies by preventing unnecessary validation of hitchhiking loci that likely do not impact the phenotype. This targeted approach directs resources toward functionally relevant SNPs, making (experimental or computational) validation of impact more feasible.

We have made FlyCADD impact prediction scores readily available on Zenodo, providing both precomputed scores for all possible single nucleotide variants on the *D. melanogaster* reference genome and a locally executable pipeline for scoring novel variants of interest (https://doi.org/10.5281/zenodo.14887337). By extending functional SNP prediction beyond coding regions and known functional domains, FlyCADD presents a valuable addition to the existing genetic toolbox of *Drosophila*. The combination of FlyCADD with the powerful functional genetic tools for *D. melanogaster* provides unique opportunities to create a better understanding of the genotype–phenotype connections underlying phenotypic variation.

## Data availability

Supplementary File 1 describes the Cactus 166-way multi-species alignment that was used for ancestral sequence reconstruction and details on the extracted ancestral sequence, including two alternative ancestral sequences that showed excessive biases. Supplementary File 2 gives an overview of the annotations and combinations thereof used by FlyCADD. Supplementary File 3 shows the weight of all features incorporated in the trained logistic regression model, describing the contribution of each feature to the impact prediction.

All supporting data and resources related to FlyCADD are publicly available. The pipeline of FlyCADD development is available at GitHub (https://github.com/JuliaBeets/FlyCADD). Precomputed scores, a locally executable FlyCADD pipeline (including the trained logistic regression model files, annotation files, and scripts) for scoring novel variants, the multi-species alignment file, the reconstructed ancestral sequence, generated derived and simulated variants, FlyCADD scores for the included lethal point mutations and FlyCADD scores at all codon positions of unique transcripts in the *D. melanogaster* genome can be found on Zenodo (https://doi.org/10.5281/zenodo.14887337).

Supplemental material available at GENETICS online.

## Acknowledgments

The authors acknowledge BAZIS HPC cluster computing facilities at the Vrije Universiteit Amsterdam. We thank Seyan Hu for generating updated pCADD scripts that we adapted to create FlyCADD. The authors thank the *D. melanogaster* research community for their contributions in generating essential resources, annotations, and sequencing datasets that formed the basis for developing FlyCADD. We acknowledge the *Drosophila* Evolutionary Population Genomics Consortium (DrosEU) for generating and maintaining essential sequencing datasets and genomic resources, including the DEST(2.0) resource. We thank the reviewers for their comments and suggestions.

## Funding

This project was supported by the 2022 Early Career Support Grant of the Amsterdam Institute for Life and Environment (A-LIFE) at Vrije Universiteit Amsterdam awarded to MB and KMH. JH was supported by Swedish Research Council (2022-00209_VR).

Conflicts of interest. None declared.

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

*Editor: P. Wittkopp*