## [Peer Review File · Genetics]

Predicting the functional impact of single nucleotide variants in *Drosophila melanogaster* with FlyCADD

Julia Beets, Julia Höglund, Bernard Kim, Jacintha Ellers, Katja Hoedjes, and Mirte Bosse

NOTE: The reviews and decision letters are unedited and appear as submitted by the reviewers.

In extremely rare instances and as determined by a Senior Editor or the EIC, portions of a review may be redacted. If a review is signed, the reviewer has agreed to no longer remain anonymous.

The review history appears in chronological order.

Review Timeline:

Submission Date:	2025-02-27
Editorial Decision:	2025-05-09
Resubmission Received:	2025-06-20
Editorial Decision:	2025-07-22
Resubmission Received:	2025-09-15
Editorial Decision:	2025-10-02
Revision Received:	2025-10-26
Accepted:	2025-11-01

April 29, 2025

GENETICS-2025-307896

Predicting the functional impact of single nucleotide variants in *Drosophila melanogaster* with FlyCADD

Dear Dr. Beets:

Two experts in the field have reviewed your manuscript, and I have read it as well. We agree that the work is interesting and potentially suitable for publication in GENETICS. We also agree that there are many potential applications of the FlyCADD resource. While your manuscript is not currently acceptable for publication in GENETICS, we would welcome a substantially revised manuscript. Both reviewers have comments and concerns to be addressed in a revised manuscript. You can read their reviews at the end of this email.

Please pay particular attention to the following points in your revision:

- (a) Clarify the specific aspects that FlyCADD scores can indeed address key questions in genotype-phenotype interactions, discussing the limitations and the need for fine-scale genetic mapping to resolve certain complexities.
- (b) Edit to remove redundancies and streamline content to increase clarity.
- (c) Address points the reviewers identified as vague, providing specific examples and explanations.
- (d) There are a number of questions and comments about the ancestral sequence reconstruction that need to be addressed.
- (e) Consider usability of FlyCADD by others in the field; this is a key part of the future impact of this work.

We look forward to receiving your revised manuscript. Please let the editorial office know approximately how long you expect to need for revisions.

Upon resubmission, please include:

1. A clean version of your manuscript;
2. A marked version of your manuscript in which you highlight significant revisions carried out in response to the major points raised by the editor/reviewers (track changes is acceptable if preferred);
3. A detailed response to the editor's/reviewers' feedback and to the concerns listed above. Please reference line numbers in this response to aid the editor and reviewers.

Your paper will likely be sent back out for review.

Additionally, please ensure that your resubmission is formatted for GENETICS
<https://academic.oup.com/genetics/pages/general-instructions>

Follow this link to submit the revised manuscript: Link Not Available

Sincerely,

Patricia Wittkopp
Associate Editor
GENETICS

Approved by:
Stanley Fields
Senior Editor
GENETICS

Reviewer #1 :

The article submitted by Julia Beets and colleagues reports a novel resource for *Drosophila melanogaster* research. The authors employed the Combined Annotation-Dependent Depletion (CADD) framework which they trained based on 691 genomic and functional fly features using 2.7 million neutral and 2.7 million non-neutral/impactful single nucleotide polymorphisms (SNPs). The training was achieved using a logistic regression model using 90 % of the SNPs, while the model was evaluated using the remaining 10% of the variants. Based on the trained model, FlyCADD scores (0 to 1) were calculated for each position/variant in the fly genome. These pre-computed FlyCADD scores were subsequently used to assess their biological relevance. Finally, the FlyCADD scores were employed for proof-of-principle showcases including functional impact analyses for naturally varying

SNPs, refinement of GWAS results and planning and interpretation of genome editing data.

Overall, the manuscript is clearly written, and the applied methods are well-described. Especially the thorough biological validation of the FlyCADD scores provide confidence in the values. Am convinced that the FlyCADD scores are a valuable resource for different areas of *Drosophila melanogaster* research incl. evolutionary biology, comparative and functional genomics and genetic screening. I have no major criticism but would like to suggest a few revisions to improve the presentation of the data. Most of my suggestions are directly available as comments in the attached manuscript file. In the following I provide a few more general aspects:

1. The introduction outlines a few challenges in defining genotype-phenotype associations, suggesting that FlyCADD may address them. I am not sure if the problem of epistatic interactions and additive relationships are indeed tackled by FlyCADD scores. Similarly, I am not sure if the likelihood of a functional impact does indeed help to distinguish between selected loci vs. linked loci. This could only be addressed by fine-scale genetic mapping assays. The authors should make clear, which specific aspects can indeed be addressed with FlyCADD scores.
2. Generally, the text contains many redundancies which hamper the accessibility of the work. I highlight most of these occasions in the attached pdf file, but the authors may carefully re-assess the text with respect to structure to reduce redundancies.
3. Some statements and concepts are vague and would profit from examples and more detailed explanations. See attached manuscript for specific examples.
4. I had a hard time understanding the description/results of the ancestral sequence reconstruction in the main text and in File S1:
 - o I admit that I do not fully understand the assessment and rationale of choosing Node B.
 - o Am I getting it right, that only 16.4 % of the ancestral (reconstructed) genome in Node B can be aligned/identified in the Reference genome version 6?
 - o Why are the percentages of CDS covered by the ancestral genomes for the 2R and 3R arms much lower than the rest of the genome?
 - o How was the bias towards chromosomes/CDS defined? Please provide a brief description about how to interpret the presented percentages.
 - o Is the overall percentage the mean of the per chromosome percentages?
5. I feel that the FlyCADD resource would be much easier to use by many researchers if it would be implemented in an accessible web-interface. Alternatively, the authors may provide a step-by-step protocol describing how to access and use the pre-computed FlyCADD scores.

Best wishes,
Nico Posnien

Reviewer #2 :

The authors propose an integrative modeling for predicting SNP impact for *Drosophila melanogaster*'s whole genome with the exception of chromosome 4 for clearly stated methodological issues.

They use an existing framework named CADD (Combined Annotation-Dependent Depletion) that they adapted to the *Drosophila* context for "interpreting impact of any single nucleotide variant".

Although the approach is of great interest and highly promising, numerous points are slightly oversold such as the pangenomic nature of the predictions. Finally, the lack of functional validation, especially through approaches used in the recent literature strongly limits the impact of the proposed predictions.

We believe that this paper will benefit from implementing the revisions we are suggesting in this review.

The authors propose an integrative modeling for predicting SNP impact for *Drosophila melanogaster's* whole genome with the exception of chromosome 4 for clearly stated methodological issues. They use an existing framework named CADD (Combined Annotation-Dependent Depletion) that they adapted to the *Drosophila* context for "interpreting impact of any single nucleotide variant".

Major concerns:

1/ The authors rightfully acknowledge the difficulties in establishing precisely genotype-phenotype associations, particularly when the phenotypic outcome results from epistasis or additive effects between multiple mutations. Nevertheless, the authors narrow down the problem to "distinguishing causal SNPs from linked neutral loci". While this is already a highly valuable goal, it does not address the complexity of the genotype-phenotype relationship. It would be beneficial if the authors would explicitly state that.

2/ The approach builds upon the variant effect predictor (VEP) CADD (doi: 10.1038/ng.2892), which underperforms compared to other VEPs in recent benchmark studies, e.g. see doi: 10.15252/msb.202211474 or proteigym.org). Moreover CADD is a supervised method, introducing the risk of data circularity.

3/ If the "set of simulated variants is thought to contain the full spectrum of deleteriousness", then why is the "probability of the variant belonging to the simulated variants class" (Set2) considered by the authors as equivalent to the "predicted functional impact to fitness"? In other words, why would a variant likely to belong to Set2 necessarily have high functional impact? It would be more intuitive to adopt the reciprocal perspective, i.e., reason with respect to the neutral class: a variant with a high probability to belong to Set1 is likely neutral. Moreover, the fact that the proportion of predicted low-impact scores is small questions the pertinence of the design choices for Set1 and Set2. According to the literature, one would expect most of the variants to be neutral/benign or well tolerated. It seems that the design adopted by the authors, imposing strict criteria for Set1 and more relaxed ones for Set2, leads to over-predicting pathogenicity (or functional impact).

4/ The authors emphasize in the introduction that FlyCADD is not limited to variants in coding regions. Yet, protein coding associated features are among the most contributing ones (e.g. gain of stop codon (Figure 2) : there is an order of magnitude between the 1st parameter and the 3rd,, missense mutations), highlighting the functional importance of variants located in these regions. . Moreover, the interpretation of the weights is not straightforward. While it is easy to understand that premature stop codon contributes negatively, how can the authors explain that stop codon loss has a high positive score? Is this a numerical or statistical artefact? If one looks at Figure 4, variants annotated as stop codon gain or stop codon loss have both very narrow distributions toward high scores (meaning high impact). Is there an enrichment in mutations altering stop codons in the training set? Do the features include stop codons that are outside the CDS? What is the value of relCDSpos in that case? In the seminal CADD paper (doi: 10.1038/ng.2892), the authors construct a series of univariate models that contrast observed and simulated variants. This allows for exploring the training set for potential biases and it could also be useful here.

5/ The model combines almost 700 features which limits its interpretability and practical usability. What is the level of redundancy? Could the authors simplify the model and achieve the same performance? While the authors claim on multiple occasions that this complex framework provides a predictive advantage, they actually do not compare the performance of FlyCADD with other methods. How does

FlyCADD compare with approaches solely relying on evolutionary conservation, for instance? The authors could consider ProteoCast for proteins (scores readily available for the Fly proteome), GPN-MSA for alignment-based genome-wide predictions, or DNA language models (Evo2) that capture evolutionary conservation without the need for alignments? figure S2 : numerous parameters seem highly redundant (i.e. PhyloP 27-way, Phylo-P 124-way, PhyloP Cactus, ...) why not keep only the overlaps of all these methods?

6/ The description of the validation of the model is unclear. Did the author perform k-fold cross-validation on 90% of the training set and performed the blind final evaluation on the held-out 10%? If yes, what value of k did they choose? Did they average the models trained on each fold? If not, then why do they mention "cross-validation"? And why was the model still modified after this cross-validation (lines 300-305)?

7/ There is a core issue in the paper that is due to the lack of functional validation of the predictions, that leads to line 701 : "Our analyses indicate that FlyCADD has a high accuracy" this cannot be claimed as no functional validation is presented

- . lines 311 - 312, 708 - 709 : "There is no validation dataset available with known deleterious or impactful SNPs in *D. melanogaster*." ProteoCast validated their predictions using FlyBase known hypomorph and dev lethal mutations

- . lines 363 - 367 : why not showing those score distributions? In addition, why not using known *Drosophila* lethal mutants?

- . lines 465 - 467 + Figure 5 : the authors suggest that their tool can help discriminate variants within the same region and give the example of the *white* gene but do not represent known functional variants on the sequence

- . figure 8 : why choosing an example that did not bring any functional validation and not a GWAS at least proposing some known candidate (i.e. Ivanov et al., 2015)

- . lines 678 - 679 : "It is important to note that while high impact scores indicate functional impact, they do not necessarily imply a harmful or deleterious fitness effect." although claimed earlier in the paper

Minor points:

- Could the authors explain more clearly the differences between FlyCADD and the original CADD method? Why does FlyCADD have many more features and many more species in the alignment than the original CADD? Are there features used for humans that are not available for flies? In what way does/can this influence the results? The strategy of transferring a protocol designed for humans on flies should be justified in the introduction.
- "Applying FlyCADD to NS or SN point mutations can help refine interpretation of the functional impact of these variants instead of solely relying on a single protein-level consequence prediction." => the authors do not provide evidence that working at the nucleotide level would allow better refining predictions for NS mutations, compared to working at the amino acid level. line 108 - 110 : put the corresponding reference for each method next to its name
- What do the authors mean by "was largest in proportion to the reference genome (~ 16 %)"?
- "Analysis shows that the first nucleotide has significantly higher FlyCADD scores compared to the third nucleotide in 50.4 % of the genes and the second nucleotide in 62.2 % of the genes (Figure 6b)." => the description of the results is unclear. This is 1st versus 3rd and 2nd versus 3rd. The sentences "The first codon position was scored significantly higher by FlyCADD compared to the second nucleotide in 12 % of the genes. This is consistent with observations across all genes..." are also unclear. Do you mean **only** 12%?

-
- What proportion of DEST polymorphism are in FlyCADD training set? Fig. 7 is not very informative, all distributions are largely overlapping. Are the differences observed between the mean values significant? Why are the distributions shifting toward higher impact for very high allele frequencies?
 - Figure 8 misses a) and b)
 - "how to prioritization" on l. 585
 - The sentence "Since the precomputed impact prediction..." seems grammatically incorrect and thus, its meaning is unclear.
 - The clarity of Fig. 1 could be improved.
 - line 171 : logistic regressor parameters, why no comparison with the CADDs for other organisms
lines 716 - 735 : why not compare the different CADDs parameters or ability to cross predict between organisms?
 - multiple times, authors claim a genome-wide prediction but lines 200 - 207 : does this mean that they do not predict all variants then? if not, this has to be clarified; line 243 : exclusion of "rare" variants; lines 332 - 333 : predictions for codons present in only 6894 genes? -> how many are predicted with impactful mutations by other proteome-wide tools? line 479 : how were those 6894 genes chosen? lines 343 - 344 : "Only samples with label "PASS" were included in this analysis to avoid inclusion of low-quality variants." How many in total? What proportion of all predictions?
 - Figure 6 : the distribution of scores along the position within codons is interesting but the authors did not present a discussion regarding the position 3 that are highly significant (is it associated with less degenerate codons,...)
 - lines 521 - 525 + Figure 7 : the dynamic range of the FlyCADD seems to be highly restricted, with the vast majority of scores being between .25 and .45 (a global distribution would help) as such, and considering the score values given page 19, it is hard to understand how the score in itself can help discriminating between variants.
line 622 : considering the above remarks, the paragraph title is an overstatement
 - line 218 - 219 : the continuous scoring is binarised for evaluation but the cut-off chosen for it is unclear, sometimes it seems to be 0.5, sometimes 0.3 (and thus 0.7 too?)
lines 350 - 351 : such binarization of the scoring could be validated using a binary phenotype.
 - since the conclusions of the paper are that the most significant SNPs actually fall within coding sequence, why not comparing with the recent cited ProteoCast that restricts its predictions to the proteome.

Response letter

Associate Editor:

Two experts in the field have reviewed your manuscript, and I have read it as well. We agree that the work is interesting and potentially suitable for publication in GENETICS. We also agree that there are many potential applications of the FlyCADD resource. While your manuscript is not currently acceptable for publication in GENETICS, we would welcome a substantially revised manuscript. Both reviewers have comments and concerns to be addressed in a revised manuscript. You can read their reviews at the end of this email.

We thank the editor and reviewers for their thoughtful and constructive feedback on our manuscript. We appreciate the opportunity to revise our work and believe the changes we have made significantly improved the manuscript and address all the comments and concerns of the reviewers. **Please pay particular attention to the following points in your revision:**

(a) Clarify the specific aspects that FlyCADD scores can indeed address key questions in genotype-phenotype interactions, discussing the limitations and the need for fine-scale genetic mapping to resolve certain complexities.

We have now added more nuance regarding the complexity of the genotype-phenotype link and the limitations of FlyCADD in research related to this topic. FlyCADD is intended as complementary tool to association studies and functional genomics by providing an impact prediction score to understand functional variant impact. Additionally, we added an analysis of known lethal point mutations to demonstrate the correlation of predicted impact by FlyCADD with empirically validated functional effects, as suggested by reviewer #2.

(b) Edit to remove redundancies and streamline content to increase clarity.

(c) Address points the reviewers identified as vague, providing specific examples and explanations.

Based on the reviewers' comments we removed redundancy and clarified vague content. For instance, details were added to illustrate the different types of annotations in FlyCADD and the impact of CRISPR-Cas9-induced genome modifications.

(d) There are a number of questions and comments about the ancestral sequence reconstruction that need to be addressed.

We thank the editor and reviewer for pointing out the difficulties with the description of the ancestral sequence reconstruction. We compared different nodes in the 166-way phylogeny to select the node with the largest ancestral sequence size, least chromosome bias, and least coding sequence bias. This ultimately resulted in an ancestral sequence that matches 16.4 % of the reference genome, while the other reference genome positions correspond to gaps in the ancestral sequence. We have rephrased the ancestral sequence reconstruction description in File S1, as well as in the Methods section to clarify these choices.

(e) Consider usability of FlyCADD by others in the field; this is a key part of the future impact of this work.

We agree with the editor that usability of FlyCADD is a key part of its future impact. As setting up a web-interface is currently beyond our resources, we added a step-by-step protocol to improve the usability of the resources FlyCADD provides. Future developments may include a web interface, potentially in collaboration with FlyBase.

Below we respond in detail to the comments and concerns raised by the two reviewers.

Response to Reviewer #1

The article submitted by Julia Beets and colleagues reports a novel resource for *Drosophila melanogaster* research. The authors employed the Combined Annotation-Dependent Depletion (CADD) framework which they trained based on 691 genomic and functional fly features using 2.7 million neutral and 2.7 million non-neutral/impactful single nucleotide polymorphisms (SNPs). The training was achieved using a logistic regression model using 90 % of the SNPs, while the model was evaluated using the remaining 10% of the variants. Based on the trained model, FlyCADD scores (0 to 1) were calculated for each position/variant in the fly genome. These pre-computed FlyCADD scores were subsequently used to assess their biological relevance. Finally, the FlyCADD scores were employed for proof-of-principle showcases including functional impact analyses for naturally varying SNPs, refinement of GWAS results and planning and interpretation of genome editing data.

Overall, the manuscript is clearly written, and the applied methods are well-described. Especially the thorough biological validation of the FlyCADD scores provide confidence in the values. Am convinced that the FlyCADD scores are a valuable resource for different areas of *Drosophila melanogaster* research incl. evolutionary biology, comparative and functional genomics and genetic screening. I have no major criticism but would like to suggest a few revisions to improve the presentation of the data. Most of my suggestions are directly available as comments in the attached manuscript file. In the following I provide a few more general aspects:

We thank the reviewer for their thorough evaluation of our manuscript, "*Predicting the functional impact of single nucleotide variants in Drosophila melanogaster with FlyCADD*". We appreciate the level of detail in the reviewers' comments and the positive feedback on the description of FlyCADD, our methodology, and the potential value of FlyCADD for the *Drosophila* research community.

We have carefully considered all comments and suggestions, including those provided in the annotated manuscript. Below, we provide detailed responses and explanations addressing each point raised. Where appropriate, we have revised the manuscript to improve clarity, correct ambiguities, and better present the data.

Line numbers mentioned in the response represent the marked version of the manuscript and, due to formatting and additional analyses, figure numbers were changed.

1. The introduction outlines a few challenges in defining genotype-phenotype associations, suggesting that FlyCADD may address them. I am not sure if the problem of epistatic interactions and additive relationships are indeed tackled by FlyCADD scores. Similarly, I am not sure if the likelihood of a functional impact does indeed help to distinguish between selected loci vs. linked loci. This could only be addressed by fine-scale genetic mapping assays. The authors should make clear, which specific aspects can indeed be addressed with FlyCADD scores.

We thank the reviewer for this important comment. We fully agree that FlyCADD does not resolve the full complexity of genotype-phenotype relationships, including epistasis or fine-mapping of selected vs. linked loci. The impact scores are designed to be used to guide experimental validation or integration with association mapping and fine-scale analyses, not to itself resolve any of the complexities pointed out by the reviewer.

Therefore, we have clarified at lines 165 - 171 that FlyCADD is not intended to tackle the problems of epistasis and selection or capture genetic interactions but rather provides a complementary prediction tool for prioritizing variants based on their predicted fitness impact.

2. Generally, the text contains many redundancies which hamper the accessibility of the work. I highlight most of these occasions in the attached pdf file, but the authors may carefully re-assess the text with respect to structure to reduce redundancies. 3. Some statements and concepts are vague and would profit from examples and more detailed explanations. See attached manuscript for specific examples.

Comment 2 and 3 of the reviewer are both related to textual changes to improve the accessibility of the work and suggestions were incorporated in the marked PDF file by the reviewer; therefore, we reply with one statement below.

We appreciate the reviewer's suggestions and thorough highlighting of examples. Highlighted suggestions regarding grammatical or textual errors have been corrected in the text. The additional highlights of redundancy and vagueness were considered, and we removed redundant text as well as added details where statements would benefit from examples. We describe these changes we have made per section below, in order of appearance.

Introduction:

- We removed redundancy from all paragraphs where it was deemed necessary, therefore, not all sentences mentioned in the reviewers' comments are present in the revised version. We describe the relevant changes below.
- The first two paragraphs consisted of several sentences that were redundant. The revised version of the manuscript combines these paragraphs in the first paragraph. Additionally, the description of challenges complicating the interpretation of SNP effects has been rephrased at lines 77 – 79.
- We added details on line 80 - 85 to illustrate that merely statistical association of SNPs can result in true positives and false negatives, which both complicate identification of functional SNPs.
- At line 88 – 96, we clarified that functional genetics focused on genes does not assess the function(s) of SNPs. This difference between gene-level and SNP-level functional genomics is once more clarified at lines 105, 150 and 171.
- The citations for computational approaches for effect prediction are now matched with the respective tools at lines 113 - 115.
- Underutilized annotations that can be utilized to gain insights into the functional impact of SNPs include regulatory elements, amino acid changes and DNA secondary structures. We have clarified this at lines 121 - 122 and line 145.
- The paragraph on *D. melanogaster* (line 142) as model organism in the current study has been updated entirely to avoid redundancy and address the readability and missing level of detail pointed out by the reviewer.
- We thank the reviewer for pointing out the difficulties with finding the Zenodo repository using the provided DOI. The link to Zenodo has now been updated throughout the manuscript to allow readers to access the data repository.

Methods:

- We have considerably reduced the first paragraph of the Methods sections to avoid redundancy whilst keeping the aim clear (lines 195 – 197). The detailed information is now incorporated into lines 206 - 207 and lines 234 - 236.
- We have restructured the text to refer chronologically to the Supplementary Files.
- The vague description of “new, annotated variants” (line 234) has been removed. We were referring to variants that are not used for training and that are of interest to the FlyCADD users, however, the term “new” might be confusing, whilst “annotated single nucleotide variants” is sufficient.

Results:

- Figure 3: The variants included in Figure 3 are all possible variants on the reference genome. The limited number of low-impact variants depicted in Figure 3 might be related to the fact that these variants are mostly consisting of variants not, and potentially never, occurring in natural populations. These variants inherently receive a higher FlyCADD score. Therefore, all possible variants on the *D.*

melanogaster reference genome are likely to harbour little non-functional, low impact variants. We have added the explanation at lines 429 – 431, 435 – 440 and 445.

- Figure 4: We have clarified the VEP consequences in legend of Figure 4 where our category of “unknown” encompasses VEP consequences “where predictions are difficult or there is no evidence of impact” (according to the Ensembl webpage “Ensembl Variation - Calculated variant consequences”). We agree that the label “Unknown” is confusing and, therefore, we have changed the description to “Other”, which can be specified for variants of interest using the raw annotation files or Ensembl website if required by the user. Moreover, we agree with the reviewer that “UTR” and “intronic” are not necessarily “protein-level consequences” and removed this expression from the Figure description and throughout the manuscript, replacing it with “consequence class” or “VEP consequence”, for example at line 457.
- The higher predicted impact for rare alleles in the DEST2.0 dataset is not unexpected as these alleles are rare and potentially won't remain in the population and therefore resemble the derived, fixed variants less, resulting in higher FlyCADD scores.
- The description as to why relying on p-values is troublesome has been stated more explicitly, describing that confounding factors such as linked variants and environmental factors influence p-values at lines 571 - 573.
- Redundancy in the introduction of the study related to the use case of p-value ranking has been resolved and the cut-off value has been provided on lines 582 – 583.
- The SNPs identified in GWAS associated to female pigmentation include variants with very low FlyCADD scores, indicating no functional impact. We show that FlyCADD can be applied to rank the remaining, not yet functionally validated SNPs to prioritize SNPs with predicted functional impact. To clarify this, we have rephrased the sentences on lines 635 – 641 and the Figure 10 legend.
- On lines 650 – 653 and line 666, we added clarification to the paragraph on CRISPR precise genome editing regarding off-target effects and intentional genome modifications. CRISPR-Cas9 can result in three types of genome modification: (1) modification, the loci of interest modified, (2) off-target effects, unintentional modifications as a result of the Cas9 machinery targeting regions elsewhere in the genome because of partial overlap with the target region, and (3) intentional modifications, modifications to the targeting site designed to disrupt the PAM-sequence and prevent retargeting, thereby increasing genome editing efficiency. We have added explanatory text to illustrate the difference between off-target effects and intentional modifications. The intentional modifications from Perlmutter et al. 2024 were deemed silent as the protein sequence was not altered by the DNA modification. We have rephrased this at line 675.

Discussion:

- We now provide the exact number of annotations throughout the manuscript, for example at line 698.
- We removed redundancy and rephrased the conclusion under “Ensuring model reliability through training and validation” (line 726) to match the rest of the paragraph. The aim of this paragraph is to show model reliability through accuracy comparisons with other species-specific CADD models.
- Line 744: The reviewer suggests speculation about the differences between factors important for variant impact in human and in *Drosophila* genomes. However, after careful assessment of the statement, we realized that this difference cannot be inferred from the data as the authors describing humanCADD did not provide the weights (proxy for feature importance) of combinatorial features and, therefore, we have removed this sentence from the manuscript.

4. I had a hard time understanding the description/results of the ancestral sequence reconstruction in the main text and in File S1:

o I admit that I do not fully understand the assessment and rationale of choosing Node B.

We thank reviewer #1 for pointing out these unclaritys in the interpretation of the ancestral sequence reconstruction. We have added clarification to File S1 regarding the rationale of choosing the reconstructed ancestral sequence of node B. In short, we chose to assess the reconstructed ancestral sequences from nodes A, B and C because nodes deeper in the phylogeny yielded substantially smaller ancestral sequences and would

thereby limit the set of training data we could obtain. The three criteria we've used to compare nodes A, B and C relative to each other were (1) largest reconstructed ancestral sequence size, (2) least chromosome bias and (3) least coding sequence bias.

o Am I getting it right, that only 16.4 % of the ancestral (reconstructed) genome in Node B can be aligned/identified in the Reference genome version 6?

Regarding the 16,4 % coverage, this indicates the proportion of the reference genome sequence that has a reconstructed and extracted ancestral allele; therefore, it indeed reflects the percentage of successfully reconstructed genome (on coordinates relative to the Release 6 reference genome). This clarification has been added at lines 252 - 256. With 16.4 % of reconstructed ancestral sequence and the above-described assessments of bias, we were able to train a robust model as the obtained training dataset consisted of > 5 million variants, each with a corresponding ancestral allele from the reconstruction.

o Why are the percentages of CDS covered by the ancestral genomes for the 2R and 3R arms much lower than the rest of the genome?

Chromosomes 2R and 3R indeed harbor less reconstructed ancestral sequence that overlaps with CDS. During ancestral sequence reconstruction, filtering steps such as removal of ancestral anti-sense sequence mapped to sense reference sequence (line 250) might have influenced these proportions. The reviewer's observation is correct, however, we do not have a conclusive explanation for this. We do not find evidence that predictions are affected by this difference as FlyCADD is trained on the annotation patterns, rather than the specific variants themselves.

o How was the bias towards chromosomes/CDS defined? Please provide a brief description about how to interpret the presented percentages.

The bias of ancestral sequence towards chromosomes and CDS was defined by comparison of the percentages between Nodes A, B and C based on visual inspection of Figure 2 and comparison within Table 1 in File S1. We have clarified these aspects in the text of File S1 as well as in the legend of Table 1 in the Supplementary File.

o Is the overall percentage the mean of the per chromosome percentages?

The overall percentage is not the mean of the per chromosome percentages. This is because the mean of percentages overestimates the genome-wide overlap if not all chromosomes are equal in size. The overall percentage is calculated as: $(\text{total_ancestral_nucleotides_in_cds} / \text{total_ancestral_nucleotides}) * 100$, whereas the individual chromosome percentages are calculated as $(\text{ancestral nucleotides in CDS on chromosome} / \text{total ancestral nucleotides on chromosome}) * 100$. Therefore, the overall percentage reflects a genome-wide proportion weighted by sequence length and may differ from the mean of the per-chromosome percentages.

5. I feel that the FlyCADD resource would be much easier to use by many researchers if it would be implemented in an accessible web-interface. Alternatively, the authors may provide a step-by-step protocol describing how to access and use the pre-computed FlyCADD scores.

We appreciate the reviewer's suggestion and agree that an accessible interface would be beneficial for FlyCADD users to access the precomputed impact prediction scores. At present, creating a standalone web interface is not feasible due to technical and resource limitations. However, we are exploring the possibility of collaborating with FlyBase to integrate FlyCADD scores into their platform.

To facilitate immediate usability of the FlyCADD scores, we have included a step-by-step protocol for downloading and using the precomputed FlyCADD scores. This protocol will be uploaded to the main page of our Zenodo repository. The reviewer can find it as Attachment 1 within this letter.

**Best wishes,
Nico Posnien**

Response to Reviewer #2

The authors propose an integrative modeling for predicting SNP impact for *Drosophila melanogaster*'s whole genome with the exception of chromosome 4 for clearly stated methodological issues. They use an existing framework named CADD (Combined Annotation-Dependent Depletion) that they adapted to the *Drosophila* context for "interpreting impact of any single nucleotide variant". Although the approach is of great interest and highly promising, numerous points are slightly oversold such as the pangenomic nature of the predictions. Finally, the lack of functional validation, especially through approaches used in the recent literature strongly limits the impact of the proposed predictions. We believe that this paper will benefit from implementing the revisions we are suggesting in this review.

We thank the reviewer for their thorough review of our manuscript and the recognition of the promising approach. FlyCADD is trained with the aim of providing a tool that can be applied genome-wide to score any single nucleotide variant without restrictions regarding its origin. The reviewer correctly points out the methodological difficulties we encountered. For chromosome 4, we were able to generate enough training data based on the reconstructed ancestral sequence, however, this was not possible for chromosome Y. Therefore, FlyCADD scores can reliably be generated for and applied to all chromosomes except chromosome Y. In recent literature, validation of effect predictors is indeed performed using a variety of datasets. Based on the suggestions, we now have incorporated functional validation of FlyCADD scores by assessing functional impact of FlyBase-derived chemically induced point mutations with lethal phenotypic outcome, which is similar to the approach applied by the authors of ProteoCast. Altogether, the reviewers' comments and concerns were valuable to add explanation regarding the genome-wide nature and capabilities or limitations of FlyCADD, validate the impact prediction scores with point mutations of known phenotypic functional consequence and describe the training dataset more precisely.

Line numbers mentioned in the response represent the marked version of the manuscript and, due to formatting and additional analyses, figure numbers were changed.

Major concerns:

1/ The authors rightfully acknowledge the difficulties in establishing precisely genotype-phenotype associations, particularly when the phenotypic outcome results from epistasis or additive effects between multiple mutations. Nevertheless, the authors narrow down the problem to "distinguishing causal SNPs from linked neutral loci". While this is already a highly valuable goal, it does not address the complexity of the genotype-phenotype relationship. It would be beneficial if the authors would explicitly state that.

We thank the reviewer for this insightful comment, and, even though not explicitly stated in the initial submission, we do acknowledge that FlyCADD does not resolve the full complexity of the genotype-phenotype link. In response, we have revised the paragraph introducing FlyCADD (lines 165 - 171) to more explicitly state that the tool does not resolve the full complexity of genotype-phenotype associations but rather serves as a complementary approach to prioritize variants for downstream analysis.

2/ The approach builds upon the variant effect predictor (VEP) CADD (doi: 10.1038/ng.2892), which underperforms compared to other VEPs in recent benchmark studies, e.g. see doi: 10.15252/msb.202211474 or proteigym.org). Moreover CADD is a supervised method, introducing the risk of data circularity.

FlyCADD indeed builds upon the CADD frameworks established for application to the human genome and the pig genome (pCADD). The original CADD framework has been developed to predict the functional impact of human genetic variants, with a focus on pathogenic variants. The paper referred to by the reviewer is a benchmarking study showing a comprehensive comparison of many VEPs. However, many of the higher-performing VEPs are developed specifically i) to predict pathogenicity, ii) to be applied in the human genome, or iii) to make predictions predominantly for coding sequence, whereas the CADD is genome-wide VEP with the potential for species-specific application. Since our goal is to assess functional impact, genome-wide in *D. melanogaster*, rather than quantify pathogenicity, the CADD framework is chosen. It is possible that other methods have more predictive power for coding sequence variants, however, CADD is capable of ranking

variants outside of coding regions, and species-specificity increases its predictive power on these respective genomes compared to tools specifically designed focused on a single genome/species, lines 133 - 135. We now refer to the recent benchmark studies comparing traditional VEPs and CADD to address these issues at lines 129 – 133.

While CADD is indeed a supervised method, the risk of data circularity is minimal due to the nature of its training data. Unlike models trained directly on clinically labelled pathogenic and benign variants - where the same datasets are often reused for both training and evaluation - CADD is trained to distinguish between naturally derived variants and simulated variants. This framework is evolutionarily informed, rather than clinically annotated, and does not include pathogenicity labels or known functional variants in training. As a result, CADD minimizes data circularity: it does not train on the same variant categories it is later evaluated on. In Figure 4, the consequences that are used to stratify the variants are VEP consequences, these are incorporated as annotations in the FlyCADD model and, therefore, could be a minor source for circularity within the figure. In our application, we further reduce potential bias by ensuring the training dataset spans the entire genome and is not manually enriched for specific chromosomal regions or even coding regions, which would introduce biased predictions in these regions. Importantly, we do not use the training data to evaluate the FlyCADD model or benchmark predictions. The dataset was split to use 90 % for model training and 10 % for model evaluation. Thus, the supervised nature of CADD does not introduce circularity in our context. The additional validation and score evaluations show the performance of FlyCADD without indications for circularity.

3/ If the "set of simulated variants is thought to contain the full spectrum of deleteriousness", then why is the "probability of the variant belonging to the simulated variants class" (Set2) considered by the authors as equivalent to the "predicted functional impact to fitness"? In other words, why would a variant likely to belong to Set2 necessarily have high functional impact? It would be more intuitive to adopt the reciprocal perspective, i.e., reason with respect to the neutral class: a variant with a high probability to belong to Set1 is likely neutral.

We regret that the reviewer finds the description and application of set 1 and set 2 not intuitive and have changed this section to clarify the design choices for the two sets. FlyCADD is based on probabilities, not on impact scores themselves. The logistic regression model underlying FlyCADD is set up to differentiate between simulated (set 2) and derived (set 1) variants and returns two probabilities for each variant it scores: the probability of a variant belonging to set 1, and the probability of the variant belonging to set 2. Compared to set 1, set 2 is enriched in impactful variants as natural selection has not acted upon these simulated variants, therefore, we have chosen to report the probability of a variant belonging to set 2 as a proxy for functional impact. The two probabilities returned by the logistic regression model equal to 1 for each prediction, and thus, adopting the reciprocal will not change the interpreted impact of a variant. On lines 224 - 226, we have rephrased the description of set 2 and on lines 234 - 236 we've added context regarding the probabilities as computed by the logistic regression model.

Moreover, the fact that the proportion of predicted low-impact scores is small questions the pertinence of the design choices for Set1 and Set2. According to the literature, one would expect most of the variants to be neutral/benign or well tolerated. It seems that the design adopted by the authors, imposing strict criteria for Set1 and more relaxed ones for Set2, leads to over-predicting pathogenicity (or functional impact).

The reviewer correctly points out that one would expect most naturally occurring variants to be not impactful. However, it is important to note that the dataset we present in Figure 3 is not a representation of tolerated or naturally present variants but rather shows the FlyCADD scores across all possible variants (> 400 million variants). Figure 7 does reflect naturally occurring variants from DEST2.0 and indeed shows that most variants are neutral and benign as expected (~ 4.5 million variants). We have added clarification on the nature of the dataset depicted in Figure 3 at lines 435 - 440.

4/ The authors emphasize in the introduction that FlyCADD is not limited to variants in coding regions. Yet, protein coding associated features are among the most contributing ones (e.g. gain of stop codon (Figure 2) : there is an order of magnitude between the 1st parameter and the 3rd,, missense mutations), highlighting the functional importance of variants located in these regions.

While FlyCADD is not limited to variants in coding regions, the strongest predictive features indeed relate to coding consequences, such as “stop gained” or “missense” mutations. This likely reflects a combination of factors: stronger evolutionary constraints on coding variants, leading to clearer depletion signals in the observed (nearly fixed) dataset; and deeper and more consistent annotation coverage for coding regions in existing databases. Consequently, coding sequence associated features can offer greater discriminative power within the model. However, features related to coding sequence also indirectly inform the model about non-coding variants. This is because non-coding variants are typically assigned a neutral or zero value for such features, which still carries information: their non-coding nature becomes part of the predictive signal. In this way, coding-related annotations contribute to the model’s discriminative power for all variants.

. Moreover, the interpretation of the weights is not straightforward. While it is easy to understand that premature stop codon contributes negatively, how can the authors explain that stop codon loss has a high positive score? Is this a numerical or statistical artefact? If one looks at Figure 4, variants annotated as stop codon gain or stop codon loss have both very narrow distributions toward high scores (meaning high impact). Is there an enrichment in mutations altering stop codons in the training set? Do the features include stop codons that are outside the CDS? What is the value of relCDSpos in that case? In the seminal CADD paper (doi: 10.1038/ng.2892), the authors construct a series of univariate models that contrast observed and simulated variants. This allows for exploring the training set for potential biases and it could also be useful here.

Importantly, the weights shown in Figure 2 reflect the discriminative power of each feature within the logistic regression model, not the biological impact or directionality of the variant itself. For example, the high weight assigned to the “stop loss” feature does not imply a beneficial effect but rather indicates that this annotation strongly helps distinguish simulated from observed variants. The score distribution (Figure 4) shows what the model predicts; the feature weight shows how that variable helps make the prediction. Feature weights describe how informative a feature is for classification in the context of all other features and should not be interpreted as stand-alone indicators of biological effect. To clarify this, we have revised the main text (lines 404 – 409 and 418 - 426) to better explain how feature weights reflect discriminative power, not impact direction or magnitude. For the same reason, we have also updated Figure 2 to more accurately reflect the predictive power of each feature in the model. We emphasize that the direction of the weights reflect the way our annotations were encoded and do not imply biological meaning at lines 331 - 334.

Regarding stop codons outside of coding region, stop codon features are not present outside coding sequences, and therefore our features do not include such stop codons. The Ensembl VEP is used to annotate these features, and their severity order was applied in case variants could be annotated with multiple consequences. The feature “relCDSpos” indicates the position within coding sequence for those variants that are in coding sequence. This feature remains empty for those variants outside of coding regions (the value is 0 in the encoded annotation output). However, this value is also used by the model to discriminate variants.

Enrichment of coding-related features (such as stop codon gain or missense mutations) in the top contributors reflects the biological relevance of these sites and is not an artifact. The relevance can be interpreted as a depletion of loci in the observed set of (nearly) fixed derived variants, likely because of purifying selection against such mutations on a longer evolutionary timescale. The simulated training set intentionally contains high-impact simulated variants, not to balance the classes, but to maximize discriminative signal. Note, however, that the frequency of the high-impact variants in the set of simulated variants is determined by mutation rate and is determined randomly by the simulator. Therefore, we did not artificially enrich this set with high-impact variants but simply rely on the proportions that arise under a neutral scenario, without (purifying or balancing) selection. As indicated on line 286 – 288, we have added a figure to File S3 reflecting

the proportions of variants with different VEP-annotated consequences in the derived and simulated training datasets.

5/ The model combines almost 700 features which limits its interpretability and practical usability. What is the level of redundancy? Could the authors simplify the model and achieve the same performance? While the authors claim on multiple occasions that this complex framework provides a predictive advantage, they actually do not compare the performance of FlyCADD with other methods. How does FlyCADD compare with approaches solely relying on evolutionary conservation, for instance? The authors could consider ProteoCast for proteins (scores readily available for the Fly proteome), GPN-MSA for alignment-based genome-wide predictions, or DNA language models (Evo2) that capture evolutionary conservation without the need for alignments?

The reviewer rightfully points out the large number of annotations incorporated in FlyCADD. When considering redundancy, both the individual annotations and combinations thereof should be considered, and we do think that the level of redundancy is not major. Not all included annotations provide high predictive power when included as individual feature, however, many combined show high predictive power in the model (File S3). For example, PhyloP124 has relatively low predictive power (-0.073), whereas its power is increased when combined with several VEP consequences such as conservation scores in intronic regions (-0.187). Additionally, the concern regarding redundancy has been systematically assessed in the publication describing mouseCADD 10.1186/s12859-018-2337-5, they have shown that the models' performance drops when removing features, especially removal of Ensembl VEP and conservation features. A CADD model where the number of features is reduced might suffice, however, meaningful annotations to improve the predictions in specific regions are present such as secondary structure predictions or conservation scores, and they show discriminative power through combinatorial features as well.

The basis for creating a CADD model is the benefit of combining all available types of annotations into one predictive score, where the features and combinations are weighted based on their performance. The reviewer's comment is valid, in the sense that some hypotheses are more robustly assessed with specific annotation(s) than others. The last years, it has been common to assess ultra-rare highly deleterious variants in human genetic studies by manually assessing several types of annotations and keeping only those with overlaps. For targeted studies, this might very well be the best approach. However, in a more exploratory study, or a hypothesis free such as prioritizing associated SNPs of a GWAS, combining all assessments into one framework, can be much more suitable. As the reviewer states, however, there is no true answer to the correct amount of input.

figure S2 : numerous parameters seem highly redundant (i.e. PhyloP 27-way, Phylo-P 124-way, PhyloP Cactus, ...) why not keep only the overlaps of all these methods?

We chose to include all three PhyloP features to reflect the depth of nucleotide conservation. Assessing evolutionary constraint (nucleotide conservation) can be done on different evolutionary timescales. When using a large phylogenetic tree (in PhyloP166), i.e., a longer evolutionary time, one might only capture overall highly conserved regions, whereas with a smaller tree (in PhyloP24) and thus shorter evolutionary time, one might capture branch-specific conservation. As these three conservation score datasets were already available for *Drosophila*, representing different phylogenetic scopes, we decided to include them all, in the hope of capturing as broad of a conservation annotation as possible.

6/ The description of the validation of the model is unclear. Did the author perform k-fold cross-validation on 90% of the training set and performed the blind final evaluation on the held-out 10%? If yes, what value of k did they choose? Did they average the models trained on each fold? If not, then why do they mention "cross-validation"? And why was the model still modified after this cross-validation (lines 300-305)?

The reviewer rightfully points out the unclear model validation. We have realized that the term cross-validation was not appropriate in this context. We applied sub-sampling validation to determine the optimal L2 penalty value, followed by training of the final model. Specifically, we split the dataset (90% training, 10% testing) five times for each value of L2 (0.01, 0.1, 1.0, 10.0, 100.0) and selected the model with the highest average accuracy. This process was intended solely for hyperparameter tuning. After identifying the optimal L2 penalty,

we trained a final model using the optimal parameters on the training set and evaluated it on the held out 10% test set. We did not modify the model after training. We have clarified this workflow in lines 322 - 326.

7/ There is a core issue in the paper that is due to the lack of functional validation of the predictions, that leads to line 701 : “Our analyses indicate that FlyCADD has a high accuracy” this cannot be claimed as no functional validation is presented

We thank the reviewer for their comment. Below we addressed the raised concerns with which we support the claim regarding FlyCADD score accuracy. The accuracy of the model (using AUC) and the accuracy of the FlyCADD scores (using validation datasets and other analyses) are now both reported, as indicated at line 732 - 738.

. lines 311 - 312, 708 - 709 : “There is no validation dataset available with known deleterious or impactful SNPs in *D. melanogaster*.” ProteoCast validated their predictions using FlyBase known hypomorph and dev lethal mutations

In other species, CADD performance was evaluated using validation datasets of variants with known impact, mostly clinically relevant, naturally occurring pathogenic variants spanning the full range of pathogenicity. Such a database is not available for *D. melanogaster*. However, as the reviewer points out, there are point mutations with known phenotypic outcome documented in FlyBase.

Based on the reviewer’s suggestion, we added validation of the FlyCADD scores by scoring the FlyBase lethal mutations induced using ethyl methanesulfonate (EMS). 2202 point mutations with the phenotypic outcome of lethality were identified using the QueryBuilder of FlyBase (see Figure below). We predicted impact of the unique variants with known position and base change (n = 2118) within this set using FlyCADD, which resulted in > 88% with predicted high functional impact (FlyCADD score > 0.6). These results are consistent with the measured phenotype of lethality and further strengthen FlyCADD performance by validation using point mutations known phenotypic outcome.

Sections were added to the Methods (lines 361 - 367) and Results (lines 507 - 526) section, together with the addition of Figure 7. Accordingly, we updated the sections referring to the lack of functional validation to refer to this form of validation, or removed statements related to the lack of a validation dataset, such as Results line 397. At line 182, we introduce the validation and at lines 736 - 738, we discussed the validation set.

. lines 363 - 367 : why not showing those score distributions? In addition, why not using known *Drosophila* lethal mutants?

The score distributions of model testing are not available via the TuriCreate logistic regressor. However, we added score distributions of lethal mutations as elaborated on in the comment above. The validation of FlyCADD now consists of model validation through the built-in accuracy metrics of TuriCreate and validation of the impact prediction score validation through the assessment of lethal point mutations from FlyBase.

. lines 465 - 467 + Figure 5 : the authors suggest that their tool can help discriminate variants within the same region and give the example of the *white* gene but do not represent known functional variants on the sequence

As stated above, very limited numbers of SNPs are functionally validated in *D. melanogaster*. Figure 5 is included in the manuscript to illustrate the patterns within genes, such as intronic and exonic regions. Based on a manual annotation one would assume exonic variants are more impactful than intronic variants or even intergenic variants. FlyCADD can be applied to discriminate variants within these regions, as can be seen by variants spanning the entire range of FlyCADD scores within exons and within introns. However, we do agree with the reviewer that the depiction of only theoretically present variants in the gene *white*, namely all possible variants across this region on the X chromosome, becomes more informative when known variants are presented. Functional variants that are experimentally validated within this region are rare. Therefore, we added naturally occurring variants of the DEST2.0 resource to the figure. The figure now includes all possible

variants in purple, and highlighted with blue diamonds are all variants naturally occurring in worldwide *D. melanogaster* populations, as indicated in the figure legend and at line 473.

. figure 8 : why choosing an example that did not bring any functional validation and not a GWAS at least proposing some known candidate (i.e. Ivanov et al., 2015)

The example is included in our manuscript to illustrate the application of FlyCADD prior to functional validation. Currently, prioritization of SNPs after GWAS identification is usually based on genomic location - which we showed with Figure 5 to give limited information- or on p-values, as in the chosen example study. With this figure, we illustrate how FlyCADD scores can be used to refine prioritization of candidate SNPs derived from GWAS studies prior to functional testing. The figure shows that even variants with a high p-value can have a low predicted impact and, more importantly, the opposite is true as well. The GWAS study brought up by the reviewer (doi: 10.1093/gerona/glv047) only includes candidate genes and does not provide functional validation. With Figure 10, we illustrate the application of FlyCADD on a GWAS where functional validation was present and variants with known phenotypic outcomes were scored.

. lines 678 - 679 : "It is important to note that while high impact scores indicate functional impact, they do not necessarily imply a harmful or deleterious fitness effect." although claimed earlier in the paper

The high impact score reflects functional impact rather than deleteriousness. This reflects the model training where the model was trained to distinguish derived from simulated variants instead of known deleterious from known benign variants.

Minor points:

- Could the authors explain more clearly the differences between FlyCADD and the original CADD method? Why does FlyCADD have many more features and many more species in the alignment than the original CADD? Are there features used for humans that are not available for flies? In what way does/can this influence the results? The strategy of transferring a protocol designed for humans on flies should be justified in the introduction.

FlyCADD has 691 features which include combinatorial annotations, however, the number of individual annotations is 38. Original CADD combines 63 annotations, chickenCADD 40 and pCADD 35 distinct annotations. As the reviewer indicates, the feature types differ between the CADDs. For example, annotations in the human CADD SIFT, methylation and several population-based allele frequencies are not available for *D. melanogaster*. On the other hand, FlyCADD incorporates combinatorial features including regulatory domain mapping, unique for FlyCADD. These differences underline once more why species-specific CADD models are currently necessary as available annotations and genomes of human, pig, chicken, mouse and fruit fly are largely different. Line 138 and 142 - 143 explain that it has been applied species-specific before, however, never for insects and highlights the importance of reliable genomic resources to build each CADD model on.

- "Applying FlyCADD to NS or SN point mutations can help refine interpretation of the functional impact of these variants instead of solely relying on a single protein-level consequence prediction." => the authors do not provide evidence that working at the nucleotide level would allow better refining predictions for NS mutations, compared to working at the amino acid level.

We show that FlyCADD provides finetuning of impact predictions even for variants within classes of VEP consequences. Instead of suggesting that we compare between nucleotide level and amino acid level, we made it more clear now that we are comparing nucleotide level FlyCADD predictions to the broader VEP consequences/classes such as "intergenic" or "synonymous" at lines 457 - 458.

line 108 - 110 : put the corresponding reference for each method next to its name

This suggestion has been implemented at lines 113 - 115.

- What do the authors mean by "was largest in proportion to the reference genome (~ 16 %)"?

We have rephrased this sentence to clarify that we refer to the proportion of the reference sequence covered by the reconstructed ancestral sequence, which explains that 16.4 % of the reference nucleotides have a corresponding reconstructed ancestral nucleotide. The revised phrasing appears in lines 252 - 256 of the manuscript.

- "Analysis shows that the first nucleotide has significantly higher FlyCADD scores compared to the third nucleotide in 50.4 % of the genes and the second nucleotide in 62.2 % of the genes (Figure 6b)." => the

description of the results is unclear. This is 1st versus 3rd and 2nd versus 3rd. The sentences "The first codon position was scored significantly higher by FlyCADD compared to the second nucleotide in 12 % of the genes. This is consistent with observations across all genes..." are also unclear. Do you mean only 12%?

We thank the reviewer for pointing out these sentences. Two of the percentages were at the wrong position in the text, resulting in unclarity in the description and interpretation of these results. At lines 497 - 507, we rephrased the entire description of the results to correctly address the results. The third nucleotide is least functionally impactful, followed by the first nucleotide. The second nucleotide was most impactful in most of the genes.

- What proportion of DEST polymorphism are in FlyCADD training set? Fig. 7 is not very informative, all distributions are largely overlapping. Are the differences observed between the mean values significant? Why are the distributions shifting toward higher impact for very high allele frequencies?

The number of variants shared between the training set and DEST2.0 resource is 159,599, which is 2.9% of the FlyCADD training data, or 3.3% of the DEST polymorphisms. Most of these variants overlap with the derived variants (set 1), namely 122,034, which reflects the setup of the training sets where derived variants should depict (nearly-)fixed variants.

The differences between mean values are significant for the rare variants only ($AF < 0.05$). This is not unexpected as rare variants can be impactful and under selection through this.

We speculate that the shift towards higher FlyCADD scores for (nearly-) fixed loci indicates potentially harmful mutations that either became fixed and therefore can no longer be purged or represent *D. melanogaster*-specific variants that are indeed impactful, but not harmful; instead, they are selected for and became fixed.

- Figure 8 misses a) and b)

This has been resolved in the revised version of the figure.

- "how to prioritization" on l. 585

This typing error has been resolved on line 606.

- The sentence "Since the precomputed impact prediction..." seems grammatically incorrect and thus, its meaning is unclear.

We corrected the grammar on this sentence to reflect the meaning of the sentence.

- The clarity of Fig. 1 could be improved.

We have made changes to Figure 1 to improve clarity: removed details that can be found in the Methods section following the figure, increased font size, and added directionality and section titles. The legend was updated (lines 1072 – 1079)

- line 171 : logistic regressor parameters, why no comparison with the CADDs for other organisms

The logistic regressor used to develop FlyCADD differs with those of the other CADDs. The currently used TuriCreate logistic regression model was not available for development of the CADDs prior to FlyCADD. Additionally, the features included differ largely between the CADDs. Therefore, a proper comparison between the models is difficult. Due to the species-specific development of each CADD, the comparison won't be informative for the purpose of methodological comparison.

lines 716 - 735 : why not compare the different CADDs parameters or ability to cross predict between organisms?

The CADD cross-prediction between species has been extensively tested by applying the mouseCADD to humanCADD and opposite in doi 10.1186/s12859-018-2337-5 (mouseCADD) and has shown that species-specific CADD models show increased performance compared to cross prediction. Therefore, we believe that CADD models that were specifically trained to score variants of a certain species should not be used to predict impact of variants on another organism, except potentially when annotations and genomes are unavailable to set up CADD, for example, for non-model organisms or at closely related species. Assessment of the performance of FlyCADD on a closely related species, such as *D. sukukii*, is out of the scope of this paper as we focus on application specifically to *D. melanogaster* but is a very relevant topic for future research. This was emphasized at lines 741 – 753.

When species-specific CADD scores are not available, using the closest one available has been shown to be informative such as using chicken CADD in pigeons. However, as briefly mentioned above, cross prediction might not be robust enough with non-model organisms or more distantly related species (both functionally

and evolutionarily), especially if validation datasets or high-resolution phenotypic data are limited or absent. Therefore, we applied CADD to an insect genome for the first time as mammalian CADD models might not cross predict accurately. FlyCADD has been optimized for the insect *D. melanogaster* genome (lines 753 – 756).

- multiple times, authors claim a genome-wide prediction but lines 200 - 207 : does this mean that they do not predict all variants then? if not, this has to be clarified; line 243 : exclusion of “rare” variants;

Genome-wide prediction refers to the application and capability of the trained FlyCADD model to predict the functional impact of any genomic variant, irrespective of its location. The lines mentioned by the reviewer refer to model training where, indeed, not all possible variants were included. Model training does not include all genomic variants, however, by ensuring the least bias towards coding sequence or specific chromosomes in ancestral sequence reconstruction and incorporating > 5 million variants, we developed FlyCADD with the goal of genome-wide application. The exclusion of rare variants from training data ensures that the CADD framework is set up to distinguish nearly-fixed derived variants from simulated variants by the depletion of the derived variant set from rare, potentially impactful variants. This does not change the genome-wide applicability of FlyCADD.

lines 332 - 333 : predictions for codons present in only 6894 genes? -> how many are predicted with impactful mutations by other proteome-wide tools? line 479 : how were those 6894 genes chosen?

Regarding the codon position analysis, the set of 6894 genes was obtained by filtering Ensembl v111 annotations to include only unique transcripts on the sense strand. The genes were not selected based on presence or absence of impactful mutations. This is described at lines 353 - 360. To reduce ambiguity in codon position analysis and to avoid confounding interaction effects, we excluded genes with multiple transcripts and retained only one transcript per gene. This resulted in 6894 genes with clear codon structures. This analysis is intended to show the performance of FlyCADD on known biologically differing nucleotides, namely the less impactful third codon. The presence of impactful mutations is not relevant and is shown in other sections of the manuscript, such as with Figure 4.

lines 343 - 344 : “Only samples with label “PASS” were included in this analysis to avoid inclusion of low-quality variants.” How many in total? What proportion of all predictions?

The DEST2.0 samples that were included consisted of the variants amongst 529 populations with the label “PASS”. This is based on the quality filtering recommendations described by Nunez et al. (2024). This does not affect the number of predictions we have computed with FlyCADD as the number of SNPs in the genomic resource remained the same ($n = 4.801.077$). We included the number of populations at line 374.

- Figure 6 : the distribution of scores along the position within codons is interesting but the authors did not present a discussion regarding the position 3 that are highly significant (is it associated with less degenerate codons,...)

The impact of variation of the third nucleotide within a codon was the main reason for the inclusion of this analysis in the manuscript. The third codon is often referred to as “wobble” position as base pairing is less strict and amino acid changes are less likely to occur based on changes to the third nucleotide, and we therefore expected the lowest FlyCADD scores for these variants. In the analysis, we aimed to show whether FlyCADD scores reflect these biologically expected patterns. Therefore, we now have explicitly discussed the impact of variation of the third nucleotide at line 491 – 493.

- lines 521 - 525 + Figure 7 : the dynamic range of the FlyCADD seems to be highly restricted, with the vast majority of scores being between .25 and .45 (a global distribution would help) as such, and considering the score values given page 19, it is hard to understand how the score in itself can help discriminating between variants. line 622 : considering the above remarks, the paragraph title is an overstatement

We appreciate the reviewers’ suggestion to include a global distribution of FlyCADD scores of naturally occurring SNPs and included this as panel a) in Figure 8.

33 % of all naturally occurring variants have a score above 0.45, and 25 % have a score below 0.2. This means that many variants are outside the range proposed by the reviewer. Additionally, even within this range the scores are meaningful and can be applied to discriminate between variants. FlyCADD can highlight outliers with high or low predicted functional impact, and discriminate variants across the full range as the full range of scores is present in natural populations. The figure includes all possible natural variants, whilst potential users of FlyCADD will have a list of candidate SNPs based on genomic location, GWAS studies or other initial

investigations. As we show with, for example, Figure 10, FlyCADD can be applied to rank naturally occurring variants of interest to remove or highlight outliers prior to follow-up studies.

- line 218 - 219 : the continuous scoring is binarised for evaluation but the cut-off chosen for it is unclear, sometimes it seems to be 0.5, sometimes 0.3 (and thus 0.7 too?) lines 350 - 351 : such binarization of the scoring could be validated using a binary phenotype.

The FlyCADD score is indeed a continuous metric, and binary classification was used for illustrative purposes in some evaluations. In other analyses (e.g., Figure 9), the full continuous score was considered without a threshold. When a threshold was needed, we used either 0.3 or 0.5 depending on the context. The interpretation of high and low FlyCADD scores is dependent on the application, however, scores below 0.3 have low impact. Additionally, we mention naturally tolerated scores with a score < 0.4 based on the score distributions in Figure 8, with > 0.8 scores being high impact outliers, which is based on the score distribution of naturally occurring variants. Scores above 0.6 indicate biologically meaningful impact. We recognize the reviewers' comment on the description of multiple thresholds and have standardized these descriptions throughout the manuscript, for example at lines 594 and 688.

FlyCADD scores should be applied as continuous scoring to rank and prioritize SNPs based on predicted functional impact. We recommend not to binarize these scores during application, however, for application a threshold is required. Regarding validation with binary phenotypes (line 380), we agree and note that such an approach is included in our analysis using lethal alleles (see Figure 7), which demonstrates that FlyCADD scores are predictive for variants of phenotypic severity. This validation has been described in response to the reviewers' comment 7.

- since the conclusions of the paper are that the most significant SNPs actually fall within coding sequence, why not comparing with the recent cited ProteoCast that restricts its predictions to the proteome.

We have demonstrated the use of FlyCADD across the entirety of the *D. melanogaster* genome and do not intend to conclude that the most significant SNPs fall in coding sequence from the presented analyses. However, we do acknowledge that the most predictive features include features restricted to coding region. With the inclusion of combinatorial features, we ensured predictive power present irrespective of the genomic position of variants. The comparison with ProteoCast presents itself with several difficulties regarding the aim of the predictive tool, the range of applications and the continuous or binarized score approach. The comparison of FlyCADD with other VEPs is outside of the scope of this manuscript. At lines 115 – 119, we describe these different properties of VEPs. However, with the inclusion of FlyBase lethal point mutations, we have validated our FlyCADD scores in a similar manner as ProteoCast, with similar outcomes. We have acknowledged this in the discussion at lines 737 - 742.

Attachment 1: Protocol FlyCADD score usage

File: *protocol_precomputedFlyCADDscores.txt*

Author: Julia Beets, contact: j.beets@vu.nl

Date: 19-05-2025

Protocol: Accessing and Using FlyCADD Scores

Overview:

FlyCADD is adapted from the original CADD framework applied to the human genome (Kircher et al., 2014) and provides genome-wide precomputed functional impact prediction scores for all possible single nucleotide variants (SNVs) in the *D. melanogaster* genome (BDGP6.32). These precomputed impact prediction scores are stored in compressed, tabular format and can be queried using standard bioinformatics tools.

1. Accessing FlyCADD Scores

Location:

FlyCADD scores are available via Zenodo <https://doi.org/10.5281/zenodo.14887337> in the compressed folder *precomputed_FlyCADDscores.zip*.

This directory (*precomputed_FlyCADDscores/*) contains readily available FlyCADD scores for all possible genomic variants on the *D. melanogaster* reference genome Release 6 in comma-separated files (.csv).

2. File description

A file is present per chromosome for chr2L, chr2R, chr3L, chr3R, chr4 and chrX, containing the following columns:

- chromosomal position (1-based positional information)
- reference allele (allele in reference genome Release 6)
- alternative allele (variant allele)
- FlyCADD score (0 - 1)

Download the zip folder and extract the contents. This allows you to access the file containing the FlyCADD scores at all positions of your respective chromosome of interest.

3. Querying FlyCADD Scores

The FlyCADD scores can be retrieved from the comma-separated files (.csv) in several ways. The columns allow for searching or matching your variant(s) of interest based on the position, reference allele and alternative allele. A score can be looked up for a (few) variant(s) using the command line, or FlyCADD scores for a positional range can be saved in an output file for manual inspection or downstream analysis, for example of positions in a specific gene. Alternatively, a .vcf file containing the variants of interest on a particular chromosome can be matched with the FlyCADD scores using a Python script (example below). Any alternatives can be used as the scores are provided as .csv format files. These three types of lookup for FlyCADD scores are illustrated below.

A) Searching a specific variant based on the variants location (specified in column 1) and saving the result to "output.csv". Example: position 1000 on chromosome 3L:

On Linux or Mac -> `$ awk -F',' '$1 == "1000"' path/to/3L_merged_chunks.csv > output.csv`

On Windows Powershell -> `$ Get-Content path\to\3L_merged_chunks.csv | Where-Object { ($_.split(',')[0] -eq '1000') } | Set-Content output.csv`

B) Users may want to extract a specific genomic region from the FlyCADD output files for manual inspection or downstream analysis. This can be done efficiently using command-line tools. Below are examples for both Unix-like systems and Windows PowerShell.

1st Revision - Authors' Response to Reviewers: June 20, 2025

Example: Extract SNPs from position 1000 to 2000 on chromosome 3L

On Linux or macOS (bash shell) -> `$ awk -F',' '$1 >= 1000 && $1 <= 2000' path/to/3L_merged_chunks.csv > chr3L_1000_2000.csv`

On Windows PowerShell -> `$ Get-Content path\to\3L_merged_chunks.csv | Where-Object {$fields = $_ -split ','; [int]$fields[0] -ge 1000 -and [int]$fields[0] -le 2000} | Set-Content -Path chr3L_1000_2000.csv`

These save the results to chr3L_1000_2000.csv for inspection in Excel or alternative downstream analysis. This approach allows users to extract any positional range of interest from the FlyCADD output files for further inspection, filtering, or visualization.

C) To extract scores for specific variants, per chromosome, use the Python script at the end of this guide. Set the score_file_path to the location of the file containing FlyCADD scores for your chromosome of interest, and input_vcf to the path of the .vcf/.tsv file containing your variants of interest on the specific chromosome.

4. Interpreting FlyCADD scores

FlyCADD scores range from 0 to 1, with higher scores indicating a higher likelihood of functional impact. As a general guide:

<0.3: likely neutral, not impactful

0.3-0.6: moderately impactful

>0.6: impactful

Note: These thresholds are empirical and context-dependent. The scores are intended to use in the prioritization of genomic variants on the *D. melanogaster* genome, not to interpret the biological functionality of a single variant or to binarize scores. For more information on the interpretation and score ranges, see <https://doi.org/10.1101/2025.02.27.640642>.

5. Citing FlyCADD

If you use FlyCADD in your research, please cite:

Beets J, Høglund J, Kim BY, Ellers J, Hoedjes KM, Bosse M. Predicting the functional impact of single nucleotide variants in *Drosophila melanogaster* with FlyCADD. bioRxiv. 2025:2025.02.27.640642.

Python script for score extraction

Example input .vcf file with variants of interest. Header lines indicated with “#” are optional.

##fileformat=VCFv4.2

#CHROM	POS	ID	REF	ALT
4	4235	.	C	T
4	25345	.	C	T
4	2453	.	G	A
4	2345	.	C	T
4	65456	.	G	T
4	5656	.	G	A

Example input .csv file with FlyCADD scores for 4_merged_chunks.csv

1,T,A,0.5797272683956932

1,T,C,0.60606710574278

1,T,G,0.6247145667403343

2,T,A,0.35236151888024436

2,T,C,0.3776537792131929

1st Revision - Authors' Response to Reviewers: June 20, 2025

2,T,G,0.3963442592934381

```
import os
import csv
import pandas as pd

# Filepath to FlyCADD scores of the chromosome of interest
score_file_path = '/path/to/2L_merged_chunks.csv'

# Load the scores for the chromosome of interest
impact_scores_df = pd.read_csv(score_file_path, header=None, names=['Pos', 'Ref', 'Alt', 'Score'])
impact_scores_df['Pos'] = impact_scores_df['Pos'].astype(int)
print("FlyCADD scores loaded.")

# Path to VCF file containing your variants of interest on the chromosome
input_vcf = 'my_2L_variants.tsv'

# Set output filename
output_file = '2L_extracted_FlyCADDscores.csv'

# Find and save the scores of the positions in your VCF file
with open(input_vcf, 'r') as vcf_file, open(output_file, 'w', newline='') as csvfile:
    csvwriter = csv.writer(csvfile)
    for line in vcf_file:
        line = line.strip()
        if line.startswith('#'):
            continue

        fields = line.split('\t')
        chrom, pos, ref, alt = fields[0], int(fields[1]), fields[3], fields[4]

        impact_score = "NOT FOUND"
        mask = (impact_scores_df['Pos'] == pos) & \
              (impact_scores_df['Ref'] == ref) & \
              (impact_scores_df['Alt'] == alt)
        matching_scores = impact_scores_df[mask]
        if not matching_scores.empty:
            impact_score = matching_scores.iloc[0]['Score']
        csvwriter.writerow(fields[:5] + [impact_score])
    print(f"Output written to {output_file}")
```

Comments for the Authors

We would like to thank the authors for having taken into consideration most of our suggestions. The additional validation experiments they performed clearly emphasize limitations of the method that should be discussed in the main manuscript text. While we continue to think that the tool represents a deep interest to the Drosophila community, multiple points still require attention. We present an overview here for clarity, followed by a full answer inline with the authors' responses.

- 1) The functional validation on a set of known lethal mutations only provides a recall estimate. It is impossible to compute or assess predictive accuracy in the absence of a negative set (i.e. non-lethal mutations). Moreover, the authors did not provide the list of lethal mutations they used, which compromises reproducibility.
- 2) A certain number of vague points still remain to be addressed, the over inflation of descriptive features compared to previous CADDs as well as their redundancy, the data circularity and data leakage issues still require a thorough description of the training/test sets, the term accuracy is used multiple times out of context, no accuracy is presented (see point 1).
- 3) The ability of FlyCADD to make biologically or clinically relevant predictions outside of CDS remains unclear
- 4) Comparison with other existing methods, which is essential for assessing the pertinence and relevance of the proposed novel method, is still missing from the manuscript. The recent literature about variant effect predictors is extremely active and abundant, with community-wide efforts for benchmarking novel methods. Since the central and main contribution of the manuscript is a novel method for predicting variant outcome, the authors should more carefully acknowledge relevant literature and explain how their proposed methodology compares to others (including those that use much less features or rely on unsupervised language models).

- 1) FlyCADD scores can indeed address key questions in genotype-phenotype interactions

Associate Editor:

(a) Clarify the specific aspects that FlyCADD scores can indeed address key questions in genotype-phenotype interactions, discussing the limitations and the need for fine-scale genetic mapping to resolve certain complexities.

We have now added more nuance regarding the complexity of the genotype-phenotype link and the limitations of FlyCADD in research related to this topic. FlyCADD is intended as complementary tool to association studies and functional genomics by providing an impact prediction score to understand functional variant impact. Additionally, we added an analysis of known lethal point mutations to demonstrate the correlation of predicted impact by FlyCADD with empirically validated functional effects, as suggested by reviewer #2.

Authors should clearly state the methods, papers and analyses that inspired their validation in the main body of the preprint. In addition, no information is given regarding the specificity of those predictions. They should calculate an accuracy of prediction using the distribution of scores for non-lethal mutations. In addition, no table with the used mutations and their predicted scores is presented. How can one reproduce this analysis?

- 2) Address points the reviewers identified as vague
 - a) over inflation of categories compared to other CADDs (x10)
 - b) data circularity / data leakage

2/ The approach builds upon the variant effect predictor (VEP) CADD (doi: 10.1038/ng.2892), which underperforms compared to other VEPs in recent benchmark studies, e.g. see doi: 10.15252/msb.202211474 or proteigym.org). Moreover CADD is a supervised method, introducing the risk of data circularity.

FlyCADD indeed builds upon the CADD frameworks established for application to the human genome and the pig genome (pCADD). The original CADD framework has been developed to predict the functional impact of human genetic variants, with a focus on pathogenic variants. The paper referred to by the reviewer is a benchmarking study showing a comprehensive comparison of many VEPs. However, many of the higher performing VEPs are developed specifically i) to predict pathogenicity, ii) to be applied in the human genome, or iii) to make predictions predominantly for coding sequence, whereas the CADD is genome-wide VEP with the potential for species-specific application. Since our goal is to assess functional impact, genome-wide in *D. melanogaster*, rather than quantify pathogenicity, the CADD framework is chosen. It is possible that other methods have more predictive power for coding sequence variants, however, CADD is capable of ranking variants outside of coding regions, and species-specificity increases its predictive power on these respective genomes compared to tools specifically designed focused on a single genome/species, lines 133 - 135. We now refer to the recent benchmark studies comparing traditional VEPs and CADD to address these issues at lines 129 – 133.

While CADD is indeed a supervised method, the risk of data circularity is minimal due to the nature of its training data. Unlike models trained directly on clinically labelled pathogenic and benign variants - where the same datasets are often reused for both training and evaluation - CADD is trained to distinguish between naturally derived variants and simulated variants. This framework is evolutionarily informed, rather than clinically annotated, and does not include pathogenicity labels or known functional variants in training. As a result, CADD minimizes data circularity: **it does not train on the same variant categories it is later evaluated on.**

In Figure 4, the consequences that are used to stratify the variants are VEP consequences, these are incorporated as annotations in the FlyCADD model and, therefore, could be a minor source for circularity within the figure. In our application, we further reduce potential bias by ensuring the training dataset spans the entire genome and is not manually enriched for specific chromosomal regions or even coding regions, which would introduce biased predictions in these regions. Importantly, we do not use the training data to evaluate the FlyCADD model or benchmark predictions. The dataset was split to use 90 % for model training and 10 % for model evaluation. Thus, the supervised nature of CADD does not introduce circularity in our context. The additional validation and score evaluations show the performance of FlyCADD without indications for circularity.

As far as we understood the manuscript, this statement is wrong. The only AUC and accuracy reported in the paper were computed on the 10% test data held out during training; this evaluation is performed on exactly the same categories it was trained on.

Moreover, it is not clear whether the 46 DEST1 used for training are included in the 529 DEST2 reported during evaluation: it seems to be the case, which would imply data leakage between train and test: the variants used in set 1 of training again appear in the DEST2 evaluation.

6/ The description of the validation of the model is unclear. Did the author perform k-fold cross-validation on 90% of the training set and performed the blind final evaluation on the held-out 10%? If yes, what value of k did they choose? Did they average the models trained on each fold? If not, then why do they mention “cross-validation”? And why was the model still modified after this cross-validation (lines 300-305)?

The reviewer rightfully points out the unclear model validation. We have realized that the term cross-validation was not appropriate in this context. We applied sub-sampling validation to determine the optimal L2 penalty value, followed by training of the final model. Specifically, we split the dataset (90% training, 10% testing) five times for each value of L2 (0.01, 0.1, 1.0, 10.0, 100.0) and selected the model with

the highest average accuracy. This process was intended solely for hyperparameter tuning. After identifying the optimal L2 penalty, we trained a final model using the optimal parameters on the training set and evaluated it on the held out 10% test set. We did not modify the model after training. We have clarified this workflow in lines 322 - 326.

We would like to thank the authors for clarifying the training procedure. Their detailed description suggests data leakage between the train and test. Indeed, as far as we understand, the authors did not go for k-fold cross-validation, nor for a train/validation/test framework. The same dataset set, splitted in several 90/10 partitions (without control on the overlap between the split rounds) was used for both optimising the hyperparameters and evaluating the model's performance. Proper evaluation requires testing on an independent dataset that was not used for hyperparameters optimisation. The only independent set seems to be the lethal mutation sets that we suggested, but there are no negatives.

c) poor explanation of the categories redundancy

5/ The model combines almost 700 features which limits its interpretability and practical usability. What is the level of redundancy? Could the authors simplify the model and achieve the same performance? While the authors claim on multiple occasions that this complex framework provides a predictive advantage, they actually do not compare the performance of FlyCADD with other methods. How does FlyCADD compare with approaches solely relying on evolutionary conservation, for instance? The authors could consider ProteoCast for proteins (scores readily available for the Fly proteome), GPN-MSA for alignment-based genome-wide predictions, or DNA language models (Evo2) that capture evolutionary conservation without the need for alignments?

The reviewer rightfully points out the large number of annotations incorporated in FlyCADD. When considering redundancy, both the individual annotations and combinations thereof should be considered, and we do think that the level of redundancy is not major. Not all included annotations provide high predictive power when included as individual feature, however, many combined show high predictive power in the model (File S3). For example, PhyloP124 has relatively low predictive power (-0.073), whereas its power is increased when combined with several VEP consequences such as conservation scores in intronic regions (-0.187). Additionally, the concern regarding redundancy has been systematically assessed in the publication describing mouseCADD 10.1186/s12859-018-2337-5), they have shown that the models' performance drops when removing features, especially removal of Ensembl VEP and conservation features. A CADD model where the number of features is reduced might suffice, however, meaningful annotations to improve the predictions in specific regions are present such as secondary structure predictions or conservation scores, and they show discriminative power through combinatorial features as well.

The basis for creating a CADD model is the benefit of combining all available types of annotations into one predictive score, where the features and combinations are weighted based on their performance. The reviewer's comment is valid, in the sense that some hypotheses are more robustly assessed with specific annotation(s) than others. The last years, it has been common to assess ultra-rare highly deleterious variants in human genetic studies by manually assessing several types of annotations and keeping only those with overlaps. For targeted studies, this might very well be the best approach. However, in a more exploratory study, or a hypothesis free such as prioritizing associated SNPs of a GWAS, combining all assessments into one framework, can be much more suitable. As the reviewer states, however, there is no true answer to the correct amount of input.

Dimensionality reduction methods tend to be highly sensitive to the addition of too many extra dimensions, especially if some of them are highly correlated. If we consider previous implementations of CADD, categories are in the range of 30-40, making FlyCADD increasing by an order of magnitude. A proper quantification of its impact should be proposed, especially considering that 95% of them show a low incidence on the model.

d) distribution of scores across the genome and quality of the score

We would like to thank the authors for showing the distribution of all FlyCADD scores on Figure 8. This figure is highly informative. The long-tailed distribution seems to suggest that the beta risk for a SNP with a score >0.9 would be close to 80%. The figure 9 indeed shows that SNPs associated with significant p-values are present across the whole range of FlyCADD scores. These elements support our hypothesis that FlyCADD overpredicts variant effects and does not allow us to understand how the thresholds were defined for SNPs of interest. A clarification would be needed in order to guarantee the confidence in predictions.

e) comparison with existing methods

5/ [...] How does FlyCADD compare with approaches solely relying on evolutionary conservation, for instance? The authors could consider ProteoCast for proteins (scores readily available for the Fly proteome), GPN-MSA for alignment-based genome-wide predictions, or DNA language models (Evo2) that capture evolutionary conservation without the need for alignments?

That part of the question was not answered and it seems of importance that novel methods, even though an adaptation of an existing method to a new organism, are compared with existing ones. In our review we suggested ProteoCast as it defined a dataset for testing drosophila mutational impact prediction functionally, but methods predicting mutational impacts outside of the coding region exist, whether they are based on alignments or language models.

(d) There are a number of questions and comments about the ancestral sequence reconstruction that need to be addressed.

We thank the editor and reviewer for pointing out the difficulties with the description of the ancestral sequence reconstruction. We compared different nodes in the 166-way phylogeny to select the node with the largest ancestral sequence size, least chromosome bias, and least coding sequence bias. This ultimately resulted in an ancestral sequence that matches 16.4 % of the reference genome, while the other reference genome positions correspond to gaps in the ancestral sequence. We have rephrased the ancestral sequence reconstruction description in File S1, as well as in the Methods section to clarify these choices.

It is not clear whether the low percentage of homology between the reconstructed ancestral sequence and the reference sequence can explain the over representation of high scores? This issue should be discussed with the plotting of scores for non-lethal mutations as suggested in comment 1).

3) Impact of FlyCADD predictions outside of CDS

4/ The authors emphasize in the introduction that FlyCADD is not limited to variants in coding regions. Yet, protein coding associated features are among the most contributing ones (e.g. gain of stop codon (Figure 2) : there is an order of magnitude between the 1st parameter and the 3rd,, missense mutations), highlighting the functional importance of variants located in these regions.

While FlyCADD is not limited to variants in coding regions, the strongest predictive features indeed relate to coding consequences, such as “stop gained” or “missense” mutations. This likely reflects a combination of factors: stronger evolutionary constraints on coding variants, leading to clearer depletion signals in the observed (nearly fixed) dataset; and deeper and more consistent annotation coverage for coding regions in existing databases. Consequently, coding sequence associated features can offer greater discriminative power within the model. However, features related to coding sequence also indirectly inform the model about non coding variants. This is because non-coding variants are typically assigned a neutral or zero value for such features, which still carries information: their non-coding nature becomes part of the predictive signal. In this way, coding-related annotations contribute to the model’s discriminative power for all variants.

The phrasing should thus be toned down as FlyCADD then predicts mostly the impact of coding variants. This matter was previously described for other implementations of CADD.

4) Implementation of FlyCADD

(e) Consider usability of FlyCADD by others in the field; this is a key part of the future impact of this work. We agree with the editor that usability of FlyCADD is a key part of its future impact. As setting up a web interface is currently beyond our resources, we added a step-by-step protocol to improve the usability of the resources FlyCADD provides. Future developments may include a web interface, potentially in collaboration with FlyBase.

We thank the authors for making FlyCADD reusable. FlyBase is right now unable to integrate such work considering their lack of funding as far as we are aware of.

5) scores interpretations

Moreover, the interpretation of the weights is not straightforward. While it is easy to understand that premature stop codon contributes negatively, how can the authors explain that stop codon loss has a high positive score? Is this a numerical or statistical artefact? If one looks at Figure 4, variants annotated as stop codon gain or stop codon loss have both very narrow distributions toward high scores (meaning high impact). Is there an enrichment in mutations altering stop codons in the training set? Do the features include stop codons that are outside the CDS? What is the value of relCDSpos in that case? In the seminal CADD paper (doi: 10.1038/ng.2892), the authors construct a series of univariate models that contrast observed and simulated variants. This allows for exploring the training set for potential biases and it could also be useful here.

Importantly, the weights shown in Figure 2 reflect the discriminative power of each feature within the logistic regression model, not the biological impact or directionality of the variant itself. For example, the high weight assigned to the “stop loss” feature does not imply a beneficial effect but rather indicates that this annotation strongly helps distinguish simulated from observed variants. The score distribution (Figure 4) shows what the model predicts; the feature weight shows how that variable helps make the prediction. Feature weights describe how informative a feature is for classification in the context of all other features and should not be interpreted as stand-alone indicators of biological effect. To clarify this, we have revised the main text (lines 404 – 409 and 418 - 426) to better explain how feature weights reflect discriminative power, not impact direction or magnitude. For the same reason, we have also updated Figure 2 to more accurately reflect the predictive power of each feature in the model. **We emphasize that the direction of the weights reflect the way our annotations were encoded and do not imply biological meaning at lines 331 - 334.**

Regarding stop codons outside of coding region, stop codon features are not present outside coding sequences, and therefore our features do not include such stop codons. The Ensembl VEP is used to annotate these features, and their severity order was applied in case variants could be annotated with multiple consequences. The feature “relCDSpos” indicates the position within coding sequence for those variants that are in coding sequence. This feature remains empty for those variants outside of coding regions (the value is 0 in the encoded annotation output). However, this value is also used by the model to discriminate variants. Enrichment of coding-related features (such as stop codon gain or missense mutations) in the top contributors reflects the biological relevance of these sites and is not an artifact. The relevance can be interpreted as a depletion of loci in the observed set of (nearly) fixed derived variants, likely because of purifying selection against such mutations on a longer evolutionary timescale. The simulated training set intentionally contains high-impact simulated variants, not to balance the classes, but to maximize discriminative signal. Note, however, that the frequency of the high-impact variants in the set of simulated variants is determined by mutation rate and is determined randomly by the simulator. Therefore, we did not artificially enrich this set with high-impact variants but simply rely on the proportions that arise under a neutral scenario, without (purifying or balancing) selection. As indicated on line 286 – 288, we have added a figure to File S3 reflecting the proportions of variants with different VEP-annotated consequences in the derived and simulated training datasets.

We would like to thank the authors for clarifying the meaning of the weights’ signs.

The authors chose to represent predictions for mutants of the *white* gene and we previously highlighted that they did not represent functional variants. They gave a partial response with the sentence below in red.

lines 465 - 467 + Figure 5 : the authors suggest that their tool can help discriminate variants within the same region and give the example of the white gene but do not represent known functional variants on the sequence

As stated above, very limited numbers of SNPs are functionally validated in *D. melanogaster*. Figure 5 is included in the manuscript to illustrate the patterns within genes, such as intronic and exonic regions. Based on a manual annotation one would assume exonic variants are more impactful than intronic variants or even intergenic variants. FlyCADD can be applied to discriminate variants within these regions, as can be seen by variants spanning the entire range of FlyCADD scores within exons and within introns. However, we do agree with the reviewer that the depiction of only theoretically present variants invariants in the gene *white*, namely all possible variants across this region on the X chromosome, becomes more informative when known variants are presented. **Functional variants that are experimentally validated within this region are rare.** Therefore, we added naturally occurring variants of the DEST2.0 resource to the figure. The figure now includes all possible variants in purple, and highlighted with blue diamonds are all variants naturally occurring in worldwide *D. melanogaster* populations, as indicated in the figure legend and at line 473.

Multiple mutations are described in Mackenzie et al., 1999 with a functional impact. Could the authors locate them on their prediction map and indicate their predicted scores?

3/ If the "set of simulated variants is thought to contain the full spectrum of deleteriousness", then why is the "probability of the variant belonging to the simulated variants class" (Set2) considered by the authors as equivalent to the "predicted functional impact to fitness"? In other words, why would a variant likely to belong to Set2 necessarily have high functional impact? It would be more intuitive to adopt the reciprocal perspective, i.e., reason with respect to the neutral class: a variant with a high probability to belong to Set1 is likely neutral.

We regret that the reviewer finds the description and application of set 1 and set 2 not intuitive and have changed this section to clarify the design choices for the two sets. FlyCADD is based on probabilities, not on impact scores themselves. The logistic regression model underlying FlyCADD is set up to differentiate between simulated (set 2) and derived (set 1) variants and returns two probabilities for each variant it scores: the probability of a variant belonging to set 1, and the probability of the variant belonging to set 2. Compared to set 1, set 2 is enriched in impactful variants as natural selection has not acted upon these simulated variants, therefore, we have chosen to report the probability of a variant belonging to set 2 as a proxy for functional impact. The two probabilities returned by the logistic regression model equal to 1 for each prediction, and thus, adopting the reciprocal will not change the interpreted impact of a variant. On lines 224 - 226, we have rephrased the description of set 2 and on lines 234 - 236 we've added context regarding the probabilities as computed by the logistic regression model.

The issue with the two sets is not merely the interpretation of the probability to belong to one or the other but the meaning of it. The interpretation of set2 as being enriched with deleterious mutants is an extrapolation that should be tested regarding non-*Drosophila* organisms.

Moreover, the fact that the proportion of predicted low-impact scores is small questions the pertinence of the design choices for Set1 and Set2. According to the literature, one would expect most of the variants to be neutral/benign or well tolerated. It seems that the design adopted by the authors, imposing strict criteria for Set1 and more relaxed ones for Set2, leads to over-predicting pathogenicity (or functional impact).

The reviewer correctly points out that one would expect most naturally occurring variants to be not impactful. However, it is important to note that the dataset we present in Figure 3 is not a representation of tolerated or naturally present variants but rather shows the FlyCADD scores across all possible variants (> 400 million variants). Figure 7 does reflect naturally occurring variants from DEST2.0 and indeed shows that most variants are neutral and benign as expected (~ 4.5 million variants). We have added clarification on the

nature of the dataset depicted in Figure 3 at lines 435 - 440.

A likely explanation could be an over estimation of high impact scores due to a) the low percentage of homology between the reference genome and the inferred ancestral genome and b) taking only the *Drosophila* genomes as a reference for deciding what polymorphism hasn't been explored.

Representing the distribution of the scores specifically on non-lethal mutations compared to that of the lethal ones would help figuring that point out.

- since the conclusions of the paper are that the most significant SNPs actually fall within coding sequence, why not comparing with the recent cited ProteoCast that restricts its predictions to the proteome. We have demonstrated the use of FlyCADD across the entirety of the *D. melanogaster* genome and do not intend to conclude that the most significant SNPs fall in coding sequence from the presented analyses. However, we do acknowledge that the most predictive features include features restricted to coding region. With the inclusion of combinatorial features, we ensured predictive power present irrespective of the genomic position of variants. The comparison with ProteoCast presents itself with several difficulties regarding the aim of the predictive tool, the range of applications and the continuous or binarized score approach. The comparison of FlyCADD with other VEPs is outside of the scope of this manuscript. At lines 115 – 119, we describe these different properties of VEPs. However, with the inclusion of FlyBase lethal point mutations, we have validated our FlyCADD scores in a similar manner as ProteoCast, with similar outcomes. We have acknowledged this in the discussion at lines 737 - 742.

We find it hard to defend the argument for not comparing with other methods as being outside the scope of this paper. As the authors rightfully emphasize, FlyCADD is designed and trained specifically on one organism/species with a large number of organism-specific annotations. Hence, one would expect that it performs much better than general-purpose VEP on this particular organism. It is thus essential that the authors quantitatively assess the gain in predictive performance of FlyCADD compared to these VEPs.

July 22, 2025

GENETICS-2025-308265

Predicting the functional impact of single nucleotide variants in *Drosophila melanogaster* with FlyCADD

Dear Dr. Beets:

Two experts in the field have reviewed your manuscript, and I have read it as well. While your manuscript is not currently acceptable for publication in GENETICS, we would welcome a substantially revised manuscript. Both reviewers have identified additional information that needs to be added to a further revised manuscript. You can read their reviews at the end of this email.

It is critical that the new protocol is included in the Zenodo repository and that the comments of reviewer 2 (in the summary below and in the attached, annotated file) are more fully addressed, including more controls and benchmarking to increase the rigor of the work.

We look forward to receiving your revised manuscript. Please let the editorial office know approximately how long you expect to need for revisions.

Upon resubmission, please include:

1. A clean version of your manuscript;
2. A marked version of your manuscript in which you highlight significant revisions carried out in response to the major points raised by the editor/reviewers (track changes is acceptable if preferred);
3. A detailed response to the editor's/reviewers' feedback and to the concerns listed above. Please reference line numbers in this response to aid the editor and reviewers.

Your paper will likely be sent back out for review.

Additionally, please ensure that your resubmission is formatted for GENETICS
<https://academic.oup.com/genetics/pages/general-instructions>

Follow this link to submit the revised manuscript: Link Not Available

Sincerely,

Patricia Wittkopp
Associate Editor
GENETICS

Approved by:
Stanley Fields
Senior Editor
GENETICS

Reviewer #1 :

Dear authors, thank you very much for the extensive revision of your manuscript "Predicting the functional impact of single nucleotide variants in *Drosophila melanogaster* with FlyCADD". My concerns are fully addressed. Special thanks for the provision of a hands-on protocol, which will be very helpful to access the data. Maybe I missed it, but the protocol does not seem to be uploaded to the zenodo repository as mentioned in the revised manuscript. This should be done prior to publication of the manuscript. Greetings, Nico Posnien

Reviewer #2 :

We thank the authors for having taken into consideration most of our suggestions. We continue to think that the tool represents a deep interest to the *Drosophila* community, however, multiple points still require attention. We present an overview here for clarity, followed by a full answer inline with the authors' responses.

the functional validation by using a list of known lethal mutations has been done without comparison to non-lethal ones, their list

is absent from the manuscript

a certain number of vague points still remain to be addressed, the over inflation of categories compared to previous CADDs as well as their redundancy, the data circularity and data leakage issues still requires a thorough description of the training/test sets, the term accuracy is used multiple times out of context, no accuracy is presented (see point 1).

the ability of FlyCADD to really predict outside of CDS

the absence of comparison with other existing methods, beyond ProteoCast.

The detailed comments are presented in the PDF.

Response to Reviewers

Reviewer #1

Dear authors, thank you very much for the extensive revision of your manuscript "Predicting the functional impact of single nucleotide variants in *Drosophila melanogaster* with FlyCADD". My concerns are fully addressed. Special thanks for the provision of a hands-on protocol, which will be very helpful to access the data. Maybe I missed it, but the protocol does not seem to be uploaded to the zenodo repository as mentioned in the revised manuscript. This should be done prior to publication of the manuscript. Greetings, Nico Posnien

We thank the reviewer for their comments that guided the revisions of our manuscript. The step-by-step protocol has now been added to the Zenodo repository.

Reviewer #2

We would like to thank the authors for having taken into consideration most of our suggestions. The additional validation experiments they performed clearly emphasize limitations of the method that should be discussed in the main manuscript text. While we continue to think that the tool represents a deep interest to the *Drosophila* community, multiple points still require attention. We present an overview here for clarity, followed by a full answer inline with the authors' responses.

We appreciate that the reviewer acknowledged the value of the revisions we have made to the manuscript. We have responded below to the points still suggested by the reviewer.

- 1) The functional validation on a set of known lethal mutations only provides a recall estimate. It is impossible to compute or assess predictive accuracy in the absence of a negative set (i.e. non-lethal mutations). Moreover, the authors did not provide the list of lethal mutations they used, which compromises reproducibility.**

Validation with lethal mutations was added based on the suggestion of the reviewer during the first review. A validated negative set containing non-lethal mutations does not exist, as every nucleotide present in the genome of living individuals is by definition non-lethal. Non-lethal does not imply non-impactful, which is what is scored by FlyCADD. Therefore, such a negative set is also not included in our manuscript. Despite the absence of a negative validation set, we argue that the analysis of validated lethal mutations provides important information about the performance of the model as we have detailed in our previous response letter. If the reviewer disagrees with this, this analysis can also be moved to supplementary material. Further response to the concerns regarding FlyCADD performance can be found in response to comment 2.

The reviewer is correct that the predictive accuracy cannot be computed in an independent dataset. We addressed this in the Results (line 524-536) and only described the metric calculated using a held-out testing dataset as "accuracy".

We regret that we forgot to add the list of lethal mutations to our previous resubmission. We now added the list of these mutations and their FlyCADD scores as used for Figure 7 to Zenodo to ensure reproducibility of this experiment (as indicated on line 380).

- 2) A certain number of vague points still remain to be addressed, the over inflation of descriptive features compared to previous CADDs as well as their redundancy, the data circularity and data leakage issues still require a thorough description of the training/test sets, the term accuracy is used multiple times out of context, no accuracy is presented (see point 1).**

We regret that a number of points is still not clear to the reviewer. However, we feel it is difficult to respond to this comment because it is not very specific. We tried to dissect the criticism into the following separate concrete points:

- Overinflation/dimensionality: The number of categories ($n = 38$) and the number of features ($n = 691$, includes combinations of annotations) in FlyCADD is not overinflated compared to the scale of features used in other CADD implementations, such as humanCADD or pigCADD, and thus does not represent an unusual increase in dimensionality or overinflation. For example, humanCADD reports the implementation of 63 annotation categories, which result in 949 features, mouseCADD 68 annotations making up 931 features, and pCADD reports 39 annotations resulting in 867 features.
- Redundancy: As indicated in our first response letter, the redundancy of the features is assessed using the weight of each feature in the model. We describe that all annotations are involved in the final score, mostly through the incorporation in combinatorial features.
- Data circularity: This comment seems to be related to the DEST2.0 variants in Figure 8. These variants are presented as use-case of FlyCADD, not as validation of the impact prediction scores and, therefore, data circularity is not a valid concern.
- Data leakage: The training procedure was described in our Methods section and in our first response letter. Data leakage would imply the use of training data during model testing. We kept the held-out test-set of 10% separate from model training at all times, to ensure data leakage would not occur. We have now rephrased the Methods section accordingly to avoid this misunderstanding (described in response to the comment below, line 325).
- Accuracy: Unfortunately, the reviewer does not provide us with line numbers of the incorrect usage of the term “accuracy”. Therefore, we checked all occurrences across the manuscript and changed one where the term was not used correctly (line 546).
For clarity, we define at first occurrence in line 341 what is meant with accuracy, in concordance with previous CADD papers. We present the accuracy of the model based on the held-out dataset. Accuracy is the overall performance of the model, and an additional measure of model performance is AUC-ROC. We already provided both (as computed based on the held-out testing dataset) in the manuscript at lines 400 – 402.
Moreover, as we described in our previous rebuttal letter, we added lethal mutations as a third metric for model performance. This metric is recall as the reviewer points out, however, it was not presented in our manuscript as accuracy metric, but rather as performance metric. To avoid confusion with the term “accuracy”, we have corrected the description of the additional validation analysis to properly reflect the outcome from lethal mutations, namely the recall metric (line 546).
- Overprediction: The reviewer suggests overprediction of variant effect by FlyCADD, however, this is not shown in our analyses (more details provided in the comments below).

3) The ability of FlyCADD to make biologically or clinically relevant predictions outside of CDS remains unclear

We feel that the reviewer overlooks the evidence for relevant predictions across an entire genome using previous applications of the CADD framework. We agree not everything is known yet about non-coding regions, but we did show for example relevant predictions in agreement with functional studies in non-coding regions (Figure 10) and functionally relevant variants genome editing experiments outside coding region (final use-case). As emphasized before, a substantial validation dataset is not present, but the model is built upon a thoroughly validated and benchmarked framework.

To better emphasize the benchmarking efforts done for CADD framework performance in coding and non-coding sequence, we have added a paragraph in the Discussion. This section discusses the previous applications and benchmarking of the CADD framework (in human). We emphasize that there is no validation dataset available to test this for *D. melanogaster* directly, but that the CADD framework has been widely adopted.

4) Comparison with other existing methods, which is essential for assessing the pertinence and relevance of the proposed novel method, is still missing from the manuscript The recent literature about variant effect predictors is extremely active and abundant, with community-wide efforts for

benchmarking novel methods. Since the central and main contribution of the manuscript is a novel method for predicting variant outcome, the authors should more carefully acknowledge relevant literature and explain how their proposed methodology compares to others (including those that use much less features or rely on unsupervised language models).

We agree with the reviewer that comparing existing VEP methods is important. The CADD framework is one of these existing VEP methods and has been benchmarked in multiple studies using multiple methods, which is why we now refer to the available comparisons between CADD and other VEPs in the Discussion (lines 795-829). We remain with our opinion that benchmarking is beyond the scope of this paper due to both technical limitations regarding datasets as well as its incompatibility with the scope of this study.

1) FlyCADD scores can indeed address key questions in genotype-phenotype interactions

Associate Editor:

(a) Clarify the specific aspects that FlyCADD scores can indeed address key questions in genotype-phenotype interactions, discussing the limitations and the need for fine-scale genetic mapping to resolve certain complexities.

We have now added more nuance regarding the complexity of the genotype-phenotype link and the limitations of FlyCADD in research related to this topic. FlyCADD is intended as complementary tool to association studies and functional genomics by providing an impact prediction score to understand functional variant impact. Additionally, we added an analysis of known lethal point mutations to demonstrate the correlation of predicted impact by FlyCADD with empirically validated functional effects, as suggested by reviewer #2.

Authors should clearly state the methods, papers and analyses that inspired their validation in the main body of the preprint. In addition, no information is given regarding the specificity of those predictions. They should calculate an accuracy of prediction using the distribution of scores for non-lethal mutations. In addition, no table with the used mutations and their predicted scores is presented. How can one reproduce this analysis?

This response is a continuation of our response to Overview 1 above.

We now state the basis of our validation more clearly:

- At lines 372-374, we specify the goal of the inclusion of lethal mutations.
- At lines 524 – 536, we elaborate on our validation using lethal mutations.
- At line 534, we specify that specificity is not calculated.

The validation method does not include specificity as a curated set of known benign/neutral mutations is not available for *D. melanogaster*. Therefore, we only present model performance metrics based on the held-out set (accuracy and AUC) and performance on a curated dataset of lethal mutations.

2) Address points the reviewers identified as vague

a) over inflation of categories compared to other CADDs (x10)

b) data circularity / data leakage

2/ The approach builds upon the variant effect predictor (VEP) CADD (doi: 10.1038/ng.2892), which underperforms compared to other VEPs in recent benchmark studies, e.g. see doi: 10.15252/msb.202211474 or proteigym.org). Moreover CADD is a supervised method, introducing the risk of data circularity.

FlyCADD indeed builds upon the CADD frameworks established for application to the human genome and the pig genome (pCADD). The original CADD framework has been developed to predict the functional impact of human genetic variants, with a focus on pathogenic variants. The paper referred to by the reviewer is a benchmarking study showing a comprehensive comparison of many VEPs. However, many of the higher performing VEPs are developed specifically i) to predict pathogenicity, ii) to be applied in the human genome, or iii) to make predictions predominantly for coding sequence, whereas the CADD is genome-wide VEP with the potential for species-specific application. Since our goal is to assess functional impact, genome-wide in *D. melanogaster*, rather than quantify pathogenicity, the CADD framework is chosen. It is possible that other methods have more predictive power for coding sequence variants, however, CADD is capable of

ranking variants outside of coding regions, and species-specificity increases its predictive power on these respective genomes compared to tools specifically designed focused on a single genome/species, lines 133 - 135. We now refer to the recent benchmark studies comparing traditional VEPs and CADD to address these issues at lines 129 – 133. While CADD is indeed a supervised method, the risk of data circularity is minimal due to the nature of its training data. Unlike models trained directly on clinically labelled pathogenic and benign variants - where the same datasets are often reused for both training and evaluation - CADD is trained to distinguish between naturally derived variants and simulated variants. This framework is evolutionarily informed, rather than clinically annotated, and does not include pathogenicity labels or known functional variants in training. As a result, CADD minimizes data circularity: **it does not train on the same variant categories it is later evaluated on**. In Figure 4, the consequences that are used to stratify the variants are VEP consequences, these are incorporated as annotations in the FlyCADD model and, therefore, could be a minor source for circularity within the figure. In our application, we further reduce potential bias by ensuring the training dataset spans the entire genome and is not manually enriched for specific chromosomal regions or even coding regions, which would introduce biased predictions in these regions. Importantly, we do not use the training data to evaluate the FlyCADD model or benchmark predictions. The dataset was split to use 90 % for model training and 10 % for model evaluation. Thus, the supervised nature of CADD does not introduce circularity in our context. The additional validation and score evaluations show the performance of FlyCADD without indications for circularity.

As far as we understood the manuscript, this statement is wrong. The only AUC and accuracy reported in the paper were computed on the 10% test data held out during training; this evaluation is performed on exactly the same categories it was trained on.

This is a misunderstanding of our previous response, which was not about the testing dataset but rather about the score evaluations such as score distributions and codon impact.

The additional statement about AUC and held-out dataset is exactly what we have done, reported and how this logistic regression model should be evaluated. The procedure for model validation in the absence of a distinct and curated validation set is to test model performance based on a held-out set of data. This evaluation reports the model's ability to distinguish between these two variant categories on unseen data drawn from the same distribution, and therefore, the training and testing dataset have the same categories, on purpose. For FlyCADD, this is what we have done, and the 10% held-out test set shares the same variant categories ("derived" and "simulated") as the training data to report the model's ability to distinguish these, the model metrics are already reported as such in the manuscript (lines 401-402 and 344-346).

Moreover, it is not clear whether the 46 DEST1 used for training are included in the 529 DEST2 reported during evaluation: it seems to be the case, which would imply data leakage between train and test: the variants used in set 1 of training again appear in the DEST2 evaluation.

Again, this seems to be a misinterpretation of the reviewer of the section about case studies. This section is meant for demonstration purposes on the application of FlyCADD, not for model evaluation. Data leakage would mean the usage of training data for testing purposes. Figure 8 is presented as use-case and not as validation method for the scores, therefore, it does not act as testing method to determine specificity and does not induce data leakage.

6/ The description of the validation of the model is unclear. Did the author perform k-fold cross-validation on 90% of the training set and performed the blind final evaluation on the held-out 10%? If yes, what value of k did they choose? Did they average the models trained on each fold? If not, then why do they mention "cross-validation"? And why was the model still modified after this cross-validation (lines 300-305)?

The reviewer rightfully points out the unclear model validation. We have realized that the term cross-validation was not appropriate in this context. We applied sub-sampling validation to determine the optimal L2 penalty value, followed by training of the final model. Specifically, we split the dataset (90% training, 10% testing) five times for each value of L2 (0.01, 0.1, 1.0, 10.0, 100.0) and selected the model with the highest average accuracy. This process was intended solely for hyperparameter tuning. After identifying the optimal L2 penalty, we trained a final model using the optimal parameters on the training set and evaluated

it on the held out 10% test set. We did not modify the model after training. We have clarified this workflow in lines 322 - 326.

We would like to thank the authors for clarifying the training procedure. Their detailed description suggests data leakage between the train and test. Indeed, as far as we understand, the authors did not go for k-fold cross-validation, nor for a train/validation/test framework. The same dataset set, splitted in several 90/10 partitions (without control on the overlap between the split rounds) was used for both optimising the hyperparameters and evaluating the model's performance. Proper evaluation requires testing on an independent dataset that was not used for hyperparameters optimisation. The only independent set seems to be the lethal mutation sets that we suggested, but there are no negatives.

We regret that our extensive description of model training and testing was unclear. To avoid any misinterpretation, we updated the Methods section describing the training procedure (line 325). The entire training dataset encompasses 5.467.390 variants and was initially split into 90% for training and 10 % for testing. The 10 % testing dataset was held out and not used for any model training or hyperparameter tuning, serving solely for final evaluation of model performance through the reported ROC-AUC.

The hyperparameter tuning was performed exclusively on the 90% of the dataset that was allocated for training. This training subset was internally partitioned into 90/10 sets for optimizing the parameter.

After identifying the optimal parameter, the final model was trained on the entire 90% of the training dataset using this L2 value and then evaluated on the held-out 10 % test set. This approach ensured that the test set remained independent of any model training or hyperparameter optimization, thereby preventing data leakage and providing an unbiased metric of model performance. We have now stated this explicitly on lines 326-335.

c) poor explanation of the categories redundancy

5/ The model combines almost 700 features which limits its interpretability and practical usability. What is the level of redundancy? Could the authors simplify the model and achieve the same performance? While the authors claim on multiple occasions that this complex framework provides a predictive advantage, they actually do not compare the performance of FlyCADD with other methods. How does FlyCADD compare with approaches solely relying on evolutionary conservation, for instance? The authors could consider ProteoCast for proteins (scores readily available for the Fly proteome), GPN-MSA for alignment-based genome-wide predictions, or DNA language models (Evo2) that capture evolutionary conservation without the need for alignments?

The reviewer rightfully points out the large number of annotations incorporated in FlyCADD. When considering redundancy, both the individual annotations and combinations thereof should be considered, and we do think that the level of redundancy is not major. Not all included annotations provide high predictive power when included as individual feature, however, many combined show high predictive power in the model (File S3). For example, PhyloP124 has relatively low predictive power (-0.073), whereas its power is increased when combined with several VEP consequences such as conservation scores in intronic regions (-0.187). Additionally, the concern regarding redundancy has been systematically assessed in the publication describing mouseCADD 10.1186/s12859-018-2337-5), they have shown that the models' performance drops when removing features, especially removal of Ensembl VEP and conservation features. A CADD model where the number of features is reduced might suffice, however, meaningful annotations to improve the predictions in specific regions are present such as secondary structure predictions or conservation scores, and they show discriminative power through combinatorial features as well. The basis for creating a CADD model is the benefit of combining all available types of annotations into one predictive score, where the features and combinations are weighted based on their performance. The reviewer's comment is valid, in the sense that some hypotheses are more robustly assessed with specific annotation(s) than others. The last years, it has been common to assess ultra-rare highly deleterious variants in human genetic studies by manually assessing several types of annotations and keeping only those with overlaps. For targeted studies, this might very well be the best approach. However, in a more exploratory study, or a hypothesis free such as prioritizing associated SNPs of a GWAS, combining all assessments into one

framework, can be much more suitable. As the reviewer states, however, there is no true answer to the correct amount of input.

Dimensionality reduction methods tend to be highly sensitive to the addition of too many extra dimensions, especially if some of them are highly correlated. If we consider previous implementations of CADD, categories are in the range of 30-40, making FlyCADD increasing by an order of magnitude. A proper quantification of its impact should be proposed, especially considering that 95% of them show a low incidence on the model.

As we have stated in our response to Overview 2, this comment is not correct. Previous implementations of CADD used similar numbers of annotations and features.

As reported on lines 235-236, FlyCADD logistic regression model is trained on the 691 combinatorial features, encompassing 38 individual annotations (Supplementary File S2). This distinction between the number of annotations and features is important to understand the dimensionality of these categories which the reviewer refers to. The previous implementations of CADD indeed report numbers of categories in the range of 30-40, which is in fact similar to the number of annotations implemented in FlyCADD (n = 38) and thus FlyCADD does not represent an unusual increase in dimensionality.

Throughout the manuscript, we refer consistently to the 691 dimensions as “features” and to the 38 categories as “annotations” to indicate this distinction. All features are included in a genome-wide manner, each position has a value for all features. Feature importance is discussed by presenting their weights (line 409, Supplementary File S3), indicating that all annotations contribute, especially as part of the combinatorial features.

d) distribution of scores across the genome and quality of the score

We would like to thank the authors for showing the distribution of all FlyCADD scores on Figure 8. This figure is highly informative. The long-tailed distribution seems to suggest that the beta risk for a SNP with a score >0.9 would be close to 80%. The figure 9 indeed shows that SNPs associated with significant p-values are present across the whole range of FlyCADD scores. These elements support our hypothesis that FlyCADD overpredicts variant effects and does not allow us to understand how the thresholds were defined for SNPs of interest. A clarification would be needed in order to guarantee the confidence in predictions.

It is unclear to us how the reviewer gets to 80 % beta risk; therefore, we cannot respond to this comment. The thresholds proposed are not cut-off values as FlyCADD scores are a continuous measure with the goal of variant prioritization rather than exact scoring. This is indicated multiple times in the manuscript.

Additionally, it is unclear to us how overprediction by FlyCADD is detected, our analyses do not indicate overprediction of variant effects by FlyCADD, as shown by several use-cases, validation analysis and model metrics.

e) comparison with existing methods

5/ [...] How does FlyCADD compare with approaches solely relying on evolutionary conservation, for instance? The authors could consider ProteoCast for proteins (scores readily available for the Fly proteome), GPN-MSA for alignment-based genome-wide predictions, or DNA language models (Evo2) that capture evolutionary conservation without the need for alignments?

That part of the question was not answered and it seems of importance that novel methods, even though an adaptation of an existing method to a new organism, are compared with existing ones. In our review we suggested ProteoCast as it defined a dataset for testing drosophila mutational impact prediction functionally, but methods predicting mutational impacts outside of the coding region exist, whether they are based on alignments or language models.

As indicated in the Overview, a full comparison with existing effect predictors is out of the scope of this manuscript. We do acknowledge the presence of alternative methods, with their unique areas of focus compared to FlyCADD in our manuscript, as indicated in our first response letter and now in the Discussion. These tools have different goals, focus species and outcome datatypes, which makes it necessary to conduct a separate benchmarking study for thorough comparison.

(d) There are a number of questions and comments about the ancestral sequence reconstruction that need to be addressed.

We thank the editor and reviewer for pointing out the difficulties with the description of the ancestral sequence reconstruction. We compared different nodes in the 166-way phylogeny to select the node with

the largest ancestral sequence size, least chromosome bias, and least coding sequence bias. This ultimately resulted in an ancestral sequence that matches 16.4 % of the reference genome, while the other reference genome positions correspond to gaps in the ancestral sequence. We have rephrased the ancestral sequence reconstruction description in File S1, as well as in the Methods section to clarify these choices.

It is not clear whether the low percentage of homology between the reconstructed ancestral sequence and the reference sequence can explain the over representation of high scores? This issue should be discussed with the plotting of scores for non-lethal mutations as suggested in comment 1).

It is not clear to us why the reviewer suspects overrepresentation of high scores. This is not indicated from our analyses. Overrepresentation of high scores can be a result of unbalanced or biased training datasets. Importantly, our model was trained using a balanced dataset comprising equal numbers of simulated and derived variants, designed to prevent bias toward high or low scores. Ancestral alleles were filtered as described in "Methods" to represent derived variants. Additionally, we assessed and minimized biases related to chromosome representation and coding sequence content during the selection of the ancestral node, as described in Supplementary File S1. Thereby, we minimized the biases that could result from incomplete ancestral sequence reconstruction. At present, we have no evidence that regions lacking ancestral sequence coverage are enriched for functionally impactful variants or that they introduce systematic bias into the model.

3) Impact of FlyCADD predictions outside of CDS

4/ The authors emphasize in the introduction that FlyCADD is not limited to variants in coding regions. Yet, protein coding associated features are among the most contributing ones (e.g. gain of stop codon (Figure 2) : there is an order of magnitude between the 1st parameter and the 3rd,, missense mutations), highlighting the functional importance of variants located in these regions.

While FlyCADD is not limited to variants in coding regions, the strongest predictive features indeed relate to coding consequences, such as "stop gained" or "missense" mutations. This likely reflects a combination of factors: stronger evolutionary constraints on coding variants, leading to clearer depletion signals in the observed (nearly fixed) dataset; and deeper and more consistent annotation coverage for coding regions in existing databases. Consequently, coding sequence associated features can offer greater discriminative power within the model. However, features related to coding sequence also indirectly inform the model about non coding variants. This is because non-coding variants are typically assigned a neutral or zero value for such features, which still carries information: their non-coding nature becomes part of the predictive signal. In this way, coding-related annotations contribute to the model's discriminative power for all variants.

The phrasing should thus be toned down as FlyCADD then predicts mostly the impact of coding variants. This matter was previously described for other implementations of CADD.

*We respectfully disagree with the reviewer on this point. As FlyCADD is both trained on coding and non-coding training variants, as well as coding and non-coding annotations, the predictions FlyCADD makes are based on this integrative approach combining these large sets of data. The notion that FlyCADD scores might be more empowered in coding regions, due to more annotations for coding regions is emphasized multiple times in the manuscript already (line 136, 807). This does, however, not automatically imply that the impact predictions for non-coding regions are less reliable. At the state of the art, FlyCADD is the only prediction tool for *D. melanogaster* that integrates annotations on non-coding regions. This makes it a valuable tool to investigate impact of non-coding variants, which is a largely unexplored field to date.*

4) Implementation of FlyCADD

(e) Consider usability of FlyCADD by others in the field; this is a key part of the future impact of this work.

We agree with the editor that usability of FlyCADD is a key part of its future impact. As setting up a web interface is currently beyond our resources, we added a step-by-step protocol to improve the usability of the resources FlyCADD provides. Future developments may include a web interface, potentially in collaboration with FlyBase.

We thank the authors for making FlyCADD reusable. FlyBase is right now unable to integrate such work considering their lack of funding as far as we aware of.

We agree that recent developments at FlyBase make it unfortunately uncertain if and when FlyCADD could be integrated. To improve usability, the step-by-step protocol is now added to the Zenodo repository.

5) scores interpretations

Moreover, the interpretation of the weights is not straightforward. While it is easy to understand that premature stop codon contributes negatively, how can the authors explain that stop codon loss has a high positive score? Is this a numerical or statistical artefact? If one looks at Figure 4, variants annotated as stop codon gain or stop codon loss have both very narrow distributions toward high scores (meaning high impact). Is there an enrichment in mutations altering stop codons in the training set? Do the features include stop codons that are outside the CDS? What is the value of relCDSpos in that case? In the seminal CADD paper (doi: 10.1038/ng.2892), the authors construct a series of univariate models that contrast observed and simulated variants. This allows for exploring the training set for potential biases and it could also be useful here. Importantly, the weights shown in Figure 2 reflect the discriminative power of each feature within the logistic regression model, not the biological impact or directionality of the variant itself. For example, the high weight assigned to the “stop loss” feature does not imply a beneficial effect but rather indicates that this annotation strongly helps distinguish simulated from observed variants. The score distribution (Figure 4) shows what the model predicts; the feature weight shows how that variable helps make the prediction. Feature weights describe how informative a feature is for classification in the context of all other features and should not be interpreted as stand-alone indicators of biological effect. To clarify this, we have revised the main text (lines 404 – 409 and 418 - 426) to better explain how feature weights reflect discriminative power, not impact direction or magnitude. For the same reason, we have also updated Figure 2 to more accurately reflect the predictive power of each feature in the model. We emphasize that the direction of the weights reflect the way our annotations were encoded and do not imply biological meaning at lines 331 - 334. Regarding stop codons outside of coding region, stop codon features are not present outside coding sequences, and therefore our features do not include such stop codons. The Ensembl VEP is used to annotate these features, and their severity order was applied in case variants could be annotated with multiple consequences. The feature “relCDSpos” indicates the position within coding sequence for those variants that are in coding sequence. This feature remains empty for those variants outside of coding regions (the value is 0 in the encoded annotation output). However, this value is also used by the model to discriminate variants. Enrichment of coding-related features (such as stop codon gain or missense mutations) in the top contributors reflects the biological relevance of these sites and is not an artifact. The relevance can be interpreted as a depletion of loci in the observed set of (nearly) fixed derived variants, likely because of purifying selection against such mutations on a longer evolutionary timescale. The simulated training set intentionally contains high-impact simulated variants, not to balance the classes, but to maximize discriminative signal. Note, however, that the frequency of the high-impact variants in the set of simulated variants is determined by mutation rate and is determined randomly by the simulator. Therefore, we did not artificially enrich this set with high-impact variants but simply rely on the proportions that arise under a neutral scenario, without (purifying or balancing) selection. As indicated on line 286 – 288, we have added a figure to File S3 reflecting the proportions of variants with different VEP-annotated consequences in the derived and simulated training datasets.

We would like to thank the authors for clarifying the meaning of the weights' signs. The authors chose to represent predictions for mutants of the *white* gene and we previously highlighted that they did not represent functional variants. They gave a partial response with the sentence below in red.

lines 465 - 467 + Figure 5 : the authors suggest that their tool can help discriminate variants within the same region and give the example of the *white* gene but do not represent known functional variants on the sequence

As stated above, very limited numbers of SNPs are functionally validated in *D. melanogaster*. Figure 5 is included in the manuscript to illustrate the patterns within genes, such as intronic and exonic regions. Based on a manual annotation one would assume exonic variants are more impactful than intronic variants or even intergenic variants. FlyCADD can be applied to discriminate variants within these regions, as can be seen by variants spanning the entire range of FlyCADD scores within exons and within introns. However, we do agree with the reviewer that the depiction of only theoretically present variants invariants in the gene *white*, namely all possible variants across this region on the X chromosome, becomes more informative when known variants are presented. Functional variants that are experimentally validated within this region are rare. Therefore, we added naturally occurring variants of the DEST2.0 resource to the figure. The figure now includes all possible variants in purple, and highlighted with blue diamonds are all variants

naturally occurring in worldwide *D. melanogaster* populations, as indicated in the figure legend and at line 473.

Multiple mutations are described in Mackenzie et al., 1999 with a functional impact. Could the authors locate them on their prediction map and indicate their predicted scores?

We thank the reviewer for pointing out these variants and we have updated Figure 5 in the manuscript (see updated Figure below), as well as line 485. These variants are phenotypically validated for functional impact and also are predicted to be of high functional importance by FlyCADD.

3/ If the "set of simulated variants is thought to contain the full spectrum of deleteriousness", then why is the "probability of the variant belonging to the simulated variants class" (Set2) considered by the authors as equivalent to the "predicted functional impact to fitness"? In other words, why would a variant likely to belong to Set2 necessarily have high functional impact? It would be more intuitive to adopt the reciprocal perspective, i.e., reason with respect to the neutral class: a variant with a high probability to belong to Set1 is likely neutral.

We regret that the reviewer finds the description and application of set 1 and set 2 not intuitive and have changed this section to clarify the design choices for the two sets. FlyCADD is based on probabilities, not on impact scores themselves. The logistic regression model underlying FlyCADD is set up to differentiate between simulated (set 2) and derived (set 1) variants and returns two probabilities for each variant it scores: the probability of a variant belonging to set 1, and the probability of the variant belonging to set 2. Compared to set 1, set 2 is enriched in impactful variants as natural selection has not acted upon these simulated variants, therefore, we have chosen to report the probability of a variant belonging to set 2 as a proxy for functional impact. The two probabilities returned by the logistic regression model equal to 1 for each prediction, and thus, adopting the reciprocal will not change the interpreted impact of a variant. On lines 224 - 226, we have rephrased the description of set 2 and on lines 234 - 236 we've added context regarding the probabilities as computed by the logistic regression model.

The issue with the two sets is not merely the interpretation of the probability to belong to one or the other but the meaning of it. The interpretation of set2 as being enriched with deleterious mutants is an extrapolation that should be tested regarding non-Drosophila organisms.

With FlyCADD, we built upon 10 years of CADD development and implementation. The design of the two sets of training variants is based on the CADD framework principle from 2014 (humanCADD) and 2020 (pigCADD), and extensively described again for the updated humanCADDv1.7 (Schubach et al., 2024). The CADD framework with these exact types of derived and simulated variant sets (called "proxy-benign" and "proxy deleterious" (Schubach et al., 2024)) has been extensively used in these, and subsequent, papers. This assumption regarding the purging of impactful mutations and fixation of neutral or benign variants is strongly based on population genetics theory. Exactly as originally developed and later described in Schubach et al. 2024, we set up our variant sets to reflect this, and, thus, this interpretation of these sets is commonly used terminology in this field.

Moreover, the fact that the proportion of predicted low-impact scores is small questions the pertinence of the design choices for Set1 and Set2. According to the literature, one would expect most of the variants to be neutral/benign or well tolerated. It seems that the design adopted by the authors, imposing strict

criteria for Set1 and more relaxed ones for Set2, leads to over-predicting pathogenicity (or functional impact).

The reviewer correctly points out that one would expect most naturally occurring variants to be not impactful. However, it is important to note that the dataset we present in Figure 3 is not a representation of tolerated or naturally present variants but rather shows the FlyCADD scores across all possible variants (> 400 million variants). Figure 7 does reflect naturally occurring variants from DEST2.0 and indeed shows that most variants are neutral and benign as expected (~ 4.5 million variants). We have added clarification on the nature of the dataset depicted in Figure 3 at lines 435 - 440.

A likely explanation could be an over estimation of high impact scores due to a) the low percentage of homology between the reference genome and the inferred ancestral genome and b) taking only the Drosophila genomes as a reference for deciding what polymorphism hasn't been explored. Representing the distribution of the scores specifically on non-lethal mutations compared to that of the lethal ones would help figuring that point out.

As stated in our initial response letter and in the manuscript, the dataset in Figure 3 is not a representation of solely naturally occurring SNPs, but shows all possible variants on the entire *D. melanogaster* genome with its predicted impact. With Figure 8 we show the distribution of FlyCADD scores of SNPs in natural populations. Therefore, we believe that the occurrence of predicted high impact scores is not an overestimation or artefact of model training, but rather a feature of the model and variants themselves.

- since the conclusions of the paper are that the most significant SNPs actually fall within coding sequence, why not comparing with the recent cited ProteoCast that restricts its predictions to the proteome. We have demonstrated the use of FlyCADD across the entirety of the *D. melanogaster* genome and do not intend to conclude that the most significant SNPs fall in coding sequence from the presented analyses. However, we do acknowledge that the most predictive features include features restricted to coding region. With the inclusion of combinatorial features, we ensured predictive power present irrespective of the genomic position of variants. The comparison with ProteoCast presents itself with several difficulties regarding the aim of the predictive tool, the range of applications and the continuous or binarized score approach. The comparison of FlyCADD with other VEPs is outside of the scope of this manuscript. At lines 115 – 119, we describe these different properties of VEPs. However, with the inclusion of FlyBase lethal point mutations, we have validated our FlyCADD scores in a similar manner as ProteoCast, with similar outcomes. We have acknowledged this in the discussion at lines 737 - 742.

We find it hard to defend the argument for not comparing with other methods as being outside the scope of this paper. As the authors rightfully emphasize, FlyCADD is designed and trained specifically on one organism/species with a large number of organism-specific annotations. Hence, one would expect that it performs much better than general-purpose VEP on this particular organism. It is thus essential that the authors quantitatively assess the gain in predictive performance of FlyCADD compared to these VEPs.

We agree that a benchmarking study for various effect predictors and in multiple species would be valuable, but this will not be a part of this manuscript. Benchmarking is outside the scope of this manuscript as we show an application of the CADD framework to *D. melanogaster*. The CADD framework has been extensively benchmarked against various variant effect predictors using human, pig, mouse and chicken data. However, as we have previously indicated in our response and in our previous rebuttal letter, for *D. melanogaster* we lack a benchmarking dataset for validation with curated, functionally validated variants of known consequence making meaningful benchmarking specifically for this species a separate study. We did include an extra section to the Discussion describing the benchmarking previously performed for CADD framework, now on lines 796 - 829.

Response to Reviewers

Reviewer #1

Dear authors, thank you very much for the extensive revision of your manuscript "Predicting the functional impact of single nucleotide variants in *Drosophila melanogaster* with FlyCADD". My concerns are fully addressed. Special thanks for the provision of a hands-on protocol, which will be very helpful to access the data. Maybe I missed it, but the protocol does not seem to be uploaded to the zenodo repository as mentioned in the revised manuscript. This should be done prior to publication of the manuscript. Greetings, Nico Posnien

We thank the reviewer for their comments that guided the revisions of our manuscript. The step-by-step protocol has now been added to the Zenodo repository.

Reviewer #2

We would like to thank the authors for having taken into consideration most of our suggestions. The additional validation experiments they performed clearly emphasize limitations of the method that should be discussed in the main manuscript text. While we continue to think that the tool represents a deep interest to the *Drosophila* community, multiple points still require attention. We present an overview here for clarity, followed by a full answer inline with the authors' responses.

We appreciate that the reviewer acknowledged the value of the revisions we have made to the manuscript. We have responded below to the points still suggested by the reviewer.

1) The functional validation on a set of known lethal mutations only provides a recall estimate. It is impossible to compute or assess predictive accuracy in the absence of a negative set (i.e. non-lethal mutations). Moreover, the authors did not provide the list of lethal mutations they used, which compromises reproducibility.

Validation with lethal mutations was added based on the suggestion of the reviewer during the first review. A validated negative set containing non-lethal mutations does not exist, as every nucleotide present in the genome of living individuals is by definition non-lethal. Non-lethal does not imply non-impactful, which is what is scored by FlyCADD. Therefore, such a negative set is also not included in our manuscript.

By definition, observed mutations are non-lethal as duly noted by the authors, so a set of randomly chosen DEST or DGRP SNPs is likely to have less impactful mutations than those known to be lethal. Despite the absence of a negative validation set, we argue that the analysis of validated lethal mutations provides important information about the performance of the model as we have detailed in our previous response letter. If the reviewer disagrees with this, this analysis can also be moved to supplementary material. Further response to the concerns regarding FlyCADD performance can be found in response to comment 2.

Suggesting to move the proposed validation to the supp mat would not solve the core matter which is the absence of a "neutral" validation set. We disagree as previously explained.

The reviewer is correct that the predictive accuracy cannot be computed in an independent dataset. We addressed this in the Results (line 524-536) and only described the metric calculated using a held-out testing dataset as "accuracy".

We thank the authors for this correction.

We regret that we forgot to add the list of lethal mutations to our previous resubmission. We now added the list of these mutations and their FlyCADD scores as used for Figure 7 to Zenodo to ensure reproducibility of this experiment (as indicated on line 380).

We thank the authors for this correction.

2) A certain number of vague points still remain to be addressed, the over inflation of descriptive features compared to previous CADDs as well as their redundancy, the data circularity and data leakage issues still require a thorough description of the training/test sets, the term accuracy is used multiple times out of context, no accuracy is presented (see point 1).

We regret that a number of points is still not clear to the reviewer. However, we feel it is difficult to respond to this comment because it is not very specific. We tried to dissect the criticism into the following separate concrete points:

- Overinflation/dimensionality: The number of categories ($n = 38$) and the number of features ($n = 691$, includes combinations of annotations) in FlyCADD is not overinflated compared to the scale of features used in other CADD implementations, such as humanCADD or pigCADD, and thus does not represent an unusual increase in dimensionality or overinflation. For example, humanCADD reports the implementation of 63 annotation categories, which result in 949 features, mouseCADD 68 annotations making up 931 features, and pCADD reports 39 annotations resulting in 867 features.

- Redundancy: As indicated in our first response letter, the redundancy of the features is assessed using the weight of each feature in the model. We describe that all annotations are involved in the final score, mostly through the incorporation in combinatorial features.

- Data circularity: This comment seems to be related to the DEST2.0 variants in Figure 8. These variants are presented as use-case of FlyCADD, not as validation of the impact prediction scores and, therefore, data circularity is not a valid concern.

- Data leakage: The training procedure was described in our Methods section and in our first response letter. Data leakage would imply the use of training data during model testing. We kept the held-out test-set of 10% separate from model training at all times, to ensure data leakage would not occur. We have now rephrased the Methods section accordingly to avoid this misunderstanding (described in response to the comment below, line 325).

- Accuracy: Unfortunately, the reviewer does not provide us with line numbers of the incorrect usage of the term “accuracy”. Therefore, we checked all occurrences across the manuscript and changed one where the term was not used correctly (line 546). For clarity, we define at first occurrence in line 341 what is meant with accuracy, in concordance with previous CADD papers. We present the accuracy of the model based on the held-out dataset. Accuracy is the overall performance of the model, and an additional measure of model performance is AUC-ROC. We already provided both (as computed based on the held-out testing dataset) in the manuscript at lines 400 – 402.

Moreover, as we described in our previous rebuttal letter, we added lethal mutations as a third metric for model performance. This metric is recall as the reviewer points out, however, it was not presented in our manuscript as accuracy metric, but rather as performance metric. To avoid confusion with the term “accuracy”, we have corrected the description of the additional validation analysis to properly reflect the outcome from lethal mutations, namely the recall metric (line 546).

- Overprediction: The reviewer suggests overprediction of variant effect by FlyCADD, however, this is not shown in our analyses (more details provided in the comments below).

3) The ability of FlyCADD to make biologically or clinically relevant predictions outside of CDS remains unclear

We feel that the reviewer overlooks the evidence for relevant predictions across an entire genome using previous applications of the CADD framework. We agree not everything is known yet about non-coding regions, but we did show for example relevant predictions in agreement with functional studies in non-coding regions (Figure 10) and functionally relevant variants genome editing experiments outside coding region (final use-case). As emphasized before, a substantial validation dataset is not present, but the model is built upon a thoroughly validated and benchmarked framework.

To better emphasize the benchmarking efforts done for CADD framework performance in coding and non-coding sequence, we have added a paragraph in the Discussion. This section discusses the previous applications and benchmarking of the CADD framework (in human). We emphasize that there is no validation dataset available to test this for *D. melanogaster* directly, but that the CADD framework has been widely adopted.

The authors clearly show that the

4) Comparison with other existing methods, which is essential for assessing the pertinence and relevance of the proposed novel method, is still missing from the manuscript. The recent literature about variant effect predictors is extremely active and abundant, with community-wide efforts for benchmarking novel methods. Since the central and main contribution of the manuscript is a novel method for predicting variant outcome, the authors should more carefully acknowledge relevant literature and explain how their proposed methodology compares to others (including those that use much less features or rely on unsupervised language models).

We agree with the reviewer that comparing existing VEP methods is important. The CADD framework is one of these existing VEP methods and has been benchmarked in multiple studies using multiple methods, which is why we now refer to the available comparisons between CADD and other VEPs in the Discussion (lines 795-829). We remain with our opinion that benchmarking is beyond the scope of this paper due to both technical limitations regarding datasets as well as its incompatibility with the scope of this study.

It is unclear whether the changes made by the authors to transpose CADD to flies would affect the capabilities of the framework. In addition, it is very hard to motivate that publishing a novel method, let it be the adaptation of an existing one to a novel context, does not require a comparison with existing methods.

Nevertheless, as stated above, the authors do what they want with their manuscript and we will not go through endless revisions motivated by personal convictions.

1) FlyCADD scores can indeed address key questions in genotype-phenotype interactions

Associate Editor:

(a) Clarify the specific aspects that FlyCADD scores can indeed address key questions in genotype-phenotype interactions, discussing the limitations and the need for fine-scale genetic mapping to resolve certain complexities.

We have now added more nuance regarding the complexity of the genotype-phenotype link and the limitations of FlyCADD in research related to this topic. FlyCADD is intended as complementary tool to association studies and functional genomics by providing an impact prediction score to understand functional variant impact. Additionally, we added an analysis of known lethal point mutations to demonstrate the correlation of predicted impact by FlyCADD with empirically validated functional effects,

as suggested by reviewer #2.

Authors should clearly state the methods, papers and analyses that inspired their validation in the main body of the preprint. In addition, no information is given regarding the specificity of those predictions. They should calculate an accuracy of prediction using the distribution of scores for non-lethal mutations. In addition, no table with the used mutations and their predicted scores is presented. How can one reproduce this analysis?

This response is a continuation of our response to Overview 1 above.

We now state the basis of our validation more clearly:

- At lines 372-374, we specify the goal of the inclusion of lethal mutations.
- At lines 524 – 536, we elaborate on our validation using lethal mutations.

- At line 534, we specify that specificity is not calculated.

The validation method does not include specificity as a curated set of known benign/neutral mutations is not available for *D. melanogaster*. Therefore, we only present model performance metrics based on the held-out set (accuracy and AUC) and performance on a curated dataset of lethal mutations.

As discussed in point 1) and suggested by ourselves during the first revision, it is totally legit to use viable mutations as a control for lethal ones. It does not imply that the latter is “impactful” while the former is “neutral”, simply that there is a difference to be expected in the scoring of both.

2) Address points the reviewers identified as vague

a) over inflation of categories compared to other CADDs (x10)

b) data circularity / data leakage

2/ The approach builds upon the variant effect predictor (VEP) CADD (doi: 10.1038/ng.2892), which underperforms compared to other VEPs in recent benchmark studies, e.g. see doi: 10.15252/msb.202211474 or proteigym.org). Moreover CADD is a supervised method, introducing the risk of data circularity.

FlyCADD indeed builds upon the CADD frameworks established for application to the human genome and the pig genome (pCADD). The original CADD framework has been developed to predict the functional impact of human genetic variants, with a focus on pathogenic variants. The paper referred to by the reviewer is a benchmarking study showing a comprehensive comparison of many VEPs. However, many of the higher performing VEPs are developed specifically i) to predict pathogenicity, ii) to be applied in the human genome, or iii) to make predictions predominantly for coding sequence, whereas the CADD is genome-wide VEP with the potential for species-specific application. Since our goal is to assess functional impact, genome-wide in *D. melanogaster*, rather than quantify pathogenicity, the CADD framework is chosen. It is possible that other methods have more predictive power for coding sequence variants, however, CADD is capable of ranking variants outside of coding regions, and species-specificity increases its predictive power on these respective genomes compared to tools specifically designed focused on a single genome/species, lines 133- 135. We now refer to the recent benchmark studies comparing traditional VEPs and CADD to address these issues at lines 129 – 133. While CADD is indeed a supervised method, the risk of data circularity is minimal due to the nature of its training data. Unlike models trained directly on clinically labelled pathogenic and benign variants - where the same datasets are often reused for both training and evaluation - CADD is trained to distinguish between naturally derived variants and simulated variants. This framework is evolutionarily informed, rather than clinically annotated, and does not include pathogenicity labels or known functional variants in training. As a result, CADD minimizes data circularity: it does not train on the same variant categories it is later evaluated on. In Figure 4, the consequences that are used to stratify the variants are VEP consequences, these are incorporated as annotations in the FlyCADD model and, therefore, could be a minor source for circularity within the figure. In our application, we further reduce potential bias by ensuring the training dataset spans the entire genome and is not manually enriched for specific chromosomal regions or even coding regions, which would introduce biased predictions in these regions. Importantly, we do not use the training data to evaluate the FlyCADD model or benchmark predictions. The dataset was split to use 90 % for model training and 10 % for model evaluation. Thus, the supervised nature of CADD does not introduce circularity in our context. The additional validation and score evaluations show the performance of FlyCADD without indications for circularity.

As far as we understood the manuscript, this statement is wrong. The only AUC and accuracy reported in the paper were computed on the 10% test data held out during training; this evaluation is performed on exactly the same categories it was trained on.

This is a misunderstanding of our previous response, which was not about the testing dataset but rather

about the score evaluations such as score distributions and codon impact.

The additional statement about AUC and held-out dataset is exactly what we have done, reported and how this logistic regression model should be evaluated. The procedure for model validation in the absence of a distinct and curated validation set is to test model performance based on a held-out set of data. This evaluation reports the model's ability to distinguish between these two variant categories on unseen data drawn from the same distribution, and therefore, the training and testing dataset have the same categories, on purpose. For FlyCADD, this is what we have done, and the 10% held-out test set shares the same variant categories ("derived" and "simulated") as the training data to report the model's ability to distinguish these, the model metrics are already reported as such in the manuscript (lines 401-402 and 344-346).

Moreover, it is not clear whether the 46 DEST1 used for training are included in the 529 DEST2 reported during evaluation: it seems to be the case, which would imply data leakage between train and test: the variants used in set 1 of training again appear in the DEST2 evaluation.

Again, this seems to be a misinterpretation of the reviewer of the section about case studies. This section is meant for demonstration purposes on the application of FlyCADD, not for model evaluation. Data leakage would mean the usage of training data for testing purposes. Figure 8 is presented as use-case and not as validation method for the scores, therefore, it does not act as testing method to determine specificity and does not induce data leakage.

Validation or simply presenting a use-case, the training and test sets should have no overlap. If the authors confirm that then the approach is valid.

6/ The description of the validation of the model is unclear. Did the author perform k-fold cross-validation on 90% of the training set and performed the blind final evaluation on the held-out 10%? If yes, what value of k did they choose? Did they average the models trained on each fold? If not, then why do they mention "cross-validation"? And why was the model still modified after this cross-validation (lines 300- 305)?

The reviewer rightfully points out the unclear model validation. We have realized that the term cross-validation was not appropriate in this context. We applied sub-sampling validation to determine the optimal L2 penalty value, followed by training of the final model. Specifically, we split the dataset (90% training, 10% testing) five times for each value of L2 (0.01, 0.1, 1.0, 10.0, 100.0) and selected the model with the highest average accuracy. This process was intended solely for hyperparameter tuning. After identifying the optimal L2 penalty, we trained a final model using the optimal parameters on the training set and evaluated it on the held out 10% test set. We did not modify the model after training. We have clarified this workflow in lines 322 - 326.

We would like to thank the authors for clarifying the training procedure. Their detailed description suggests data leakage between the train and test. Indeed, as far as we understand, the authors did not go for k-fold cross-validation, nor for a train/validation/test framework. The same dataset set, splitted in several 90/10 partitions (without control on the overlap between the split rounds) was used for both optimising the hyperparameters and evaluating the model's performance. Proper evaluation requires testing on an independent dataset that was not used for hyperparameters optimisation. The only independent set seems to be the lethal mutation sets that we suggested, but there are no negatives.

We regret that our extensive description of model training and testing was unclear. To avoid any misinterpretation, we updated the Methods section describing the training procedure (line 325). The entire training dataset encompasses 5.467.390 variants and was initially split into 90% for training and 10 % for testing. The 10 % testing dataset was held out and not used for any model training or hyperparameter tuning, serving solely for final evaluation of model performance through the reported ROC-AUC.

The hyperparameter tuning was performed exclusively on the 90% of the dataset that was allocated for training. This training subset was internally partitioned into 90/10 sets for optimizing the parameter.

After identifying the optimal parameter, the final model was trained on the entire 90% of the training dataset using this L2 value and then evaluated on the held-out 10 % test set. This approach ensured that the test set remained independent of any model training or hyperparameter optimization, thereby preventing data leakage and providing an unbiased metric of model performance. We have now stated this explicitly on lines 326-335.

We thank the authors for this modification of the manuscript.

c) poor explanation of the categories redundancy

5/ The model combines almost 700 features which limits its interpretability and practical usability. What is the level of redundancy? Could the authors simplify the model and achieve the same performance? While the authors claim on multiple occasions that this complex framework provides a predictive advantage, they actually do not compare the performance of FlyCADD with other methods. How does FlyCADD compare with approaches solely relying on evolutionary conservation, for instance? The authors could consider ProteoCast for proteins (scores readily available for the Fly proteome), GPN-MSA for alignment-based genome-wide predictions, or DNA language models (Evo2) that capture evolutionary conservation without the need for alignments?

The reviewer rightfully points out the large number of annotations incorporated in FlyCADD. When considering redundancy, both the individual annotations and combinations thereof should be considered, and we do think that the level of redundancy is not major. Not all included annotations provide high predictive power when included as individual feature, however, many combined show high predictive power in the model (File S3). For example, PhyloP124 has relatively low predictive power (-0.073), whereas its power is increased when combined with several VEP consequences such as conservation scores in intronic regions (-0.187). Additionally, the concern regarding redundancy has been systematically assessed in the publication describing mouseCADD 10.1186/s12859-018-2337-5), they have shown that the models' performance drops when removing features, especially removal of Ensembl VEP and conservation features.

A CADD model where the number of features is reduced might suffice, however, meaningful annotations to improve the predictions in specific regions are present such as secondary structure predictions or conservation scores, and they show discriminative power through combinatorial features as well. The basis for creating a CADD model is the benefit of combining all available types of annotations into one predictive score, where the features and combinations are weighted based on their performance. The reviewer's comment is valid, in the sense that some hypotheses are more robustly assessed with specific annotation(s) than others. The last years, it has been common to assess ultra-rare highly deleterious variants in human genetic studies by manually assessing several types of annotations and keeping only those with overlaps. For targeted studies, this might very well be the best approach. However, in a more exploratory study, or a hypothesis free such as prioritizing associated SNPs of a GWAS, combining all assessments into one framework, can be much more suitable. As the reviewer states, however, there is no true answer to the correct amount of input.

Dimensionality reduction methods tend to be highly sensitive to the addition of too many extra dimensions, especially if some of them are highly correlated. If we consider previous implementations of CADD, categories are in the range of 30-40, making FlyCADD increasing by an order of magnitude. A proper quantification of its impact should be proposed, especially considering that 95% of them show a low incidence on the model.

As we have stated in our response to Overview 2, this comment is not correct. Previous implementations

of CADD used similar numbers of annotations and features. As reported on lines 235-236, FlyCADD logistic regression model is trained on the 691 combinatorial features, encompassing 38 individual annotations (Supplementary File S2). This distinction between the number of annotations and features is important to understand the dimensionality of these categories which the reviewer refers to. The previous implementations of CADD indeed report numbers of categories in the range of 30-40, which is in fact similar to the number of annotations implemented in FlyCADD ($n = 38$) and thus FlyCADD does not represent an unusual increase in dimensionality.

Throughout the manuscript, we refer consistently to the 691 dimensions as “features” and to the 38 categories as “annotations” to indicate this distinction. All features are included in a genome-wide manner, each position has a value for all features. Feature importance is discussed by presenting their weights (line 409, Supplementary File S3), indicating that all annotations contribute, especially as part of the combinatorial features.

We thank the authors for clarifying this point.

d) distribution of scores across the genome and quality of the score

We would like to thank the authors for showing the distribution of all FlyCADD scores on Figure 8. This figure is highly informative. The long-tailed distribution seems to suggest that the beta risk for a SNP with a score >0.9 would be close to 80%. The figure 9 indeed shows that SNPs associated with significant p-values are present across the whole range of FlyCADD scores. These elements support our hypothesis that FlyCADD overpredicts variant effects and does not allow us to understand how the thresholds were defined for SNPs of interest. A clarification would be needed in order to guarantee the confidence in predictions.

It is unclear to us how the reviewer gets to 80 % beta risk; therefore, we cannot respond to this comment. The thresholds proposed are not cut-off values as FlyCADD scores are a continuous measure with the goal of variant prioritization rather than exact scoring. This is indicated multiple times in the manuscript. Additionally, it is unclear to us how overprediction by FlyCADD is detected, our analyses do not indicate overprediction of variant effects by FlyCADD, as shown by several use-cases, validation analysis and model metrics.

We estimated an approximate beta error from figure 8a distribution. We have clearly understood that FlyCADD scores are continuous measures for variant prioritization. However, as we have indicated in our previous comment, the scores are present all over the place, whatever the considered p-value, which makes it hard to prioritize any SNP.

e) comparison with existing methods

5/ [...] How does FlyCADD compare with approaches solely relying on evolutionary conservation, for instance? The authors could consider ProteoCast for proteins (scores readily available for the Fly proteome), GPN-MSA for alignment-based genome-wide predictions, or DNA language models (Evo2) that capture evolutionary conservation without the need for alignments?

That part of the question was not answered and it seems of importance that novel methods, even though an adaptation of an existing method to a new organism, are compared with existing ones. In our review we suggested ProteoCast as it defined a dataset for testing drosophila mutational impact prediction functionally, but methods predicting mutational impacts outside of the coding region exist, whether they are based on alignments or language models.

As indicated in the Overview, a full comparison with existing effect predictors is out of the scope of this manuscript. We do acknowledge the presence of alternative methods, with their unique areas of focus compared to FlyCADD in our manuscript, as indicated in our first response letter and now in the Discussion. These tools have different goals, focus species and outcome datatypes, which makes it necessary to conduct a separate benchmarking study for thorough comparison.

We still disagree with that point, proposing a new method should be accompanied with a comparison of that method with existing ones.

(d) There are a number of questions and comments about the ancestral sequence reconstruction that need to be addressed.

We thank the editor and reviewer for pointing out the difficulties with the description of the ancestral sequence reconstruction. We compared different nodes in the 166-way phylogeny to select the node with the largest ancestral sequence size, least chromosome bias, and least coding sequence bias. This ultimately resulted in an ancestral sequence that matches 16.4 % of the reference genome, while the other reference genome positions correspond to gaps in the ancestral sequence. We have rephrased the ancestral sequence reconstruction description in File S1, as well as in the Methods section to clarify these choices.

It is not clear whether the low percentage of homology between the reconstructed ancestral sequence and the reference sequence can explain the over representation of high scores? This issue should be discussed with the plotting of scores for non-lethal mutations as suggested in comment 1).

It is not clear to us why the reviewer suspects overrepresentation of high scores. This is not indicated from our analyses. Overrepresentation of high scores can be a result of unbalanced or biased training datasets. Importantly, our model was trained using a balanced dataset comprising equal numbers of simulated and derived variants, designed to prevent bias toward high or low scores. Ancestral alleles were filtered as described in “Methods” to represent derived variants. Additionally, we assessed and minimized biases related to chromosome representation and coding sequence content during the selection of the ancestral node, as described in Supplementary File S1. Thereby, we minimized the biases that could result from incomplete ancestral sequence reconstruction. At present, we have no evidence that regions lacking ancestral sequence coverage are enriched for functionally impactful variants or that they introduce systematic bias into the model.

We do not suspect an over-representation of high scores; distribution of scores in figure 8a shows that the proportion of scores > 0.5 is only 30% lower than those below 0.5, which is also seen in 8b.

3) Impact of FlyCADD predictions outside of CDS

4/ The authors emphasize in the introduction that FlyCADD is not limited to variants in coding regions. Yet, protein coding associated features are among the most contributing ones (e.g. gain of stop codon (Figure 2) : there is an order of magnitude between the 1st parameter and the 3rd,, missense mutations), highlighting the functional importance of variants located in these regions.

While FlyCADD is not limited to variants in coding regions, the strongest predictive features indeed relate to coding consequences, such as “stop gained” or “missense” mutations. This likely reflects a combination of factors: stronger evolutionary constraints on coding variants, leading to clearer depletion signals in the observed (nearly fixed) dataset; and deeper and more consistent annotation coverage for coding regions in existing databases. Consequently, coding sequence associated features can offer greater discriminative power within the model. However, features related to coding sequence also indirectly inform the model about non coding variants. This is because non-coding variants are typically assigned a neutral or zero value for such features, which still carries information: their non-coding nature becomes part of the predictive signal. In this way, coding-related annotations contribute to the model’s discriminative power for all variants.

The phrasing should thus be toned down as FlyCADD then predicts mostly the impact of coding variants. This matter was previously described for other implementations of CADD.

We respectfully disagree with the reviewer on this point. As FlyCADD is both trained on coding and non-coding training variants, as well as coding and non-coding annotations, the predictions FlyCADD makes are based on this integrative approach combining these large sets of data. The notion that FlyCADD

scores might be more empowered in coding regions, due to more annotations for coding regions is emphasized multiple times in the manuscript already (line 136, 807). This does, however, not automatically imply that the impact predictions for non-coding regions are less reliable. At the state of the art, FlyCADD is the only prediction tool for *D. melanogaster* that integrates annotations on non-coding regions. This makes it a valuable tool to investigate impact of non-coding variants, which is a largely unexplored field to date.

We have noticed that FlyCADD was trained on both types of sequences alike; however, we have also noticed that the weight of parameters related to coding sequences, especially stop codon gain/loss, dramatically outweigh (approx. 10X) those corresponding to non-coding sequences.

4) Implementation of FlyCADD

(e) Consider usability of FlyCADD by others in the field; this is a key part of the future impact of this work.

We agree with the editor that usability of FlyCADD is a key part of its future impact. As setting up a web interface is currently beyond our resources, we added a step-by-step protocol to improve the usability of the resources FlyCADD provides. Future developments may include a web interface, potentially in collaboration with FlyBase.

We thank the authors for making FlyCADD reusable. FlyBase is right now unable to integrate such work considering their lack of funding as far as we aware of.

We agree that recent developments at FlyBase make it unfortunately uncertain if and when FlyCADD could be integrated. To improve usability, the step-by-step protocol is now added to the Zenodo repository.

We thank the authors for this.

5) scores interpretations

Moreover, the interpretation of the weights is not straightforward. While it is easy to understand that premature stop codon contributes negatively, how can the authors explain that stop codon loss has a high positive score? Is this a numerical or statistical artefact? If one looks at Figure 4, variants annotated as stop codon gain or stop codon loss have both very narrow distributions toward high scores (meaning high impact). Is there an enrichment in mutations altering stop codons in the training set? Do the features include stop codons that are outside the CDS? What is the value of relCDSpos in that case? In the seminalCADD paper (doi: 10.1038/ng.2892), the authors construct a series of univariate models that contrast observed and simulated variants. This allows for exploring the training set for potential biases and it could also be useful here. Importantly, the weights shown in Figure 2 reflect the discriminative power of each feature within the logistic regression model, not the biological impact or directionality of the variant itself. For example, the high weight assigned to the “stop loss” feature does not imply a beneficial effect but rather indicates that this annotation strongly helps distinguish simulated from observed variants. The score distribution (Figure 4) shows what the model predicts; the feature weight shows how that variable helps make the prediction. Feature weights describe how informative a feature is for classification in the context of all other features and should not be interpreted as stand-alone indicators of biological effect. To clarify this, we have revised the main text (lines 404 – 409 and 418 - 426) to better explain how feature weights reflect discriminative power, not impact direction or magnitude. For the same reason, we have also updated Figure 2 to more accurately reflect the predictive power of each feature in the model. We emphasize that the direction of the weights reflect the way our annotations were encoded and do not imply biological meaning at lines 331 - 334. Regarding stop codons outside of coding region, stop codon features are not present outside coding sequences, and therefore our features do not include such stop codons. The Ensembl VEP is used to annotate these features, and their severity order was applied in case variants could be annotated with multiple

consequences. The feature “relCDSpos” indicates the position within coding sequence for those variants that are in coding sequence. This feature remains empty for those variants outside of coding regions (the value is 0 in the encoded annotation output). However, this value is also used by the model to discriminate variants. Enrichment of coding-related features (such as stop codon gain or missense mutations) in the top contributors reflects the biological relevance of these sites and is not an artifact. The relevance can be interpreted as a depletion of loci in the observed set of (nearly) fixed derived variants, likely because of purifying selection against such mutations on a longer evolutionary timescale. The simulated training set intentionally contains high-impact simulated variants, not to balance the classes, but to maximize discriminative signal. Note, however, that the frequency of the high-impact variants in the set of simulated variants is determined by mutation rate and is determined randomly by the simulator.

Therefore, we did not artificially enrich this set with high-impact variants but simply rely on the proportions that arise under a neutral scenario, without (purifying or balancing) selection. As indicated on line 286 – 288, we have added a figure to File S3 reflecting the proportions of variants with different VEP-annotated consequences in the derived and simulated training datasets.

We would like to thank the authors for clarifying the meaning of the weights’ signs. The authors chose to represent predictions for mutants of the white gene and we previously highlighted that they did not represent functional variants. They gave a partial response with the sentence below in red.

lines 465 - 467 + Figure 5 : the authors suggest that their tool can help discriminate variants within the same region and give the example of the white gene but do not represent known functional variants on the sequence. As stated above, very limited numbers of SNPs are functionally validated in *D. melanogaster*. Figure 5 is included in the manuscript to illustrate the patterns within genes, such as intronic and exonic regions. Based on a manual annotation one would assume exonic variants are more impactful than intronic variants or even intergenic variants. FlyCADD can be applied to discriminate variants within these regions, as can be seen by variants spanning the entire range of FlyCADD scores within exons and within introns. However, we do agree with the reviewer that the depiction of only theoretically present variants invariants in the gene white, namely all possible variants across this region on the X chromosome, becomes more informative when known variants are presented. Functional variants that are experimentally validated within this region are rare. Therefore, we added naturally occurring variants of the DEST2.0 resource to the figure. The figure now includes all possible variants in purple, and highlighted with blue diamonds are all variants naturally occurring in worldwide *D. melanogaster* populations, as indicated in the figure legend and at line 473.

Multiple mutations are described in Mackenzie et al., 1999 with a functional impact. Could the authors locate them on their prediction map and indicate their predicted scores?

We thank the reviewer for pointing out these variants and we have updated Figure 5 in the manuscript (see updated Figure below), as well as line 485. These variants are phenotypically validated for functional impact and also are predicted to be of high functional importance by FlyCADD.

We thank the authors for this.

3/ If the "set of simulated variants is thought to contain the full spectrum of deleteriousness", then why is the "probability of the variant belonging to the simulated variants class" (Set2) considered by the authors as equivalent to the "predicted functional impact to fitness"? In other words, why would a variant likely to belong to Set2 necessarily have high functional impact? It would be more intuitive to adopt the reciprocal perspective, i.e., reason with respect to the neutral class: a variant with a high probability to belong to Set1 is likely neutral.

We regret that the reviewer finds the description and application of set 1 and set 2 not intuitive and have changed this section to clarify the design choices for the two sets. FlyCADD is based on probabilities, not on impact scores themselves. The logistic regression model underlying FlyCADD is set up to differentiate between simulated (set 2) and derived (set 1) variants and returns two probabilities for each variant it scores: the probability of a variant belonging to set 1, and the probability of the variant belonging to set 2.

Compared to set 1, set 2 is enriched in impactful variants as natural selection has not acted upon these simulated variants, therefore, we have chosen to report the probability of a variant belonging to set 2 as a proxy for functional impact. The two probabilities returned by the logistic regression model equal to 1 for each prediction, and thus, adopting the reciprocal will not change the interpreted impact of a variant. On lines 224 - 226, we have rephrased the description of set 2 and on lines 234 - 236 we've added context regarding the probabilities as computed by the logistic regression model.

The issue with the two sets is not merely the interpretation of the probability to belong to one or the other but the meaning of it. The interpretation of set2 as being enriched with deleterious mutants is an extrapolation that should be tested regarding non-Drosophila organisms.

With FlyCADD, we built upon 10 years of CADD development and implementation. The design of the two sets of training variants is based on the CADD framework principle from 2014 (humanCADD) and 2020 (pigCADD), and extensively described again for the updated humanCADDv1.7 (Schubach et al., 2024). The CADD framework with these exact types of derived and simulated variant sets (called "proxy-benign" and "proxy deleterious" (Schubach et al., 2024)) has been extensively used in these, and subsequent, papers. This assumption regarding the purging of impactful mutations and fixation of neutral or benign variants is strongly based on population genetics theory. Exactly as originally developed and later described in Schubach et al. 2024, we set up our variant sets to reflect this, and, thus, this interpretation of these sets is commonly used terminology in this field.

We are well aware of the concept of purging of impactful mutations that we do use in our approaches. However,

Moreover, the fact that the proportion of predicted low-impact scores is small questions the pertinence

of the design choices for Set1 and Set2. According to the literature, one would expect most of the variants to be neutral/benign or well tolerated. It seems that the design adopted by the authors, imposing strict criteria for Set1 and more relaxed ones for Set2, leads to over-predicting pathogenicity (or functional impact).

The reviewer correctly points out that one would expect most naturally occurring variants to be not impactful. However, it is important to note that the dataset we present in Figure 3 is not a representation of tolerated or naturally present variants but rather shows the FlyCADD scores across all possible variants (> 400 million variants). Figure 7 does reflect naturally occurring variants from DEST2.0 and indeed shows that most variants are neutral and benign as expected (~ 4.5 million variants). We have added clarification on the nature of the dataset depicted in Figure 3 at lines 435 - 440.

A likely explanation could be an over estimation of high impact scores due to a) the low percentage of homology between the reference genome and the inferred ancestral genome and b) taking only the *Drosophila* genomes as a reference for deciding what polymorphism hasn't been explored. Representing the distribution of the scores specifically on non-lethal mutations compared to that of the lethal ones would help figuring that point out.

As stated in our initial response letter and in the manuscript, the dataset in Figure 3 is not a representation of solely naturally occurring SNPs, but shows all possible variants on the entire *D. melanogaster* genome with its predicted impact. With Figure 8 we show the distribution of FlyCADD scores of SNPs in natural populations. Therefore, we believe that the occurrence of predicted high impact scores is not an overestimation or artefact of model training, but rather a feature of the model and variants themselves.

If the authors believe it. It could have been estimated by using *Drosophila melanogaster* annotated data of known lethals, known hypomorphs and natural polymorphism.

- since the conclusions of the paper are that the most significant SNPs actually fall within coding sequence, why not comparing with the recent cited ProteoCast that restricts its predictions to the proteome. We have demonstrated the use of FlyCADD across the entirety of the *D. melanogaster* genome and do not intend to conclude that the most significant SNPs fall in coding sequence from the presented analyses. However, we do acknowledge that the most predictive features include features restricted to coding region. With the inclusion of combinatorial features, we ensured predictive power present irrespective of the genomic position of variants. The comparison with ProteoCast presents itself with several difficulties regarding the aim of the predictive tool, the range of applications and the continuous or binarized score approach. The comparison of FlyCADD with other VEPs is outside of the scope of this manuscript. At lines 115 – 119, we describe these different properties of VEPs. However, with the inclusion of FlyBase lethal point mutations, we have validated our FlyCADD scores in a similar manner as ProteoCast, with similar outcomes. We have acknowledged this in the discussion at lines 737 - 742.

We find it hard to defend the argument for not comparing with other methods as being outside the scope of this paper. As the authors rightfully emphasize, FlyCADD is designed and trained specifically on one organism/species with a large number of organism-specific annotations. Hence, one would expect that it performs much better than general-purpose VEP on this particular organism. It is thus essential that the authors quantitatively assess the gain in predictive performance of FlyCADD compared to these VEPs.

We agree that a benchmarking study for various effect predictors and in multiple species would be valuable, but this will not be a part of this manuscript. Benchmarking is outside the scope of this manuscript as we show an application of the CADD framework to *D. melanogaster*. The CADD framework has been extensively benchmarked against various variant effect predictors using human, pig, mouse and chicken data. However, as we have previously indicated in our response and in our previous rebuttal

letter, for *D. melanogaster* we lack a benchmarking dataset for validation with curated, functionally validated variants of known consequence making meaningful benchmarking specifically for this species a separate study. We did include an extra section to the Discussion describing the benchmarking previously performed for CADD framework, now on lines 796 - 829.

We have proposed a solution for testing the functional impact, using known lethal mutants against polymorphism found in natural populations which, by definition are non-lethal - we do not imply that they have no functional impact, just that they are the control category for lethality.

October 2, 2025

RE: GENETICS-2025-308595

Dear Dr. Beets:

I am pleased to accept your manuscript titled "Predicting the functional impact of single nucleotide variants in *Drosophila melanogaster* with FlyCADD" for publication in GENETICS, pending minor revision.

Please submit your revision along with a brief description of how you modified the manuscript in response to the reviewers' concerns and suggestions, which can be viewed at the bottom of this email. Most important are to ensure that the final version includes explicit statements about the limitations of the work and need for benchmarking (and other types of validation mentioned by reviewer 2) in the final version of the manuscript. Please note that in the PDF file of comments from reviewer 2, there are incomplete statements on pages 3 and 11. I've been in contact with the reviewer about this, and they plan to send a corrected version next week that I will pass along to you at that time. Please do not submit a revision prior to receiving (and incorporating) this additional feedback.

I expect you should be able to submit a revised manuscript within 45 days. I anticipate that a suitably revised manuscript will be acceptable for publication, but I will make the final decision about whether or not it will need to be seen again by a reviewer after it is submitted.

When revising the ms., please make an effort to shorten it, because that almost always improves a manuscript. We urge authors to heed the advice of Strunk and White: "omit needless words"¹. Follow this link to submit the revised manuscript: Link Not Available

Thank you for submitting this work to Genetics.

Sincerely,

Patricia Wittkopp
Associate Editor
GENETICS

Approved by:
Stanley Fields
Senior Editor
GENETICS

Reviewer comments:

Reviewer #1 :

Thanks for the extensive revision!

Reviewer #2 :

This second round of revisions clarified the training procedure, etc., and we thank the authors. The work still lacks validation: estimation of "specificity" or FDR (false discovery rate) comparison with the state of the art which makes it difficult to estimate the contribution of the work. Our detailed responses are written in green in the attached pdf.

Associate Editor comments:

I will forward along the updated PDF from reviewer 2 as soon as I receive it (expected next week).

Revision letter – FlyCADD

We thank the reviewer and editor for these final comments and suggestions. We appreciate the time and effort invested in the evaluation of our manuscript. Given the limited number of remaining comments to address, we do not repeat the full previous rebuttal document here but focus on the remaining comments and corresponding revisions. Any line numbers refer to the cleaned manuscript PDF.

Response to editorial comments

- *Include explicit statements about the limitations of the work and need for benchmarking (and other types of validation mentioned by reviewer 2)*

These are important and valid statements to include. They are included in the paragraph “Variant effect predictors” (Line 789) in the Discussion section. In this paragraph, we discuss existing benchmarking efforts of the CADD framework and indicate how that relates to possible limitations of FlyCADD.

- *Shorten the manuscript: "omit needless words"*

Throughout the manuscript, we removed needless words or phrases. These changes can be found in the marked manuscript.

Response to reviewer comments

We have categorized the remaining suggestions of Reviewer 2 based on the four topics addressed and respond per topic.

Benchmarking

- *We still disagree with that point, proposing a new method should be accompanied with a comparison of that method with existing ones.*
- *(...) In addition, it is very hard to motivate that publishing a novel method, let it be the adaptation of an existing one to a novel context, does not require a comparison with existing methods. Nevertheless, as stated above, the authors do what they want with their manuscript and we will not go through endless revisions motivated by personal convictions.*

We regret that the reviewer does not agree with our response. We acknowledge the value of a comparison to existing methods in our manuscript at Lines 115-119 and 815-823, together with a description of the benchmarking studies where the CADD framework was included (from line 789). This is in line with our response to the request of the editor above.

Comparison to existing tools is important, however, should be carried out in a meaningful way. We lack a benchmarking dataset for *D. melanogaster* variant validation with curated, functionally validated variants of known consequence, therefore meaningful benchmarking specifically for this species, warrants a separate study.

Lethal or non-lethal point mutations as validation

- *By definition, observed mutations are non-lethal as duly noted by the authors, so a set of randomly chosen DEST or DGRP SNPs is likely to have less impactful mutations than those*

known to be lethal. Suggesting to move the proposed validation to the supp mat would not solve the core matter which is the absence of a “neutral” validation set. We disagree as previously explained.

- *As discussed in point 1) and suggested by ourselves during the first revision, it is totally legit to use viable mutations as a control for lethal ones. It does not imply that the latter is “impactful” while the former is “neutral”, simply that there is a difference to be expected in the scoring of both.*
- *We have proposed a solution for testing the functional impact, using known lethal mutants against polymorphism found in natural populations which, by definition are non-lethal - we do not imply that they have no functional impact, just that they are the control category for lethality.*
- *The set of known lethal mutations can be used as a surrogate validation dataset and can be mentioned as such.*

Naturally occurring variants are indeed non-lethal and less likely to have impactful mutations. The distribution of FlyCADD scores for lethal variants and naturally occurring variants differs greatly. We have added a description of the difference in predicted functional impact between lethal and non-lethal/natural SNPs at Lines 758-762.

The final comment on this topic refers to the use and description of known lethal mutations. We describe these as the only available dataset for testing, in the absence of a curated validation dataset (Lines 369-373, and Results 519-531). The description of lethal mutations as surrogate is thereby present in the manuscript.

Capabilities of the framework

- *It is unclear whether the changes made by the authors to transpose CADD to flies would affect the capabilities of the framework. (...)*
- *We are well aware of the concept of purging of impactful mutations that we do use in our approaches. However, it is hard to accept this as a valid argument for defining a set enriched in deleterious effects while rejecting the use of lethal mutants as a validation set for the predictions*

The CADD framework has been successfully transposed to a range of different species before, without affecting the capabilities of the framework. This indicates that species-specific applications of the CADD framework perform well. It has been shown that species-specific annotations and training datasets allow improved prediction, with every model relying on its own specific set of annotations. This was extensively analysed in the description of mouseCADD doi.org/10.1186/s12859-018-2337-5. By making use of *D. melanogaster*-specific annotations, genomes and additional resources, FlyCADD is custom made to ensure model fitting and functioning.

Regarding the second comment on this topic, we do not reject the use of lethal mutants as a validation set for the predictions. As described in response to the previous topic and based on the suggestions of Reviewer 2 in the previous rounds of reviews, we make use of this set of known lethal mutations to estimate recall, in the absence of a curated validation dataset that includes a range of phenotype-associated mutants specifically for *D. melanogaster*.

Overprediction by FlyCADD

- *We estimated an approximate beta error from figure 8a distribution. We have clearly understood that FlyCADD scores are continuous measures for variant prioritization. However, as we have indicated in our previous comment, the scores are present all over the place, whatever the considered p-value, which makes it hard to prioritize any SNP.*
- *We do not suspect an over-representation of high scores; distribution of scores in figure 8a shows that the proportion of scores > 0.5 is only 30% lower than those below 0.5, which is also seen in 8b.*

High FlyCADD scores are indeed present for naturally occurring variants, and don't indicate deleteriousness but rather functional impact. The first comment of the reviewer acknowledges that the (predicted) functional impact of SNPs might be different from what one would expect, and shows that FlyCADD provides new insights into this functional impact. SNP prioritization can be carried out by combining p-values and FlyCADD scores as the example described on Lines 615-619.

The threshold of 0.5 mentioned by the reviewer is, as discussed in the manuscript (lines 584-590 and 399-406), not a correct representation of the distribution of impact as FlyCADD scores are a continuous measure. We show that the predicted high-impact variants occur mostly amongst rare variants as expected, indicating that the pattern is meaningful (Lines 575-576).

October 31, 2025

RE: GENETICS-2025-308595R1

Ms. Julia Beets
Vrije Universiteit Amsterdam
Amsterdam Institute for Life and Environment
Van der Boechorststraat 3, 1081 BT Amsterdam
Amsterdam
Netherlands

Dear Dr. Beets:

Congratulations, your manuscript titled "Predicting the functional impact of single nucleotide variants in *Drosophila melanogaster* with FlyCADD" is accepted for publication in GENETICS! Many thanks for submitting your research to the journal.

To Proceed to Publication:

1. Format your article according to GENETICS style: <https://academic.oup.com/genetics/pages/author-guidelines>
2. Ensure that you comply with data and community resource citation guidelines:
<https://academic.oup.com/genetics/pages/author-guidelines#section-5-9-2>
3. Upload your final files at <https://genetics.msubmit.net>
4. Add oupsupport@scipris.com and genetics.oup@novatechset.com (or the domains @scipris.com and @novatechset.com) to your email program's "safe senders" list. You will be contacted by both at various points during the production process.

Notes:

- Your currently-accepted manuscript (unedited, as submitted, reviewed, and accepted) will be published at GENETICS and deposited into PubMed as an Advance Access article. Notify sourcefiles@thegsajournals.org before signing your license if you do not wish to publish your article via Advance Access.
- We invite you to submit an original color figure related to your paper for consideration as cover art. Please email your submission to the editorial office or upload it with your final files. You can submit a small-sized image for evaluation, and if selected, the final image must be a TIFF file 2513px wide by 3263px high (8.375 by 10.875 inches; resolution of 600ppi). Please avoid graphs and small type.
- After files are sent to Oxford University Press we use SciPris to manage article licensing and payment. If you do not have a SciPris account, you will receive an email from no-reply@scipris.com to sign up to use Oxford University Press' author portal. After logging in, follow the online instructions to sign your license and arrange any payment due.

If you have any questions or encounter any problems while uploading your accepted manuscript files, please email the editorial office at sourcefiles@thegsajournals.org.

Sincerely,

Patricia Wittkopp
Associate Editor
GENETICS

Approved by:
Stanley Fields
Senior Editor
GENETICS